# Strong and ductile titanium–oxygen–iron alloys by additive manufacturing

Tingting Song[1,7], Zibin Chen[2,3,4,5,7], Xiangyuan Cui[2,3], Shenglu Lu[1], Hansheng Chen[2,3], Hao Wang[2,3], Tony Dong[6], Bailiang Qin[4], Kang Cheung Chan[4,5], Milan Brandt[1], Xiaozhou Liao[2,3], Simon P. Ringer[2,3✉] & Ma Qian[1✉]

Titanium alloys are advanced lightweight materials, indispensable for many critical applications[1,2]. The mainstay of the titanium industry is the α–β titanium alloys, which are formulated through alloying additions that stabilize the α and β phases[3–5]. Our work focuses on harnessing two of the most powerful stabilizing elements and strengtheners for α–β titanium alloys, oxygen and iron[1–5], which are readily abundant. However, the embrittling effect of oxygen[6,7], described colloquially as 'the kryptonite to titanium'[8], and the microsegregation of iron[9] have hindered their combination for the development of strong and ductile α–β titanium–oxygen–iron alloys. Here we integrate alloy design with additive manufacturing (AM) process design to demonstrate a series of titanium–oxygen–iron compositions that exhibit outstanding tensile properties. We explain the atomic-scale origins of these properties using various characterization techniques. The abundance of oxygen and iron and the process simplicity for net-shape or near-net-shape manufacturing by AM make these α–β titanium–oxygen–iron alloys attractive for a diverse range of applications. Furthermore, they offer promise for industrial-scale use of off-grade sponge titanium or sponge titanium–oxygen–iron[10,11], an industrial waste product at present. The economic and environmental potential to reduce the carbon footprint of the energy-intensive sponge titanium production[12] is substantial.

Most industrial titanium (Ti) alloys possess microstructures based on the two basic phases of Ti, the hexagonal close-packed (HCP) α and the body-centred cubic (BCC) β. Represented by Ti–6Al–4V (wt% used throughout unless specified), α–β Ti alloys are the backbone of the Ti industry[1,2]. They can form microstructures comprising[2–5] (1) lamellar α–β with a near-Burgers orientation relationship, (2) equiaxed α and β or (3) globular α among the α–β lamellae. Each of these microstructures has merits and drawbacks, making α–β Ti alloys versatile for diverse industrial applications[1–5]. Of these, the lamellar α–β microstructure has been commonly applied.

The α–β Ti alloys are formulated by alloying Ti with α-phase and β-phase stabilizers. The α-phase stabilizers are limited to Al, N, O, C, Ga and Ge (refs. 3–5), of which N and C are tightly controlled impurities (0.05% N, 0.08% C)[2,3], whereas Ga and Ge are not commercially viable. Hence, as well as Al, O is the only other practical option. Supplementary Table 1 lists the main α–β Ti alloys using Al as the α-phase stabilizer. Notably, O outshines Al in (1) strengthening the α-phase by a factor of about 20 (calculated according to the data given in Table 4 on page 16 of ref. 1), (2) stabilizing the α-phase by a factor of about 10 (based on the aluminium equivalence formula given on page 380 of ref. 5) and (3) restricting the growth of prior-β grains during solidification by a factor of more than 40 (10.8 versus 0.26)[13]. However, these

attributes of O have remained underused in the development of α–β Ti alloys.

The issue with O as a principal α-phase stabilizer in Ti is its embrittling effect owing to its strong interactions with dislocations during deformation[6,7]. Furthermore, O changes the phase equilibria, promoting the formation of the embrittling $α_2$-phase (Ti₃Al)[14]. These constraints have led to the following empirical design rule for industrial Ti alloys: Al + 10(O + C + 2N) + 1/3Sn + 1/6Zr < 9.0% (ref. 5). For Ti–6Al–4V, this design rule requires less than 0.12% O (ref. 15) at 0.05% N and 0.08% C, which was relaxed to 0.13% O for Grade 23 Ti–6Al–4V and 0.20% O for Grade 5 Ti–6Al–4V. Following this rule, a lower Al content allows for a higher O content. Indeed, the latest industrial α–β Ti alloy ATI 425 (Ti–4.5Al–3V–1.8Fe–0.3O)[16], allows 0.3% O maximum because of its lower Al content, for which the above empirical rule accepts a maximum of 0.31% O. If no Al is included, this rule allows a maximum of 0.72% O.

More options exist for β-phase stabilizers in Ti (refs. 3–5), with Fe being the most effective and inexpensive one[17,18]. Furthermore, Fe is the second lightest β-phase stabilizer. However, its use has been constrained by the formation of Fe-stabilized β-flecks during ingot solidification[9] (up to centimetres in size; Supplementary Note 1), which can markedly affect the mechanical properties[9]. Therefore, the use of

[1]Centre for Additive Manufacturing, School of Engineering, RMIT University, Melbourne, Victoria, Australia. [2]School of Aerospace, Mechanical and Mechatronic Engineering, The University of Sydney, Sydney, New South Wales, Australia. [3]Australian Centre for Microscopy & Microanalysis, The University of Sydney, Sydney, New South Wales, Australia. [4]Research Institute for Advanced Manufacturing, Department of Industrial and Systems Engineering, The Hong Kong Polytechnic University, Hong Kong, China. [5]State Key Laboratory of Ultra-precision Machining Technology, Department of Industrial and Systems Engineering, The Hong Kong Polytechnic University, Hong Kong, China. [6]Hexagon Manufacturing Intelligence, Doncaster, Victoria, Australia. [7]These authors contributed equally: Tingting Song, Zibin Chen. ✉e-mail: simon.ringer@sydney.edu.au; ma.qian@rmit.edu.au

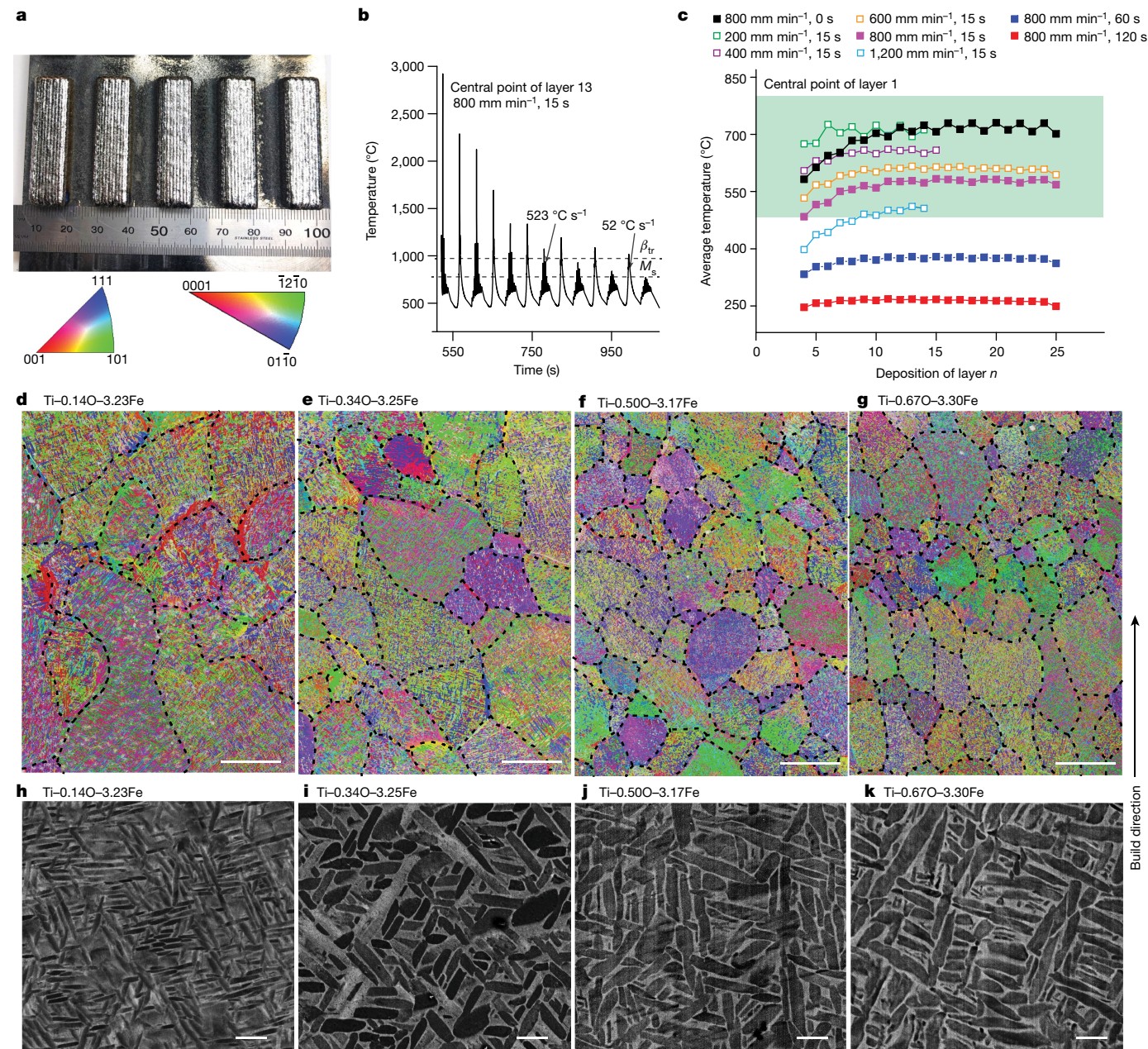

**Fig. 1 | Microstructure of DED-printed Ti–O–Fe alloys. a**, As-built rectangular coupons of $40 \times 10 \times 5\ mm^3$ at the layer thickness of 200 μm. The five coupons in **a** all have the same composition of Ti–0.34O–3.25Fe. **b**, Temperature profile of the central point of layer 13 in a 25-layer coupon by simulation. **c**, Processing window (green zone) determined by simulation. **d–k**, Electron backscatter diffraction (EBSD) inverse pole figure images (**d–g**; scale bars, 100 μm) and backscattered electron images (**h–k**; scale bars, 1 μm) of the printed Ti–0.14O–3.23Fe (**d,h**), Ti–0.34O–3.25Fe (**e,i**), Ti–0.50O–3.17Fe (**f,j**) and Ti–0.67O–3.30Fe (**g,k**) alloys. The light-grey phase in **h–k** is the β phase and this contrast arises owing to Fe enrichment.

Fe is usually limited to about 2% in industrial Ti alloys such as ATI 425 and Ti–10V–2Fe–3Al (ref. 2).

Nonetheless, the aforementioned advantages of O and Fe (≤2%) once attracted substantial efforts to develop α–β Ti–O–Fe alloys as alternatives to Ti–6Al–4V (refs. 19–22). The desire for improved hot workability and surface finish during ingot-breakdown hot-working operations was the particular motive at that time[19]. These efforts yielded mixed success, with the two well-investigated compositions being Ti–0.35O–1Fe–0.01N and Ti–0.3O–1Fe–0.04N (refs. 19,20,22). Both alloys exhibited tensile properties comparable with Ti–6Al–4V in hot-worked and annealed conditions, but lower tensile strength (600–700 MPa) and ductility (2–3%) in as-cast conditions[20].

We introduce another consideration of the combined use of O and Fe in Ti, which is related to revitalizing off-grade sponge Ti owing to excess O and Fe contamination[10,11,23]. Sponge Ti production is highly energy-intensive[12]. Given that off-grade sponge Ti (Ti–O–Fe) accounts for 5–10% of all sponge Ti production[10,11,23], their use as input feedstock for powder production for AM has the potential to add notable value and reduce the carbon footprint of the Ti industry.

We have sought to circumvent the metallurgical challenges arising from alloying Ti with O and Fe by integrating alloy design concepts with AM process design. Our goal was to create a new class of ductile and strong α–β Ti–O–Fe alloys through AM. By its molybdenum equivalence, an addition of 3.5% Fe to Ti can retain the prior-β phase to room

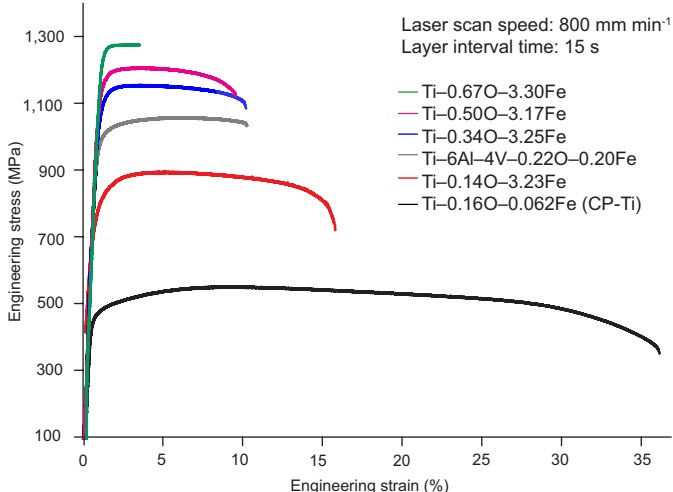

Laser scan speed: 800 mm min⁻¹
Layer interval time: 15 s

Ti–0.67O–3.30Fe
Ti–0.50O–3.17Fe
Ti–0.34O–3.25Fe
Ti–6Al–4V–0.22O–0.20Fe
Ti–0.14O–3.23Fe
Ti–0.16O–0.062Fe (CP-Ti)

**Fig. 2 | Tensile properties of DED-printed Ti–O–Fe alloys at room temperature by focusing on varying alloy composition without changing the processing conditions.** Engineering stress–strain curves of the Ti–(0.14–0.67)O–(3.17–3.30)Fe, ultralow-iron Ti–0.16O–0.062Fe and Ti–6Al–4V alloys, printed using the same DED conditions (laser scan speed: 800 mm min$^{-1}$, layer interval: 15 s; see Extended Data Table 2 for other conditions). Extended Data Table 1 lists the tensile properties of each alloy.

temperature by water quenching[17,18]. This sets an upper limit for Fe. The maximum O content was set at 0.7% based on the empirical rule discussed earlier (0.72% O). Furthermore, most Ti alloys lose tensile ductility above 0.7% O (based on Fig. 1 on page 6 of ref. 1). Four levels of O (0.15%, 0.35%, 0.50%, 0.70%) were thus considered with 3% Fe, leading to ten experimental alloys (Extended Data Table 1). In terms of AM process selection, we chose laser metal powder directed energy deposition (DED), which—aided by high-fidelity simulations—allows for the fabrication of large-scale near-net-shape components with a consistent microstructure.

We first simulated DED of rectangular coupons (Fig. 1a), using parameters from Extended Data Table 2. The simulations predicted high cooling rates after solidification (Fig. 1b), which are expected to lead to metastable phases such as α′-martensite. However, the large number of thermal pulses (Fig. 1b), high stabilization temperatures (Fig. 1c) and approximately isothermal durations should ensure that only α and β phases are present. For this purpose, we chose 480–800 °C as the required thermal history bounds (Methods), delimited as the green zone in Fig. 1c. Rectangular coupons of the designed compositions (Extended Data Table 1) were then printed within and outside this window, together with two reference alloys, Ti–6Al–4V–0.22O–0.20Fe and Ti–0.16O–0.062Fe. The printed coupons exhibited consistent compositions (Supplementary Note 2).

The microstructure of the low-oxygen Ti–0.14O–3.23Fe alloy comprises short columnar and equiaxed prior-β grains (Fig. 1d), akin to the laser additively manufactured Ti–6Al–4V–3Fe alloy[24]. Fine equiaxed prior-β grains formed with increasing O content (Fig. 1e–g and Supplementary Fig. 1) compared with long columnar prior-β grains in Ti–6Al–4V printed under similar[25] or different conditions[26–28]. At room temperature, fine α–β lamellae were the prevailing microstructure in each alloy (Fig. 1h–k). The α-lath thickness increased from 180 ± 33 nm at 0.14% O to 375 ± 76 nm at 0.67% O, whereas the β-phase volume fraction simultaneously increased from 21 ± 2.3% to 31 ± 0.5% (Supplementary Fig. 2). No Fe-stabilized β-flecks were observed in any of these printed alloys, but they were prevalent in the copper-mould-cast Ti–0.35O–3Fe alloy (Extended Data Fig. 1). The reasons are discussed in Supplementary Note 1. The avoidance of β-flecks is an important advantage of AM in the fabrication of these alloys (Supplementary Note 3).

Figure 2 shows representative tensile properties of four designed Ti–O–Fe alloys and two reference alloys. Our goal here was to change alloy composition without changing the AM processing conditions, within the required thermal history bounds (Fig. 1c). The complete engineering stress–strain curves are provided in Extended Data Fig. 2a. Separate coupons were printed by fixing the alloy composition to Ti–0.35O–3Fe and varying the AM processing condition. The tensile stress–strain curves of these alloys and the copper-mould-cast Ti–0.35O–3Fe alloy are shown in Extended Data Fig. 2b,c, whereas Extended Data Fig. 3 shows their microstructures. Without optimization, our Ti–(0.34–0.50)O–(3.17–3.32)Fe alloys printed within the processing window exhibited tensile ductility ($\varepsilon_f$) from 9.0 ± 0.5% to 21.9 ± 2.2% (the change in $\varepsilon_f$ is not because of porosity; Supplementary Note 2) and ultimate tensile strength ($\sigma_{UTS}$) from 1,034 ± 9 to 1,194 ± 8 MPa (Extended Data Table 1). The as-cast Ti–0.35O–3Fe alloy demonstrated a more than 50% lower $\varepsilon_f$ at similar $\sigma_{UTS}$.

The strengthening potency of the Fe reached 105 MPa/1.0 wt% Fe by comparing the $\sigma_{UTS}$ values of the Ti–0.14O–3.23Fe and Ti–0.16O–0.062Fe alloys, close to the reported experimental value of 75 MPa/1.0 wt% Fe (based on the data given in Table 12 on page 29 of ref. 1). The strengthening potency of the O registered 76 MPa/0.1 wt% O by comparing the $\sigma_{UTS}$ values of the Ti–0.14O–3.23Fe and Ti–0.67O–3.30Fe alloys, which is within the reported typical experimental range of >70 MPa/0.1 wt% O (refs. 29,30). Both Fe and O played an important role in strengthening these alloys.

To explain the strengthening mechanisms, we investigated the atomic distribution of the O and Fe in three alloys, Ti–0.14O–3.23Fe, Ti–0.34O–3.25Fe and Ti–6Al–4V–0.22O–0.20Fe. Combined with integrated differential phase contrast (iDPC), scanning transmission electron microscopy (STEM) enables direct observations of interstitial light elements[31]. We observed strong segregation of O atoms to the α-lath rims in the Ti–0.34O–3.25Fe alloy (Fig. 3a). They existed in the HCP interstitial sites near the α/β interface, forming a unique nano-heterogeneity from the interior of the α-lath (low O, ductile) to the interface regions (high O, strong). By contrast, this nano-heteromicrostructure was rarely observed in the low-oxygen Ti–0.14O–3.23Fe alloy (Fig. 3b). Atom probe tomography (APT) analysis confirmed the same observations (Fig. 3c and Extended Data Fig. 4a).

The local charge density in the Ti–0.34O–3.25Fe alloy was analysed using differential phase contrast (DPC)STEM (Fig. 3d). Here the vectors represent the local electric-field directions, whereas the colours denote the relative strength of the local electric field (green = weak, yellow = strong). An increase in the local charge density (yellow) was observed in the α-phase rims in which O interstitials had segregated (Fig. 3d). This is indicative of a further bonding contribution in the lattice from the O interstitials, which we suggest contributes to strengthening by further impeding dislocation motion. Figure 3e shows a collection of O-interstitials at a dislocation in an α-phase rim, similar to Cottrell atmospheres in BCC crystals[32], recorded using combined iDPC and high-angle annular dark-field (HAADF)-STEM techniques. The dislocation is effectively pinned by such O atmospheres.

Our APT data further showed that the β-phase was virtually free of O (0.03 at%). This was supported by the CALPHAD (CALculation of PHAse Diagrams) predictions performed at 650 °C in the green zone of Fig. 1c (Supplementary Table 2). By comparison, the β-phase in the printed Ti–6Al–4V–0.22O–0.20Fe alloy contained substantial O (0.27 at%) (Extended Data Fig. 4b,c), consistent with the literature (Supplementary Table 4). Our APT data also showed that the α-phase was virtually free of Fe (0.02 at%; Fig. 3c), whereas the substantial Fe in the β-phase was non-uniformly distributed (Fig. 3f), which adds a strengthening capacity to the alloy by generating local strains for impeding dislocation motion.

Density functional theory (DFT) simulations (Fig. 4a) predict that the O atoms prefer to reside in the α-phase, especially close to the α/β interface (Fig. 4b), but the Fe atoms show no tendency to segregate to

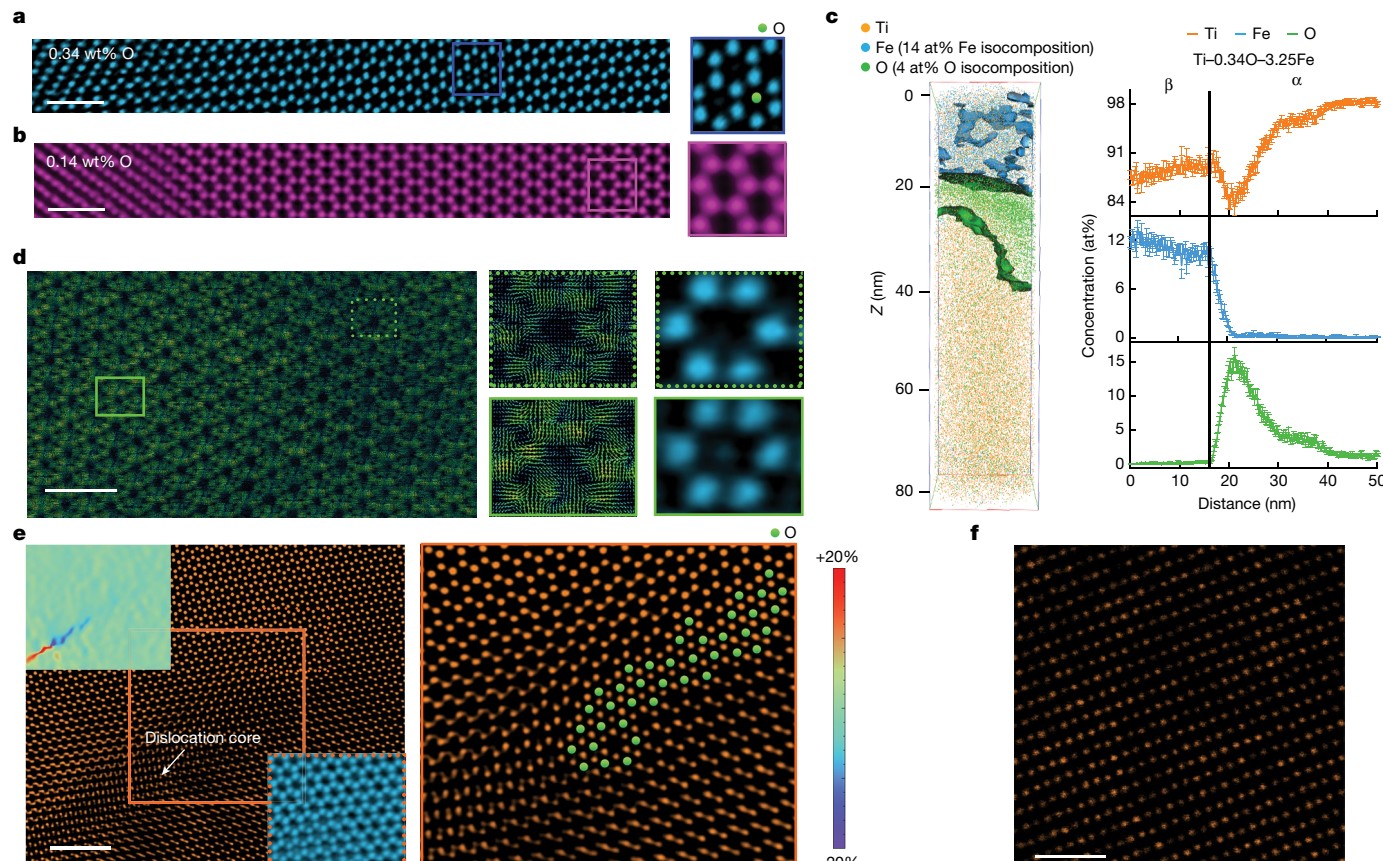

**Fig. 3 | Distribution of O and Fe atoms in DED-printed α–β Ti–O–Fe alloys. a,b**, iDPC-STEM images of the α/β interfaces in Ti–0.34O–3.25Fe and Ti–0.14O–3.23Fe. Oxygen atoms are observed at the interstitial positions in the α-phase near the α/β interface in Ti–0.34O–3.25Fe, whereas few such O atoms are detected in the Ti–0.14O–3.23Fe alloy. Scale bars, 1 nm. **c**, APT data from Ti–0.34O–3.25Fe, highlighting the tendency of O atoms to segregate towards the edges of the α-phase near the α/β interfaces. **d**, Tensor-flow DPC-STEM images along a [0001]α direction. The tensor direction represents the local electric field direction. The tensor colours represent the strength of the local electric field (green = weaker, yellow = stronger). Scale bar, 1 nm. **e**, HAADF-STEM image of a dislocation inhibited by an O interstitial array. Geographic phase analysis (top-left corner inset) shows the strain condition of the defect and the surrounding region. The dislocation core is defined by the red–blue intersection in the inset. The iDPC-STEM image (bottom-right corner inset) shows the O interstitial array. The right-hand enlarged HAADF-STEM image shows the strong presence of O interstitials (extracted from the iDPC image and marked as green balls) around the dislocation, impeding dislocation movement. Scale bar, 2 nm. **f**, An HAADF-STEM image of a β-phase region, highlighting the non-uniform distribution of Fe, shown by the uneven $Z$-contrast, for which the zone axis is [110]β. Bright contrast means more Fe atoms in the local β-Ti lattice. Scale bar, 1 nm.

the interface (Fig. 4c). These predictions fully support the observations in Fig. 3. Furthermore, the 4th and 3rd nearest-neighbour separation configurations were most favourable for Fe–Fe pairs in the β-phase (Fig. 4d), whereas the 5th nearest neighbour ($d$ = 0.506 nm) separation configuration was most favourable for O–O pairs in the α-phase (Fig. 4e). On this basis, we estimated the [Fe] and [O] content in each phase, which gives 15 at% Fe in β and 18 at% O in α. These estimates match our APT measurements well (Fig. 3c).

We also sought a direct comparison of the crystallographic arrangements of the α-variants in our printed Ti–(0.14–0.67)O–(3.17–3.30)Fe and Ti–6Al–4V alloys. The prevalent misorientation between neighbouring α-variants in each Ti–O–Fe alloy was [11$\bar{2}$0]/60°, compared with [1$\bar{0}$55$\bar{3}$]/63.26° in Ti–6Al–4V (Extended Data Fig. 5). The former is close to {10$\bar{1}$1} twinning[33], possessing lower energy than the latter, and is therefore an energetically preferred configuration[34,35].

Dislocation activity in the vicinity of tensile crack tips determines whether materials exhibit brittle or ductile behaviour when fracture occurs[36]. Active dislocation multiplication can relieve stress concentration and blunt crack tips, enabling further plastic deformation[37]. Our α–β Ti–(0.35–0.70)O–3Fe alloys comprise roughly 30 vol% β-phase versus about 5 vol% β-phase in Ti–6Al–4V. We suggest that this large volume fraction of the virtually oxygen-free β-phase (Fig. 3c) and the

oxygen nano-heterogeneity inside the α-phase have combined to play a vital role in mediating the overall deformation process, resulting in excellent tensile properties. Therefore, we examined areas abutting the tensile fracture surfaces of the Ti–0.34O–3.25Fe ($\varepsilon_f$ = 9.0%) and Ti–0.67O–3.30Fe ($\varepsilon_f$ = 3.0%) alloys using TEM.

Dislocation tangles developed at the centre of the α-phase laths in the Ti–0.34O–3.25Fe alloy, whereas extensive dislocation multiplication was apparent in the β-phase (Extended Data Fig. 6). This provides direct evidence to support our above hypothesis. In the Ti–0.67O–3.30Fe alloy, dislocation multiplication was observed, but to a lesser extent (Extended Data Fig. 7), corresponding to the limited overall plastic deformation ($\varepsilon_f$ = 3.0%). The tendency towards fracture increases with increasing applied tensile stress ($\sigma$) until the local stress intensity factor $K$ (proportional to $\sigma$) exceeds the critical value for the α-phase or the prior-β grain boundaries or the interfacial regions between the high-oxygen α-lath rim and β-phase. A local fracture would then proceed through one or all of them, for example, when $\sigma$ reached 1,271 ± 6 MPa for the Ti–0.67O–3.30Fe alloy, leading to low $\varepsilon_f$.

The above understanding corresponds well to the fracture features of each alloy (Extended Data Fig. 8a–f). For example, the fracture surface of the Ti–0.34O–3.25Fe alloy consisted of large, deep dimples (extensive dislocation multiplication) and a portion of facets with small,

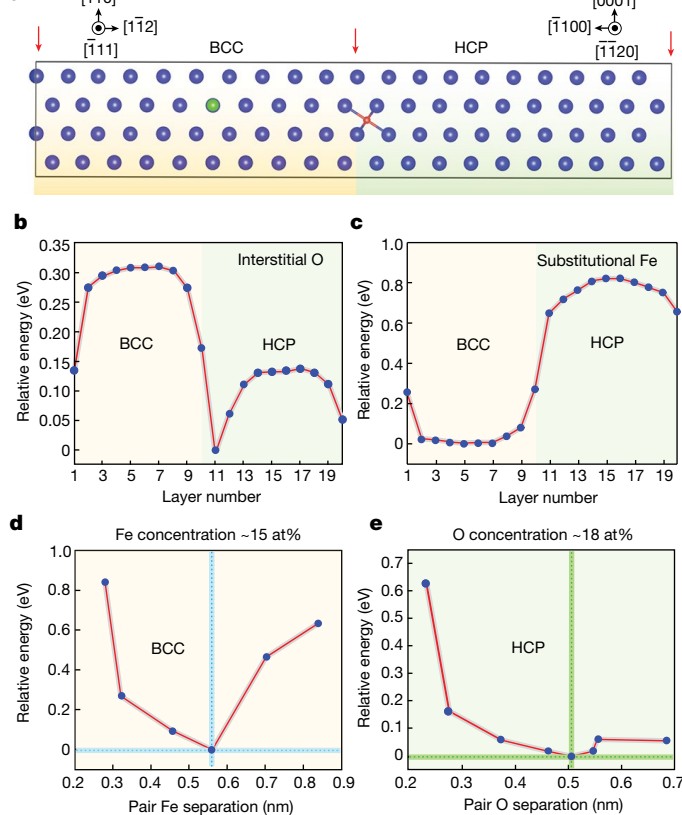

**Fig. 4 | DFT simulations of the distribution of Fe and O atoms in BCC (β) and HCP (α) phases of α–β Ti–O–Fe alloys. a,** A DFT BCC ($\overline{1}1\overline{2}$)/HCP ($\overline{1}100$) interface model containing one octahedral interstitial O atom (in red) and one substitutional Fe atom (in green) at their respective favourable positions. On the basis of Fig. 3, we constructed a 1 × 1 in-plane ten-layer BCC ($\overline{1}1\overline{2}$)/1 × 2 in-plane ten-layer HCP ($\overline{1}100$) interface structure (ten layers on each side) to simulate an α/β interface. The red arrows indicate the equivalent BCC/HCP interfaces owing to the periodic boundary conditions. **b,c,** The calculated layer-resolved relative energy mapping for an interstitial O atom (**b**) and a substitutional Fe atom (**c**) throughout the interfaces. The relative energy as a function of pair separation for O and Fe in each phase was also calculated. **d,e,** The calculated relative total energy as a function of pair separation for pair interstitial O atoms in a 96-atom HCP-Ti supercell (**d**) and pair substitutional Fe atoms in a 54-atom BCC-Ti supercell (**e**). The calculated equilibrium O and Fe concentrations were based on the respective energetically most favourable configurations (see text).

shallow dimples (less ductile). Conversely, the fracture surface of the Ti–0.67O–3.30Fe alloy exhibited notable facets with a size comparable with the prior-β grain size, indicative of inter-(prior-β)-granular fracture, and only a fraction of large, deep dimples (localized dislocation multiplication). The fracture characteristics of the other alloys could be understood similarly.

The relative phase fractions and grain sizes also play an important role in the strengthening mechanisms. The increase in the β-phase volume fraction with increasing O stems from the strong partitioning of O and Fe in each phase. For instance, at 800 °C (in the green zone of Fig. 1c), CALPHAD predicts substantially more β-phase in the Ti–0.14O–3Fe alloy (53.5 vol%) than in the Ti–0.67O–3Fe alloy (39.5 vol%) (Supplementary Note 5). The lower the β-phase fraction, the higher the average Fe content in the β-phase and the more stable the β-phase will be. This leads to more retained β-phase. In this particular case, the β-phase in the Ti–0.67O–3Fe alloy contained 7.30% Fe versus 4.85% Fe in the β-phase of the Ti–0.14O–3Fe alloy. In terms of the increase in the α-lath thickness with increasing O, we ascribe this to

the higher formation temperature of the α-phase, which accelerates diffusion while reducing strain energy from the formation of the α-phase.

To summarize, we have demonstrated an integration between alloy design and simulation-based AM process design to create a new class of strong and ductile α–β Ti–(0.35–0.50)O–3Fe alloys ($\varepsilon_f$ = 9.0 ± 0.5% to 21.9 ± 2.2%; $\sigma_{UTS}$ = 1,034 ± 9 to 1,194 ± 8 MPa), available across a generous AM processing window, using the readily abundant elements O and Fe. We attribute the success of these alloys to a combination of multiscale microstructural features arising from this integration. These include: (1) the fine α–β lamellae distributed within the fine equiaxed prior-β grains; (2) the high potency of O and Fe in strengthening the α-phase (virtually Fe-free) and β-phase (about 30 vol%, virtually O-free), respectively; and (3) the preferred misorientation configuration between the neighbouring α-variants in these alloys. Among these factors, the unique partitioning of O and Fe is fundamental, and of particular criticality is our report of a nano-heteropartitioning in the α-phase leading to high-oxygen (strong) and low-oxygen (ductile) distributions that affect the local nature of the atomic bonding, shown by the DPC and iDPC.

These strong and ductile Ti–O–Fe alloys are expected to have implications across a wide range of potential applications at room temperature (Supplementary Note 6). Furthermore, sponge zirconium (Zr) is produced in the same way as sponge Ti. Therefore, the same could be expected for the use of off-grade sponge Zr to develop strong and ductile Zr–O–Fe alloys. In addition, this work provides a potential pathway for future interstitial engineering by AM, such as mitigating nitrogen (N) embrittlement in Ti and Zr, and oxygen embrittlement in other metals (Supplementary Note 6).

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

## Methods

### Materials for alloy fabrication

The feedstock powders used for alloy fabrication through laser DED include argon-gas-atomized commercially pure Ti (CP-Ti) powder (50–100 μm, TLS Technik GmbH & Co.), water-atomized Fe powder (20–50 μm, Höganäs) and $TiO_2$ powder (<5 μm, Sigma-Aldrich). The composition of the CP-Ti powder is Ti–0.14O–0.139Fe–0.01N–0.011C–0.0011H and that of the Fe powder is Fe–0.003C–0.09O–0.01S (in wt%). The powders were blended in a Turbula mixer (model T2F) as per each designed alloy composition for 2 h at ambient temperature in a sealed plastic container. For comparison purposes, powders of an ultralow-iron CP-Ti composition (Ti–0.16O–0.062Fe) and a Ti–6Al–4V–0.22O–0.20Fe alloy were also used to build reference samples. Both powders were supplied by TLS Technik GmbH & Co. with a particle size range of 50–100 μm.

Furthermore, CP-Ti (Ti–0.13O–0.15Fe) and pure Fe nuggets (99.99%, ZhongNuo Advanced Material Technology Co.) and $TiO_2$ powder (<5 μm, Aladdin) were used for alloy fabrication by means of vacuum arc melting and casting.

### Thermal history simulation

The DED module in Simufact Welding was used to track temperature evolution in the build[25,38]. The predictability of Simufact Welding (DED) in terms of melt pool shape and size and thermal cooling was evaluated in Supplementary Note 2. We used the temperature-dependent thermophysical data for Ti–6Al–4V owing to the lack of similar data for Ti–O–Fe alloys. The simulation focuses on the deposition of a 25-layer rectangular coupon ($40 \times 10 \times 5$ mm$^3$) on a 10-mm-thick Ti–6Al–4V substrate.

The DED module adopts a finite element method without considering melt pool convections but focusing on heat conduction[38]. This is a common simplification in weld modelling because both the filler metal and the interface scale are negligible compared with the base metal dimensions[39]. In the case of laser metal powder DED, the melt pool size is substantially smaller than the substrate or part size so that most of the heat is conducted away from the melt pool towards the substrate.

To use the DED module, the deposit geometry is first introduced as a defined precursor, the meshes of which are set to the quiet mode (inactive) at the outset. Once the simulation starts, each mesh is activated (element-wise) by means of the moving volumetric heat source entity. The elements that are touched by the heat source during its propagation will be permanently activated using an element-birth algorithm. As a result, all the activated elements are exposed to the heat source, which represents a 3D volumetric heat flux that has the dimensions of a typical melt pool. Being a nodal load, the heat flux from this heat source will increase the temperature of the just-activated elements to values above their melting or liquidus temperature.

Extended Data Table 2 lists the DED parameters and illustrates the bidirectional scan strategy used for simulation, including an equivalent laser spot size of 1.5 mm and a laser power of 500 W. The selection of DED parameters is discussed in the next section. The scanning path files (G-code) with the DED parameters are imported into the DED module to build the part geometry. Each layer ($40 \times 10$ mm) consists of 6,400 points with a resolution of $0.25 \times 0.25$ mm. The time resolution is set to 0.02 s to capture heating and cooling information. The simulation focuses on the influence of the layer-to-layer interval (0 s, 15 s, 60 s and 120 s).

The multipeaks or thermal pulses in Fig. 1b and Supplementary Fig. 3 arise from the bidirectional scan strategy used. Each odd-number layer has nine scan paths parallel to the length direction of the sample across its width (10 mm). As a result, nine peaks or nine thermal pulses corresponding to these nine scan paths will appear in previously solidified layers. By contrast, each even-number layer has 38 short scan paths perpendicular to the length direction of the sample. However, each of these short scan paths is not thermally strong enough to result in a detectable thermal pulse in the previously solidified layers. Collectively, they only lead to a limited number of small thermal pulses. Supplementary Fig. 3 shows the correspondence in detail.

### Alloy fabrication by DED

Four groups of α–β Ti–O–Fe alloys were designed: Ti–0.15O–3Fe, Ti–0.35O–3Fe, Ti–0.50O–3Fe and Ti–0.70O–3Fe. Extended Data Table 1 lists the measured compositions of each fabricated alloy in the range of Ti–(0.14–0.67)O–(3.11–3.36)Fe. They were all fabricated using a TRUMPF TruLaser Cell 7020 system (thin-disk laser, 1,030 nm). A 12-mm-thick CP-Ti substrate plate was used for deposition. Extended Data Table 2 lists the experimental deposition parameters. As-built coupons (length: 40 mm, width: 10 mm, thickness: 5 mm) are shown in Fig. 1a and Supplementary Fig. 5a. Five coupons of each composition were printed.

**Selection of DED parameters and processing window.** The parameters selected should ensure fast and consistent high-quality fusion (virtually free of lack-of-fusion defects and keyhole pores) and result in fine and short α–β lamellae through decomposition of the α′-martensite phase after solidification. The laser energy density $E_d$ (J mm$^{-2}$) for obtaining high-quality DED of Ti–6Al–4V on a Ti–6Al–4V alloy substrate has been systematically studied in the range 16.6–36 J mm$^{-2}$ with a large laser spot size (2 mm)[40], in which $E_d = \frac{P}{dv}$ (P laser power, d laser spot site, $v$ laser scan speed). The use of high values of $E_d$ was not recommended[40]. This is consistent with our experiences of using the TRUMPF system for laser metal powder DED of Ti–6Al–4V on Ti–6Al–4V substrates with a similar laser spot size (optimum range of $E_d$ = 25–35 J mm$^{-2}$; laser spot size: 1.5 mm).

The spot size was chosen to be 1.5 mm to ensure a sufficiently large melt pool (≥1.0 mm$^3$), which is important for (1) enhancing chemical homogeneity when mixed powders are used, (2) layer height uniformity and (3) achieving a more stable thermal profile in the build. A wide range of scan speed (200–2,000 mm min$^{-1}$) is available. However, with $E_d$ = 25–35 J mm$^{-2}$, our experience with the TRUMPF system is that, for high-quality fusion with the previously solidified layers, the laser spot dwell time $d/v$ should be 0.1–0.15 s. This leads to the selection of $v$ = 600–800 mm min$^{-1}$ and, therefore, the laser power of 500 W for the expected $E_d$.

The layer thickness of 200 μm and the powder flow rate of 1.7 g min$^{-1}$ were determined together through various previous experimental studies to obtain (1) a consistent weld bead with an aspect ratio of 3–4 to 1, (2) a high-quality build that is essentially free of lack-of-fusion defects (through fracture surface characterization, assisted with micro-computed tomography analysis) and (3) a consistent microstructure.

With the above-selected parameters, the DED module in Simufact Welding predicted a semiellipsoidal melt pool with a volume of 1.08 mm$^3$ (major axis: 2.3 mm; minor axis: 1.5 mm; depth: 0.6 mm; see Supplementary Note 2), in which the liquidus temperature of our designed Ti–0.35O–3Fe alloy is about 1,659 °C.

The scan spacing was selected as 1.05 mm based on the simulated melt pool width of 1.5 mm to ensure a linear overlap of 70% for sufficient fusion between two abutting tracks, a standard practice to achieve layer and height uniformity during the build (an overlap of 30% minimum has been commonly used).

The layer-to-layer interval time (τ) can be varied over a wide range. A practical range is from 0 s to no longer than 120 s. Changing τ can substantially change the cooling rate and the entire thermal history of the build, thereby altering the microstructure. The default layer interval used in this work was 15 s but we have investigated the whole range of 0–120 s.

These preliminary selected parameters were then subjected to systematic simulations using Simufact Welding to meet the following

two criteria to obtain α–β lamellae through decomposition of the α′-martensite: (1) an appropriate cooling rate (≥ 400 °C s⁻¹) when the β-phase in each solidified layer is cooled from the single β-phase region for the formation of α′ and (2) an appropriate stabilized temperature window across each layer (≥ 480 °C, preferably 600–750 °C) (excluding the last few layers) to allow α′ → α–β.

If the selected parameters fail to satisfy these two criteria, reselection by simulation is necessary until they do. Supplementary Table 3 summarizes the predicted average cooling rate at three selected points in the build before reaching 800 °C (the assumed martensite start temperature, $M_s$), with the scan speed ranging from 200 to 1,200 mm min⁻¹ and the layer interval ranging from 0 to 120 s. They all satisfy the required cooling condition.

Figure 1c shows the evolution of the average temperature at the centre of layer 1 from all 24 subsequent layers of deposition. The average temperature at the central point was calculated over the scanning time for each subsequent layer (for example, 35.85 s at 600 mm min⁻¹ for each odd-number layer) plus the layer interval time (for example, 15 s). Stabilization occurs from the 8th or 9th layer of deposition.

Decomposition of α′ starts to occur at 400 °C (long holding is needed)[41] but complete decomposition of α′ requires isothermal exposure to 800 °C for 2 h or longer[41,42]. However, the effect of thermal pulses (Fig. 1b and Supplementary Fig. 3) introduced during DED can markedly accelerate the decomposition process[25].

On the other hand, the microstructure could risk severe coarsening at ≥800 °C, whereas below 480 °C (the minimum stress relief temperature), decomposition of α′ may not be complete. Furthermore, the stabilized temperature increases with increasing build height (Supplementary Fig. 4), which means that the stabilized temperature of the first layer should not be set too high. On the basis of these considerations, we choose 480–800 °C in Fig. 1c (the green zone) as the required thermal history bounds or processing window. The stabilized temperature increases with reducing layer interval time or scan speed or with increasing build height.

The scan speed (600–800 mm min⁻¹) and layer interval time (0–120 s) were then varied both within and outside the simulation-informed processing window to investigate the resulting microstructures and tensile properties of our designed Ti–(0.15–0.70)O–3Fe alloys.

**Chemical homogeneity of as-fabricated samples.** Each melt pool produced in the deposition process of this work (around 1 mm³) requires melting of about 6,250 Ti powder particles used in this study ($D_v50$ = 69.09 μm). This helps to mitigate the chemical inhomogeneity issue arising from the use of mixed powders. Also, because the melt pool depth (0.6 mm) is three times the thickness of each solidified layer (0.2 mm), and there is a 70% overlap between each two abutting tracks as well, substantial remelting occurs to each solidified melt pool. This further facilitates chemical homogeneity. Both compositional analysis and microstructural examination of different batches of as-fabricated coupons confirmed good overall chemical homogeneity (see Supplementary Note 2).

**Alloy fabrication by arc melting and water-cooled copper-mould casting**

As-cast coupons of the Ti–0.33O–3.11Fe alloy were prepared using a vacuum arc melter equipped with a suction casting device (Physcience Opto-electronics Co., Ltd., model WK-II type). The water-cooled copper (Cu) hearth in the melting chamber has five hemispherical cavities (cleaned before use). The $TiO_2$ powder was placed at the bottom of one cavity, followed by the properly cleaned CP-Ti and Fe nuggets to avoid powder splashing during arc melting. Furthermore, separate CP-Ti nuggets were placed in another cavity as oxygen scavenger. The furnace was first evacuated to $5 \times 10^{-3}$ Pa and then purged with argon to 0.5 Pa. The vacuum and purge processes were repeated three times to ensure a low oxygen level chamber.

The scavenging CP-Ti nuggets were first melted three times for oxygen scavenging. Then the alloy charge (45–55 g) was melted and remelted eight times in total. The alloy button ingot was flipped over after each melting for chemical homogenization. Then it was moved to the central cavity of the water-cooled Cu hearth using a small shovel in the chamber. The alloy was remelted again and kept in the molten state for 30 s. The suction casting device (a water-cooled Cu mould) was connected to the melting chamber. After remelting, an air pressure difference was introduced between the upper and lower parts of the facility by opening the air evaporation valve connected to the Cu mould cavity. The molten alloy was then air-sucked and solidified in a water-cooled Cu mould cavity (length: 120 mm, width: 12 mm, thickness: 5 mm). Five coupons were fabricated following the same process.

## Microstructural characterization

EBSD was conducted on a JEOL JSM-7200F scanning electron microscope (SEM) system with an Oxford Instruments AZtecHKL imaging system, operated at 20 kV. Each specimen was tilted at 70° and the step size used was 500 nm. The EBSD data were analysed using Channel 5 software. Microstructure and fracture surface analyses were performed using an SEM (FEI Verios 460L) in either the secondary electron or backscattered electron mode. We conducted quantitative image analysis using the Image-Pro Plus package for the measurement of the width of α-laths and areas of the α and β phases for the volume fraction of β. STEM observations were carried out using a Themis Z double-corrected microscope operated at 300 kV equipped with an FEI DF4 detector for DPC and iDPC imaging.

The APT characterization was performed using a laser-assisted CAMECA local electrode atom probe 4000X Si. The needle-shaped specimens were prepared by the lift-out method and annual-milled in a traditional Ga FIB-SEM (Zeiss Auriga) and a plasma FIB-SEM (Helios G4). The specimens were analysed in laser mode with a specimen temperature of 50 K, a laser energy of 50 pJ and a pulse frequency of 200 kHz. Visualization and Analysis Software version 3.8.4 was used for the 3D reconstruction and data analyses. The default values of detector efficiency (0.57), ICF (1.65) and kf (3.30) were used. The error bars shown in 1D concentration profiles were calculated as $E = (C_i(1 - C_i)/N)^{1/2}$, in which $C_i = N_i/N$, $N_i$ represents the number of $i$ solute ions/atoms and $N$ represents the total number of counts with the given bin. The bin size is 0.2 nm.

The chemical compositions (Extended Data Table 1) of the as-built Ti samples were determined by inductively coupled plasma atomic emission spectroscopy and glow discharge mass spectrometry.

## Measurements of tensile mechanical properties

**Extraction of tensile specimens and microstructural uniformity.** DED-fabricated (Fig. 1a) and as-cast rectangular coupons were machined into dog-bone-like tensile specimens with gauge dimensions of $12 \times 3 \times 2$ mm³ (as per Australian Standard AS 1391-2007). Each tensile specimen was extracted from the middle nine layers along the length direction of the rectangular coupon. Supplementary Fig. 5 shows an array of ten as-fabricated coupons and the schematic extraction of each tensile sample. This is to ensure that each batch of tensile specimens has consistent microstructures for high repeatability. As shown in Supplementary Fig. 4, in the selected processing window (the green zone), stabilization of the average temperature occurs from the 8th or 9th layer of deposition. Although the average temperature increases with increasing build height, the increase across every eight layers after layer 9 is limited to approximately 30 °C (negligible for microstructural evolution). Microstructural examination of different batches of tensile specimens along the build height confirmed consistent microstructures (see Supplementary Note 2).

**Tensile testing.** All tensile specimens were uniaxially tested at room temperature with an initial strain rate of $1 \times 10^{-3}$ s⁻¹ using an MTS

universal testing facility (MTS 810, 100 kN) equipped with a non-contact laser extensometer.

**Repeatability of tensile stress–strain curves.** Supplementary Note 7 has briefly analysed the repeatability issue of the tensile stress–strain curves produced in this work (Extended Data Fig. 2) in accordance with ASTM E8/E8M-21 ('Standard Test Methods for Tension Testing of Metallic Materials'). The tensile data produced in this work meet the expected repeatability requirements.

## The CALPHAD method

The Pandat software and PanTi2021 database developed by CompuTherm[43] were used to calculate the equilibrium volume fraction of the β-phase and the equilibrium partition of the O and Fe atoms between the α-phase and the β-phase. The use of the CALPHAD method in this work was restricted to the approximate isothermal holding stage, that is, at one of the predicted stabilized temperatures (such as 650 °C) in Fig. 1c. The predictions were used to help understand the microstructural evolution during this isothermal holding stage. It was not used to predict the far-from-equilibrium cooling process. The isothermal holding time at each predicted stabilized temperature is up to around 14 min in each build. It is calculated this way: the deposition time for each odd-number layer (for example, 35.85 s at 600 mm min$^{-1}$) + the deposition time for each even-number layer (for example, 37.71 s at 600 mm min$^{-1}$) + each layer-to-layer interval time (15 s). Therefore, it is for isothermal approximation, although the deposition process is a far-from-equilibrium process.

## DFT calculations

DFT calculations were performed using the generalized gradient approximation[44] and the projector augmented-wave method implemented in the VASP code[45]. The cutoff energy for plane-wave basis sets was 500 eV. Monkhorst-Pack k-points were meshed by $6 \times 6 \times 6$, $8 \times 8 \times 8$ and $6 \times 6 \times 1$ for the 96-atom (and 128-atom), 54-atom bulk models and the interface model, respectively. For all systems, atomic relaxation was allowed until the forces were less than 0.01 eV Å$^{-1}$.

To build an α/β (HCP/BCC) interface model, we first constructed two surface slab models. For this specific interface, the in-plane lattice constants of the BCC ($\bar{1}1\bar{2}$) surface slab are $a = 4.6000$ Å and $b = 5.7260$ Å, whereas the corresponding values for the HCP ($\bar{1}100$) surface slab are $a = 4.6250$ Å and $b = 2.9236$ Å. The $a$ values are very similar but the BCC $b$ value is approximately twice the HCP $b$ value. Therefore, we constructed a $1 \times 1$ in-plane ten-layer BCC ($\bar{1}1\bar{2}$)/$1 \times 2$ in-plane ten-layer HCP ($\bar{1}100$) interface structure (Fig. 4a, ten layers on each side) to simulate an α/β interface.

## Data availability

All core data generated and analysed for this study can be found in this article, its Supplementary Information file, which contains 20 supplementary figures and 8 supplementary tables, and the source data spreadsheet files provided (Fig. 1b,c, Fig. 4, Extended Data Fig. 5, Supplementary Fig. 4 and Supplementary Table 8). Further source data (>40 GB) leading to these core data are available from the corresponding authors without any restrictions.

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

**Acknowledgements** We gratefully acknowledge support from the Australian Research Council (DP180103205, DP220103407, DP200100940, DP200102666, DP190102243 and IC180100005), the Australia–US Multidisciplinary University Research Initiative programme supported by the Australian Government through the Department of Defence under the Next Generation Technologies Fund, the Research Committee of The Hong Kong Polytechnic University (PolyU) (Project code: CD4F and UAMT), PolyU Research and Innovation Office (Project code: BBR5 and BBX2), and the funding support for the State Key Laboratories in Hong Kong from the Innovation and Technology Commission of the Government of the Hong Kong Special Administrative Region, China. We thank the RMIT Advanced Manufacturing Precinct (AMP), RMIT Microscopy & Microanalysis Facility (especially M. Field) and Sydney Microscopy & Microanalysis at the University of Sydney, which is a node of Microscopy Australia, for their facilities. We thank other team members for their contributions, including S. Luo for discussions on adding oxygen to titanium, A. Jones for printing the samples for Supplementary Fig. 15 (assisted by Z. Wu) and Supplementary Fig. 19 (assisted by Q. Zhou), Q. Zhou for performing the tensile tests for Supplementary Fig. 19, the technical team at RMIT AMP for machining all samples of this work from January 2019 to February 2023 and R. Hu for the porosity analysis of the as-cast samples. T.S. thanks A. Jones for training her on the TRUMPF TruLaser Cell 7020 system. Our DFT calculations were supported by the National Computational Infrastructure (NCI), with expert facilitation by the Sydney Informatics Hub team at the University of Sydney. Both Microscopy Australia and the NCI are supported by the Australian Government's National Collaborative Research Infrastructure Scheme.

**Author contributions** M.Q. and T.S. designed the experimental alloys. T.D. and T.S. performed simulations. T.S., M.Q. and M.B. conceived the AM process. T.S. printed the samples and completed essential characterization, data analysis and compilation. Z.C., X.L. and S.P.R. led the atomic-scale characterization, data analyses, DFT calculations and proposed the strengthening mechanisms. Z.C. conducted electron microscopy experiments and analyses. X.C. carried out DFT calculations. H.C. conducted APT experiments. S.L. performed EBSD analyses. M.Q. and S.L. analysed fracture behaviours. H.W. assisted in electron microscopy experiments and EBSD analyses. B.Q. and K.C.C. manufactured ingot samples. M.Q., M.B., S.P.R., X.L. and Z.C. acquired funding. M.Q. and T.S. led the response to reviews. All authors critically reviewed the results and edited the manuscript drafted by the corresponding authors.

**Competing interests** The authors declare no competing interests.

**Additional information**
**Correspondence and requests for materials** should be addressed to Simon P. Ringer or Ma Qian.

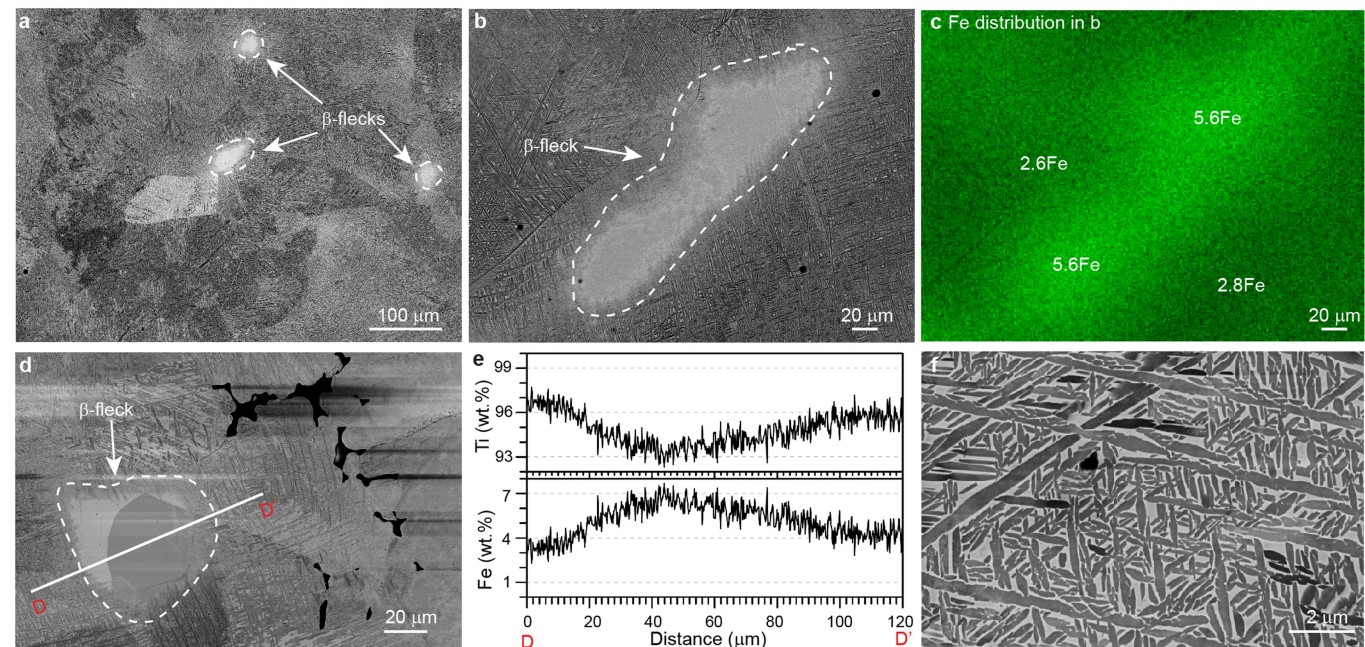

**Extended Data Fig. 1 | Microstructure of the as-cast reference alloy Ti–0.35O–3Fe (measured: Ti–0.33O–3.11Fe). a**, An overview of the microstructure comprising α–β lamellae and β-flecks. **b**, A closer view of a selected β-fleck. **c**, Energy-dispersive spectrometry (EDS) mapping of Fe in **b** to confirm the Fe-enriched β-fleck. **d**, A selected β-fleck phase for EDS line-scan analysis. Also shown is the as-cast porosity. **e**, EDS line scan further confirms the high Fe content in the β-fleck shown in **d**. **f**, A backscattered electron image showing the α–β lamellae in this as-cast alloy.

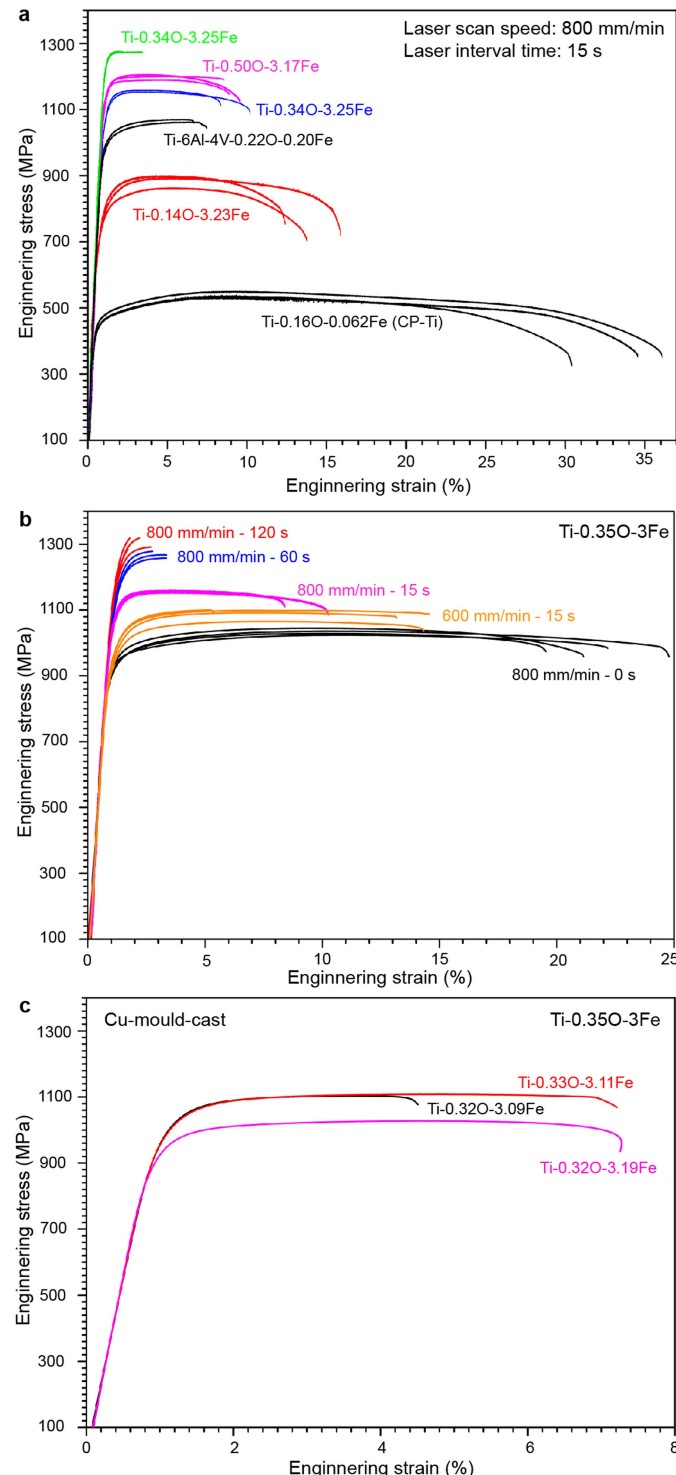

**Extended Data Fig. 2 | Engineering stress–strain curves for each DED-fabricated Ti–O–Fe alloy together with the reference Ti–6Al–4V alloy and ultralow-Fe Ti–0.16O–0.062Fe alloy. a**, Changing alloy composition but not the DED conditions (scan speed: 800 mm min⁻¹, layer interval: 15 s). **b**, Control studies to vary the microstructure through changing the DED conditions but not the alloy composition (Ti–0.35O–3Fe). **c**, Ti–0.35O–3Fe (by design) fabricated by water-cooled copper-mould casting as part of the control studies.

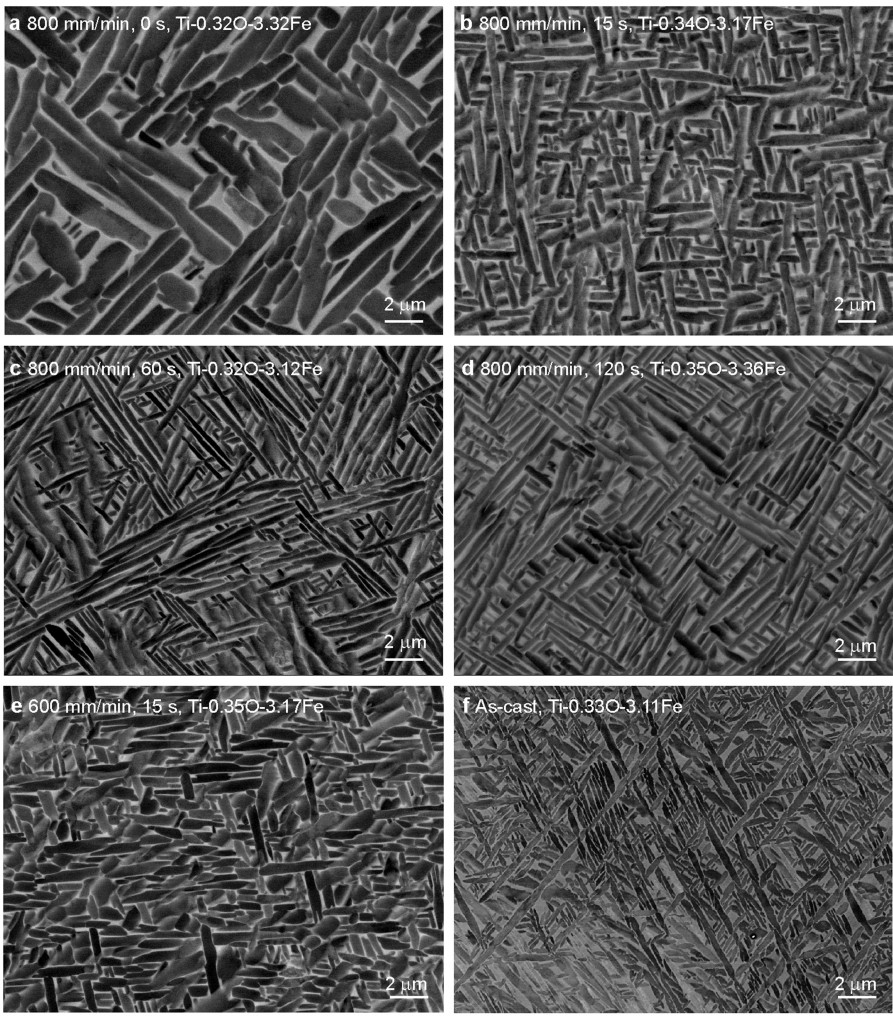

**Extended Data Fig. 3 | Control studies to vary the microstructure of the Ti–0.35O–3Fe alloy by varying the manufacturing conditions as specified in Extended Data Table 2. a–e** By DED with different scan speeds and layer interval times. **f**, By water-cooled copper-mould casting (see Methods).

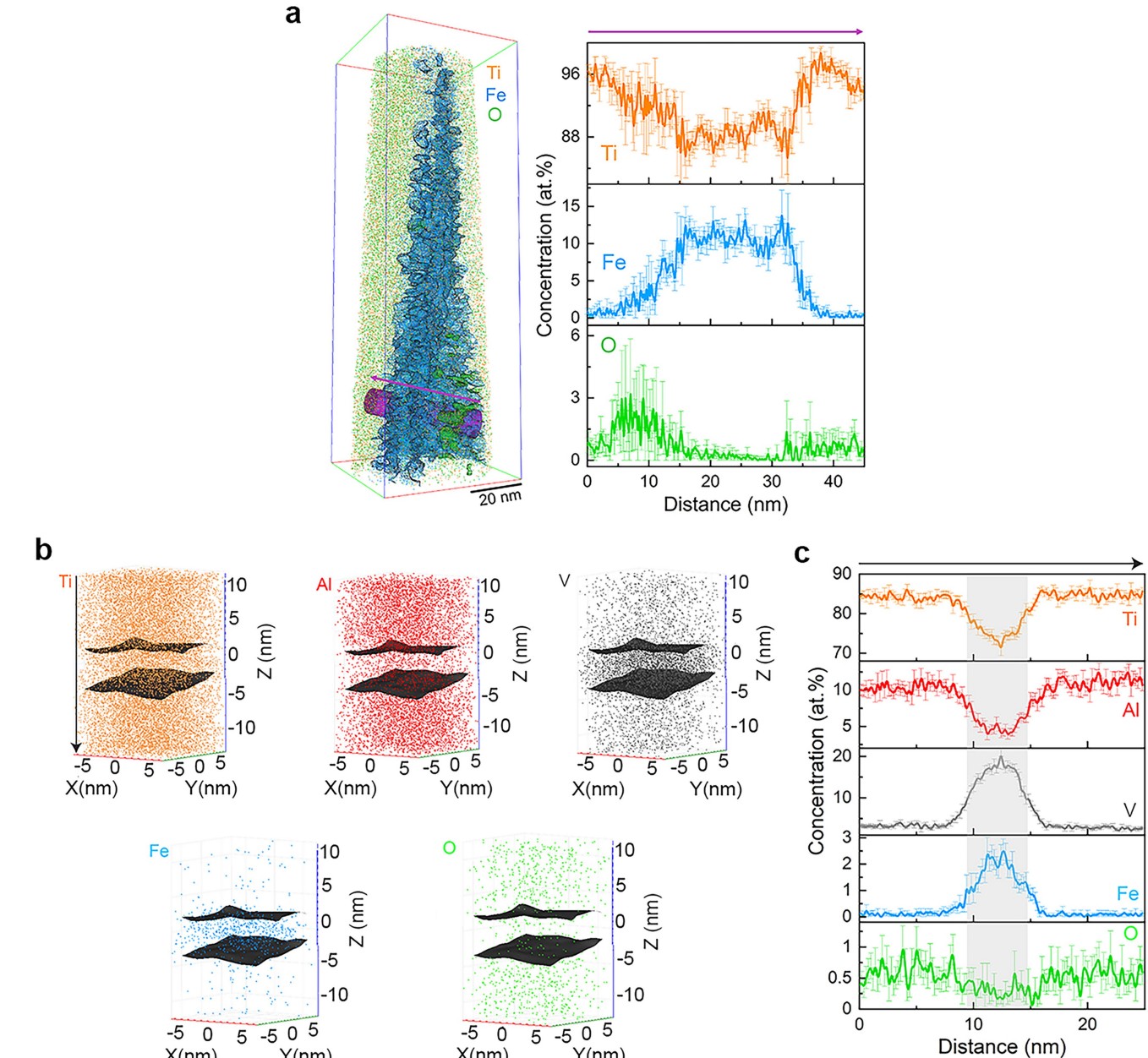

**Extended Data Fig. 4 | APT 3D reconstruction data. a,** Distribution of Ti, O and Fe in an α–β phase region in the DED-fabricated Ti–0.14O–3.23Fe alloy (800 mm min⁻¹, 15 s). The Fe-enriched blue phase is the β-phase, whereas the surrounding Fe-depleted phase is the α-phase. The thresholds for the isocomposition surface for O and Fe are 3 at% and 15 at%, respectively. The violet cylindrical bar represents the 45-nm-long range for the APT profiles shown in the right column, whereas the arrow represents the direction. The Fe-enriched β-phase contains no detectable O by APT. **b,c,** Distribution of Ti, Al,

V, Fe and O in an α–β region in DED-fabricated Ti–6Al–4V–0.22O–0.20Fe alloy (800 mm min⁻¹, 15 s). The two black crack-like planes are isoconcentration surfaces of 10 at% V, representing the α–β phase boundaries. The downward vertical arrow in **b** indicates the direction of the 25-nm-long range for the APT profiles shown in **c**. The Fe-enriched and V-enriched region (between the two black crack-like planes) is the β-phase, which lies between two α-phases. Refer to Supplementary Table 4 for the quantitative results.

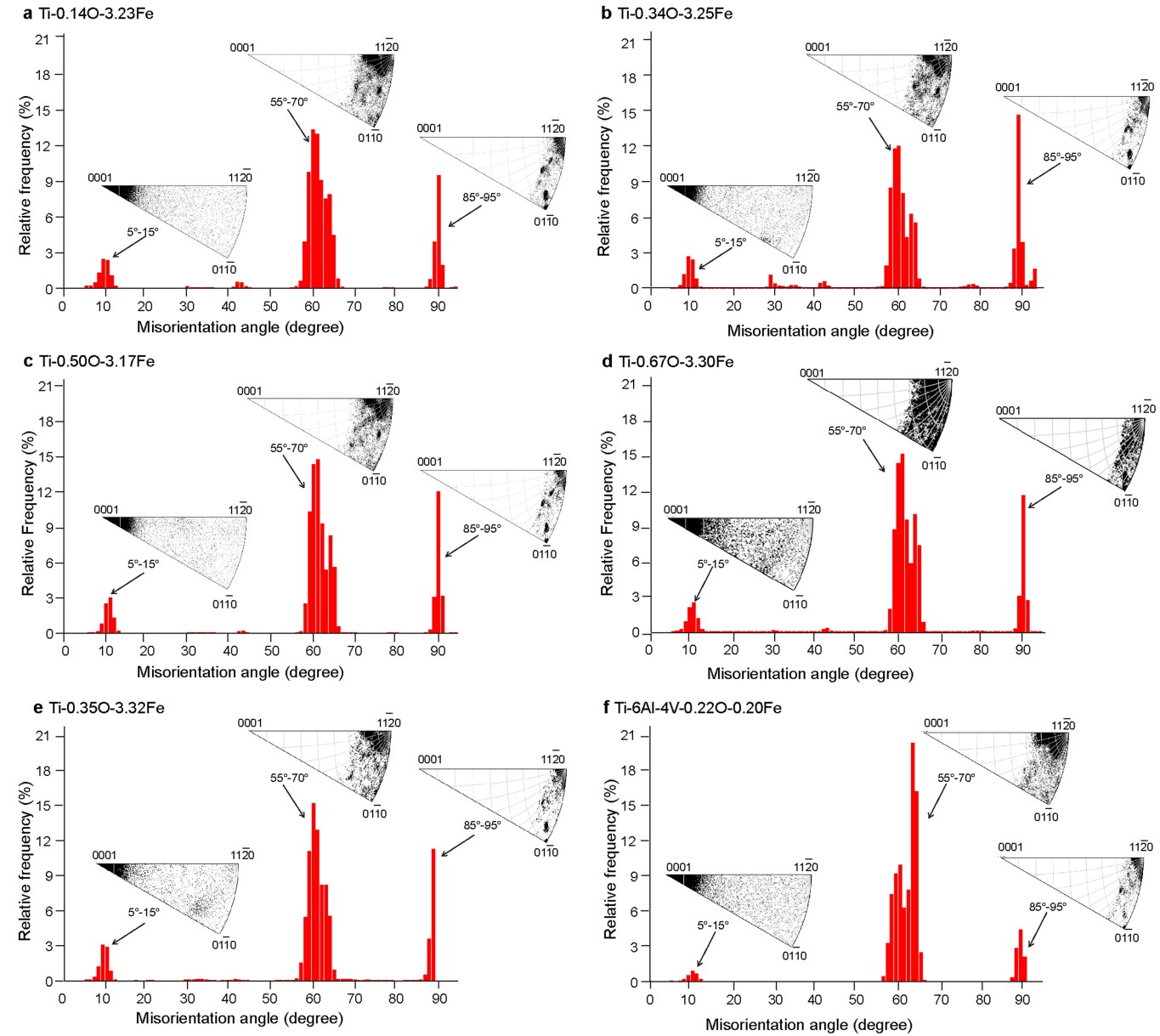

**Extended Data Fig. 5 | EBSD analyses of the crystallographic arrangements of the α-variants in DED-fabricated Ti–O–Fe alloys and reference alloy Ti–6Al–4V–0.22O–0.20Fe.** More than $10^5$ α-variants were analysed in each alloy using EBSD. When considering only abutting α-variant lamellae (that is, by removing the intermediary β-phase lath), the misorientation [11$\bar{2}$0]/60° is

prevalent in the Ti–O–Fe alloys, whereas the misorientation [$\bar{1}$0$\bar{5}$5$\bar{3}$]/63.26°) is prevalent in the Ti–6Al–4V–0.22O–0.20Fe alloy. Alloys **a**–**d** and **f** were fabricated with the same DED conditions (800 mm min⁻¹, 15 s), whereas alloy **e** was fabricated with a zero-layer interval, that is, 800 mm min⁻¹, 0 s. The other conditions were unchanged (Extended Data Table 2).

**Ti-0.34O-3.25Fe**
**α-phase (hcp)**

g

a

b

50 nm

c

50 nm

**Ti-0.34O-3.25Fe**
**β-phase (bcc)**

0$\bar{1}$1  g

d

50 nm

e

50 nm

f

50 nm

**Extended Data Fig. 6 | TEM observations of dislocations in the α and β phases of the Ti−0.34O−3.25Fe alloy after tensile testing ($\varepsilon_f$ = 9.0 ± 0.8%, $\sigma_{UTS}$ = 1,157 ± 3 MPa).** The area selected is close to the tensile fracture surface. **a**,**b**,**d**,**e**, Bright-field images, in which **b** and **e** are closer views. **c**,**f**, Dark-field image. Dislocations are best seen in **c** and **f**. As expected, profound dislocations were observed in the β-phase, which is virtually free of O, whereas dislocations were also observed in the α-phase, which was virtually free of Fe.

**Ti-0.67O-3.30Fe α-phase (hcp)**

g

a

50 nm

b

50 nm

c

50 nm

**Ti-0.67O-3.30Fe β-phase (bcc)**

g

d

50 nm

e

50 nm

f

50 nm

**Extended Data Fig. 7 | TEM observations of dislocations in the α and β phases of the Ti–0.67O–3.30Fe alloy after tensile testing ($\varepsilon = 3.0 \pm 0.8\%$, $\sigma_{UTS} = 1,271 \pm 6$ MPa).** The area selected is close to the tensile fracture surface. **a**,**b**,**d**,**e**, Bright-field images, in which **b** and **e** are closer views. **c**,**f**, Dark-field image. Dislocations are best seen in **c** and **f**. As expected, more dislocations were observed in the β-phase, which is virtually free of O, whereas few dislocations were observed in the α-phase owing to its high O content.

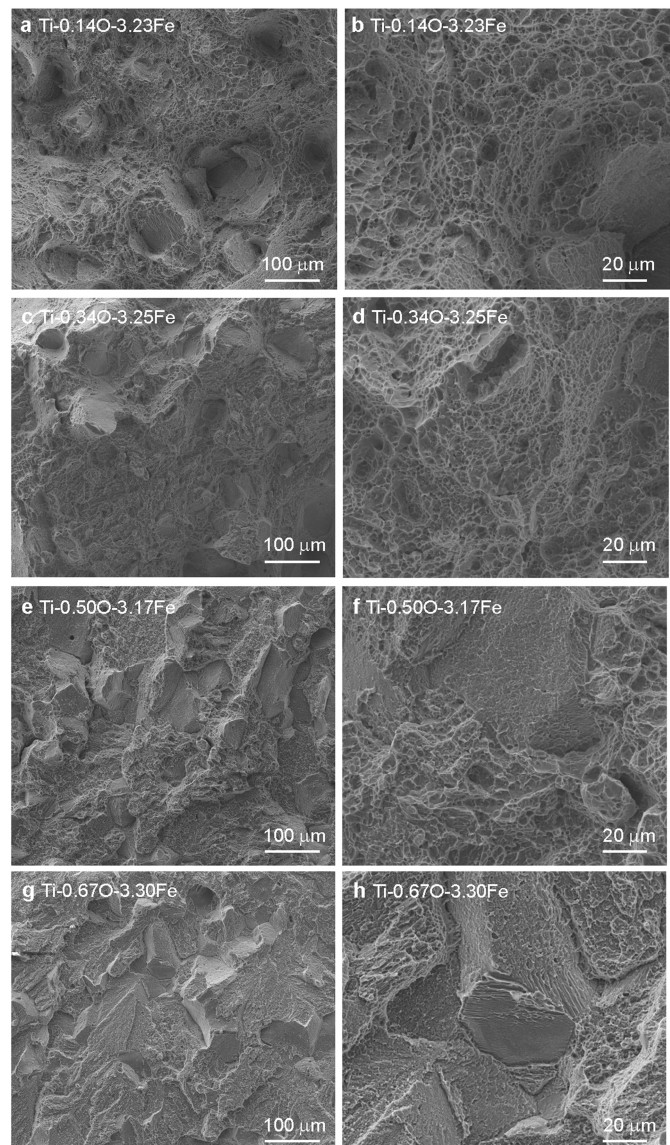

**Extended Data Fig. 8 | Tensile fractographs of each DED-fabricated Ti–O–Fe alloy (800 mm min⁻¹, 15 s).** The tensile properties of each alloy are listed in Extended Data Table 1.

**Extended Data Table 1 | Composition and tensile properties of as-fabricated alloys**

| Alloy composition (wt.%) | Fabrication method | | $\sigma_{0.2}$ (MPa) | $\sigma_{UTS}$ (MPa) | $\varepsilon_f$ (%) | E (GPa) |
|---|---|---|---|---|---|---|
| | DED | | | | | |
| | $v$ (mm/min) | $\tau$ (s) | | | | |
| Ti-0.16O-0.062Fe | 800 | 15 | 443 ± 9 | 541 ± 8 | 33.6 ± 2 | 109.6 ± 5.9 |
| Ti-0.14O-3.23Fe | 800 | 15 | 744 ± 19 | 886 ± 19 | 14.3 ± 1.7 | 111.6 ± 2.5 |
| Ti-0.34O-3.25Fe | 800 | 15 | 1066 ± 4 | 1157 ± 3 | 9.0 ± 0.8 | 117.5 ± 1.5 |
| Ti-0.50O-3.17Fe | 800 | 15 | 1124 ± 7 | 1194 ± 8 | 9.0 ± 0.5 | 119.2 ± 3.9 |
| Ti-0.67O-3.30Fe | 800 | 15 | 1235 ± 4 | 1271 ± 6 | 3.0 ± 0.8 | 116.4 ± 1.4 |
| Ti-0.35O-3.32Fe | 800 | 0 | 898 ± 13 | 1034 ± 9 | 21.9 ± 2.2 | 112.2 ± 4.3 |
| Ti-0.32O-3.12Fe | 800 | 60 | 1143 ± 14 | 1268 ± 10 | 3.2 ± 0.3 | 105.2 ± 4.9 |
| Ti-0.35O-3.36Fe | 800 | 120 | 1206 ± 10 | 1303 ± 18 | 2.2 ± 0.6 | 109.3 ± 4.6 |
| Ti-0.35O-3.17Fe | 600 | 15 | 935 ± 14 | 1085 ± 17 | 14.0 ± 0.7 | 110.9 ± 1.4 |
| Ti-0.33O-3.11Fe | Cu-mould casting | | 986 ± 40 | 1080 ± 45 | 6.3 ± 1.6 | 106.1 ± 4.8 |

$\sigma_{0.2}$, yield strength; $\sigma_{UTS}$, ultimate tensile strength; $\varepsilon_f$, strain to fracture; $E$, modulus of elasticity.

All properties are shown as mean±s.d. ($n$=3 or 4).

$v$, laser scan speed; $\tau$, layer-to-layer interval time.

**Extended Data Table 2 | Processing parameters and scan path for simulation and fabrication**

| | Heat source | Laser power (W) | Laser scan speed (mm/min) | Laser energy density (J/mm²) | Layer interval time (s) | Scan strategy (Bi-directional) |
|---|---|---|---|---|---|---|
| Simulation |  | 500 | 800 | 25 | 0 |  |
| | | 500 | 800 | 25 | 15 | |
| | | 500 | 800 | 25 | 60 | Odd layer scan |
| | | 500 | 800 | 25 | 120 | |
| | | 500 | 600 | 33.3 | 15 | |
| | | 500 | 400 | 50 | 15 | |
| | | 500 | 200 | 100 | 15 | Even layer scan |

| | Laser spot size (mm) | Laser power (W) | Laser scan speed (mm/min) | Powder flow rate (g/min) | Layer interval time (s) | |
|---|---|---|---|---|---|---|
| Experiment | 1.5 | 500 | 800 | 1.7 | 15 |  |
| | 1.5 | 500 | 800 | 1.7 | 0 | Odd layer |
| | 1.5 | 500 | 800 | 1.7 | 60 | |
| | 1.5 | 500 | 800 | 1.7 | 120 | Even layer |
| | 1.5 | 500 | 600 | 1.7 | 15 | |

*The heat source for simulation has a Gaussian parameter of 1 and an absorption efficiency of 45%.

**Layer thickness is 0.2 mm and step-over width is 1.05 mm for both simulation and fabrication.

***The fabrication processes use a carrier gas (He) flow rate of 10 l min⁻¹ and a shielding gas (Ar) flow rate of 16 l min⁻¹.