## [Peer Review File · Nature]

Manuscript Title: Strong and ductile titanium-oxygen-iron alloys by additive manufacturing

Reviewer Comments & Author Rebuttals

Reviewer Reports on the Initial Version:

Referee expertise:

Referee #1: metallurgy, additive manufacturing

Referee #2: metallurgy, additive manufacturing

Referees' comments:

Referee #1 (Remarks to the Author):

Key results: the authors reveal the origins of an ultra fine alpha beta microstructure associated to improved tensile properties. In particular, the role of oxygen and iron in the development of such titanium microstructures is examined using multiple state of the art characterisation techniques that allow to rationalise the unique range of performances of the investigated alloys.

Validity: the manuscript contains no apparent flaws and is very well written and substantiated.

Originality and significance: there are numerous investigations on the process property relationship of ternary Ti-Fe-O alloys, so the alloy design strategy per se is not novel. The insights generated by this work on how O and Fe partition in the two-phase structure are however new. It is not clear how the manufacturing route investigated brings significant advantages over conventional manufacturing techniques particularly when the microstructure investigated is a microstructure close to equilibrium which could be obtained by other manufacturing means. This is remarkable considering that the raw feedstock is heterogenous and mixing occur at a minute length scale. The alloy design strategy is derived from metallurgical insights but arguably the alloy is not conceptualised with "(additive) manufacturing in mind". In fact three of the proposed alloys (3.30Fe-0.67O, 3.17Fe-0.50O, 3.25Fe-0.34O) possess limited ductility which are likely to cause the same "printability headaches" that grade 23 displays when coupons other than rectangles are printed. If the argument is that the alloys are low-cost alternative to grade23, it should be noted that most of the costs involved in additive come from processing operations and not the material costs. Therefore, arguably, the AM community is in need of alloys capable to tolerate large processing windows which is unlikely to be the case here. It is not clear how the findings could be applied to sectors other than additive manufacturing, perhaps the authors could make a comment on that?

Data & methodology: the alloy fabrication from elemental powders will always present doubts on

chemical homogeneity. The authors use a DED approach with a relative large spot size which apparently mitigates inhomogeneous mixing. The authors point of view of this topic would be useful.

Appropriate use of statistics and treatment of uncertainties: not applicable, all fine.

Conclusions: conclusions and data interpretation are robust.

Suggested improvements: the reviewer is puzzled by the title of the manuscript. In what way additive manufacturing enables the fabrication of these ternaries' microstructures? AM encompasses a wide range of fusion techniques and the findings seems relevant to a particular process which enables the formation of near-equilibrium microstructures.

References: all references are appropriate.

Clarity and context: the physical metallurgy of titanium alloy is dealt very well and is very accessible. The applicability of the findings, supposedly materials well versed for structural room temperature applications, remains vague. If the alloy is enabled by "additive manufacturing" what are the relevant sectors that are being targeted? Ti-6Al-4V, the benchmark alloy, is so amenable because it offers a combination of structural and non-structural properties that are relevant to aerospace, biomedical and marine sectors. How does the new range of alloy investigated in the present compare? At least a prediction should be made to highlight the impact of the work.

Referee #2 (Remarks to the Author):

This paper describes the fabrication and evaluation of Ti-Fe-O alloys, manufactured using additive manufacturing, and found to be stronger than CP-Ti and Ti-6Al-4V and with notable ductility. The paper uses a number of cutting-edge techniques to characterize the material, most notably atom probe tomography to detail the spatial distributions of elements.

The key results are that by varying the fraction of two relatively inexpensive (or free) alloying elements, and fabricating materials with a directed energy deposition additive manufacturing technique, the team achieved high strength and notable ductility components. The team also detailed the location of the alloying elements within the phases of the fabricated material through atom probe tomography.

The data presented are of high quality. The authors should comment on the potential sources of variability in the stress-strain curves. While some curves are repeatable, the Ti-3.23Fe-0.14O are not, and it would be informative to note the potential sources of this variability.

The orientation of the extracted mechanical test specimens should be specified.

In directed energy deposition, the microstructure changes from near the baseplate to further up in the build, with this transient existing several mm into the height of the build. Were the samples all extracted from the same height from analogous builds? Or what method was used to check that the

samples had similar microstructures?

For the alloys examined, it is implied that the additive manufacturing processing was advantageous for fabricating the observed microstructure and achieving the measured properties. However, no discussion is provided on the selection of processing parameters. The authors did simulate the additive manufacturing process, but indicated in the supplemental document that, “Based on simulations and our experiences, the LMD processing parameters for alloy fabrication were chosen to be laser power of 500 W, laser spot size of 1.5 mm, scan speed of 800 mm/min, layer thickness of 0.2 mm, and scan spacing of 1.05 mm.” The authors should detail what metrics were used to determine these parameters. Were they bead shape, reduced thermal cycling, or some critical cooling rate? What thermal history bounds are required for achieving the observed microstructures? Was the optimization based on creating dense samples or achieving some optimal temperature profile?

The team should comment on the applicability of other manufacturing processes for this alloy. Is additive manufacturing expected to be the only or best method for these alloys? Why (in terms of thermal history/processing steps)?

The authors talk about the absence of beta-flecks. For the reader, it would be good to mention the expected size of these and concentration in the related alloys.

On page 6-7, the authors state, “The strengthening potency of the Fe is inferred by comparing the ultimate tensile strength (UTS) values of Ti-3.23Fe-0.14O and CP Ti-0.062Fe 0.16O, demonstrating ~105 MPa/1.0wt.% Fe, close to the reported 75 MPa/1.0wt.% Fe.” Is the “75 MPa/1.0wt.% Fe” for pure Ti? Is this a theoretical or experimentally result that is referenced in 31?

More details should be added on the thermodynamic calculations the team used. The CALPHAD method was mentioned several times, and only in its 4th mention (page 10 in the main document, and in the supplemental document) is it stated that equilibrium calculations were used. As additive manufacturing is a far-from-equilibrium process, the authors should discuss the applicability and limitations of these equilibrium calculations.

A key limitation of this study is that while an alloy is presented, the authors do not emphasize the mechanisms for the strength and ductility. They do provide a list of potential reasons for the observed properties in their summary paragraph, but these are not discussed at length in the body of the document, and they do not dive deeply into investigating these deformation mechanisms to identify what about the microstructure or chemistry results in the observed properties. For example, the authors could perform control studies to vary the microstructure, but not chemistry, to see how changing the size of the microstructural features impacts the properties. This points to other comments in this review – is the composition or the additive manufacturing process the key to the success of these alloys? Is it the combination? If additive manufacturing processing is needed, the authors should expand on the phase transformations/phase formations that occur, as well as the critical processing conditions needed for these alloys.

The authors state that they have a large volume fraction of “virtually O-free continuous beta phase”

– but how does this compare to the O in the beta phase in Ti-6Al-4V with oxygen for example?

Fig. 1A: are all 5 of these samples the same composition? Why is the scan pattern of the 4th sample different from the rest? The authors should note which samples these are.

Fig. 1B: in the simulation data, why are there many peaks followed by one peak? Even for a bimodal scan, the laser should pass nearby the same point multiple times during scanning a subsequent layer.

Fig. 4: the labels are cut off (O, HCP, BCC, 15% are all going to a second line and cut off in the presented images).

Extended Data Fig. 3: y-axis should be labeled stress, not strength.

Extended Data Fig. 4: there's an extra comma at the end of the caption, and a scale bar would be beneficial in the left figure (although it can be implied from the right figure).

Author Rebuttals to Initial Comments:

We would like to express our gratitude to the expert reviewers who provided very insightful comments and suggestions that helped us to reshape this work.

We first focus on addressing the comments and suggestions received from the referees, before turning to the comments and suggestions from the Editor.

Since this is a lengthy response, we have used a separate sequence of figures and tables, and duplicate data are used for similar comments for convenience.

All the new data (figures and tables) presented in this response have been incorporated into the revision, either in the Main Manuscript file, the Extended Data File, or the Supplementary Data File, as indicated where the data are presented below.

Changes made to the manuscript are specified explicitly at the end of each response. In addition, following the traditional notation for α - β Ti alloys (where the α -stabilising elements are listed first, such as in Ti-6Al-4V), we have changed Ti-Fe-O to **Ti-O-Fe**.

A snapshot of the main advances are now listed:

- 1) “Control studies” and the required thermal history bounds or processing window: a large processing window has been established by simulations and validation experiments through “control studies” – relevant for all geometries, in principle.
- 2) Outstanding tensile ductility (ϵ_f): within the established processing window, our alloys can now achieve up to $\epsilon_f = 21.9 \pm 2.2\%$ while maintaining high tensile strength ($> 1,000$ MPa).
- 3) Oxygen in the β -phase of α - β **Ti-6Al-4V**: it is fundamentally different from the near oxygen-free state in the β -phase of our α - β Ti-O-Fe alloys under the same fabrication conditions, where the β -phase in Ti-6Al-4V contains substantial oxygen — about 50% of the oxygen in the α -phase, consistent with the literature (40-80%).
- 4) Deformation mechanisms: New TEM results have clarified the difference in dislocation multiplication – closely related to the oxygen nano-heterogeneity in the α -phase and the near oxygen-free β -phase discovered in these Ti-O-Fe alloys. The deformation mechanisms are elucidated.

- 5) Shape casting of Ti-0.35O-3Fe: ingot metallurgy samples were made using copper-mould casting and their microstructures (containing ~400 β -flecks/cm²) and tensile properties (much lower tensile ductility) were characterised — highlighting the advantages of AM in producing these Ti-O-Fe alloys.
- 6) Additional impacts: our work provides **a potential game-changing approach** to revitalising large amounts of off-grade sponge Ti or sponge Ti-O-Fe (5-10% of all sponge Ti production). This is significant because sponge Ti production is extremely energy-intensive (e.g. ~5-10 times that of primary aluminium production). Our work further offers a potential pathfinding approach to various other new alloy systems (Supplementary Note 6).

Finally, we conclude that the success of our Ti-O-Fe alloys stems from the combination of alloy design and additive manufacturing (AM) process design, which leads to a unique integration of multi-scale (micro-nano-atomic) microstructural features that are responsible for the outstanding tensile properties obtained. Both are important and herald an important pathfinder or metallurgical design strategy that couples alloy theory and processing theory.

Referee #1

1. Originality and significance

1.1 It is not clear how the manufacturing route investigated brings significant advantages over conventional manufacturing techniques particularly when the microstructure investigated is a microstructure close to equilibrium which could be obtained by other manufacturing means.

Response: Thank you for this important comment. We first summarise the advantages and then substantiate each of them. Compared to conventional manufacturing techniques, the investigated laser metal deposition (**LMD**) process offers:

- 1) the capability to completely avoid β -fleck formation in the designed Ti-O-Fe alloys, allowing the use of Fe as the primary β -Ti stabiliser with microstructural homogeneity
- 2) the capability for free form fabrication (F^3) of large builds with high quality
- 3) the capability to create fine microstructural features in the as-fabricated state with outstanding tensile properties, and
- 4) the capability and flexibility of tuning mechanical properties within a large processing window for different part sizes and/or shapes.

1) Capability to completely avoid β -fleck formation

To put this issue in context, we prepared small ingot samples (120 mm \times 12 mm \times 5 mm – length \times width \times thickness) of the Ti-0.35O-3.0Fe alloy (measured: Ti-0.33O-3.11Fe), using a vacuum arc melter, followed by water-cooled Cu-mould casting (see Methods).

Fig. A1 shows the resulting as-cast microstructure (lamellar α - β type). The Fe-stabilized β -flecks (5.6 wt.%Fe) exhibited a number density of ~ 400 β -flecks/cm², **Fig. A1 a-d**, despite the use of water-cooled Cu-moulds and small sample size (32 g/sample). This demonstrates the severity of the β -fleck formation in these simple alloys when prepared using ingot metallurgy.

β -flecks can reduce the tensile ductility **by > 50%** (from 10-13% to 5-6%) and the low-cycle fatigue strength **by > 90%** in the case of Ti-1023 (Ti-10V-2Fe-3Al)¹. They are difficult to eliminate in large ingots. Therefore, we may conclude that bulk solidification-based conventional manufacturing techniques are not suitable for these Ti-O-Fe alloys.

In contrast, β -flecks were completely absent in all our LMD-fabricated Ti-O-Fe alloys with similar, or even higher Fe content.

Fig. A1 Microstructure of the as-cast reference alloy Ti-0.35O-3Fe (measured: Ti-0.33O-3.11Fe). **(a)** An overview of the microstructure comprising α - β lamellae and β -flecks. **(b)** A closer view of a selected β -fleck. **(c)** Energy Dispersive Spectroscopy (EDS) mapping of Fe in **(b)** to confirm the Fe-enriched β -fleck. **(d)** A selected β -fleck phase for EDS line scan analysis. Also shown is the as-cast porosity. **(e)** EDS line scan further confirms the Fe-enriched β -flecks. **(f)** A backscatter electron (BSE) image showing the α - β lamellae.

2) Free form fabrication (F³) of large builds with high quality

We have been working on the TRUMPF-based LMD process since ~2014 and have found that it allows high-quality F³ of large builds. Specifically, we have not observed lack-of-fusion defects on the fracture surfaces of Ti alloys using our established processing conditions. The largest part we have manufactured is 450 mm in diameter. The prevalent defect is the presence of occasional small spherical pores.

3) Capability to create fine microstructural features in the as-fabricated state with outstanding tensile properties

The microstructure generated in the as-fabricated state is free of β -flecks and consists of submicron-thick α -laths with blunt tips and thinner β -laths, dispersed within a fine equiaxed prior- β grain structure (**Fig. 1 I-K**). We claim that the corresponding tensile properties are indeed outstanding, as listed in **Table A1** and shown in **Fig. A2** (next page, Extended Data Fig. 2 provides all tensile stress-strain curves).

For example, the as-fabricated Ti-0.35O-3Fe alloy achieved tensile ductility (ϵ_f) of **21.9 ± 2.2%** and tensile strength (σ_{UTS}) of **1034 ± 9 MPa** when fabricated with a scan speed of 800 mm/min and a layer interval time of 0 s (LMD-1 in **Table A1**). Changing the processing condition to 600 mm/min and 15 s resulted in $\epsilon_f = \mathbf{14.0 \pm 0.7\%}$ and $\sigma_{UTS} = \mathbf{1085 \pm 17 MPa}$ (LMD-2 in **Table A1**). Both sets of processing conditions are within our processing window (the green zone in **Fig. 1C**). In contrast, the as-cast ingot of this alloy is much less ductile (50-70% lower) at similar σ_{UTS} , due to the prevalence of the β -flecks and uncontrolled porosity (**Fig. A1 a-d**).

Sintering-based powder metallurgy (PM) processes, including (i) conventional powder metallurgy (C-PM: press and sinter), (ii) metal injection moulding (MIM, small parts, ≤ 65 mm in most cases) and (iii) hot isostatic pressing (HIP) all represent examples of processing technologies that **can also avoid** β -fleck formation and produce lamellar α - β microstructures, close to equilibrium. However, C-PM and MIM usually leads to high residual porosity (~2 vol.%, pressureless sintering), coarse β grains (1200-1300 °C for 1-4 hours), and coarse α - β lamellae (slow furnace cooling). They are not expected to exhibit high tensile properties. HIP is a possible option but is not comparable to AM for net shape formation. In addition, our HIP experiences with Ti-6Al-4V indicate that even with a low HIP temperature (820 °C, 200 MPa), HIP still yields thick α -laths (~2 μm thick), resulting in much lower tensile strengths.

Fig. A2 Comparison of the tensile properties of water-cooled Cu-mould cast Ti-0.35O-3Fe alloy ingot (measured: Ti-0.33O-3.11Fe) and LMD-fabricated Ti-0.35O-3Fe alloy (LMD-1: Ti-0.35O-3.32Fe; LMD-2: Ti-0.35O-3.17Fe). The as-cast alloy is much less ductile (50-70% lower) than the LMD-fabricated alloys with similar strength.

Table A1 Tensile properties of as-cast and LMD-fabricated Ti-0.35O-3Fe alloy

Alloy composition (wt.%)	$\sigma_{0.2}$ (MPa)	σ_{UTS} (MPa)	Strain to fracture (ϵ_f , %)	Elastic modulus (E, GPa)
Ti-0.35O-3Fe (LMD-1*) (measured: Ti-0.35O-3.32Fe)	898 ± 13	1034 ± 9	21.9 ± 2.2	112.2 ± 4.3
Ti-0.35O-3Fe (LMD-2**) (measured: Ti-0.35O-3.17Fe)	935 ± 14	1085 ± 17	14.0 ± 0.7	110.9 ± 1.4
Ti-0.33O-3Fe (as-cast) (measured: Ti-0.33O-3.11Fe)	986 ± 40	1080 ± 45	6.3 ± 1.6	106.1 ± 4.8

* LMD-1: scan speed: 800 mm/min; layer interval: 0 s (see Extended Table 2 for other conditions)

** LMD-2: scan speed: 600 mm/min; layer interval: 15 s (see Extended Table 2 for other conditions)

4) Capability and flexibility of tuning mechanical properties

We have established a large processing window (the green zone in **Fig. 1C**, reproduced below as **Fig. A3**) by simulations and experimental validation via “control studies”. The scan speed can be varied from 200 to 1200 mm/min while the layer interval time can be tuned from 0 to 120 s (covering most practical operations for LMD).

Fig. A3 LMD processing window (green zone) determined for the fabrication of Ti-O-Fe alloys. It is based on the evolution of the average temperature at the central points of layer 1 (all samples comprise 25 layers). The green zone can accommodate scan speeds from 200-1200 mm/min, and layer interval between 0 to 120 s, covering most practical operations for LMD.

We take the Ti-0.35O-3Fe alloy as an example to demonstrate the capability and flexibility of tuning its mechanical properties. **Fig. A4** (next page) shows the tensile properties of this alloy fabricated using three sets of LMD conditions in the green zone of **Fig. A3**. As observed, the tensile properties can be tuned from $\epsilon_f = 21.9 \pm 2.2 \%$ and $\sigma_{UTS} = 1034 \pm 9 \text{ MPa}$ to $\epsilon_f = 9.0 \pm 0.8 \%$ and $\sigma_{UTS} = 1157 \pm 3 \text{ MPa}$. The range can be further extended by simply changing the processing conditions within the green zone. This capability and flexibility is desirable and important for different applications.

Therefore, **our conclusion** is that AM remains the essential method of choice for the net-shape manufacture of these alloys, with outstanding tensile properties and high flexibility.

Changes made to the manuscript: **Fig. A1** has been added as **Extended Data Fig. 1**. This section has been added as “Supplementary Note 3 Advantages of AM in producing the designed Ti-O-Fe alloys” and referred to in the manuscript.

Fig. A4 Tensile properties of the same Ti-0.35O-3Fe alloy for different combinations of strength and ductility through tuning of the LMD conditions. This illustrates the potential for building a large and/or complex parts with different wall thicknesses.

1.2 The alloy design strategy is derived from metallurgical insights but arguably the alloy is not conceptualised with “(additive) manufacturing in mind”.

Response: We appreciate this comment. In our view, it is the case that the alloy compositions are derived from metallurgical insights for additive manufacturing (AM). It is only by the appropriate combination of alloy design and AM process design that the attractive properties shown in **Fig. A4** are achieved.

Changes made to the manuscript: we have emphasised in our revised main text that it is the appropriate combination of alloy design and AM process design that delivers the attractive properties of these α - β Ti-(0.35-0.50)O-3Fe alloys.

1.3 In fact three of the proposed alloys (3.30Fe-0.67O, 3.17Fe-0.50O, 3.25Fe-0.34O) possess limited ductility which are likely to cause the same “printability headaches” that grade 23 displays when coupons other than rectangles are printed.

Response: Thank you for these two important comments (limited ductility and coupon shape). We first give our conclusions and then elaborate on how we arrived at these conclusions.

Limited ductility: within our established processing window (**Fig. A3**), we can now easily increase the tensile ductility ε_f of the Ti-0.35O-3Fe alloy to 21.9 ± 2.2 % while maintaining a σ_{UTS} ($> 1,000$ MPa). For α - β Ti alloys, this is an outstanding ε_f .

Coupon shape: As shown in **Fig. A3**, our established processing window accommodates scan speeds of 200-1200 mm/min and layer interval times of 0-120 s at the laser power of 500 W, and laser spot size of 1.5 mm. This range covers most practical operations for LMD of Ti alloys, and we note that changing the laser power would enable further modifications of the processing window. The influence of the wall thickness or coupon shape or build height can be simulated prior to the manufacture so that processing parameters can be adjusted to allow for a generally uniform microstructure throughout the part. This is an important ongoing trend for LMD processing of large and/or complex parts.

With the increased capacity of high-fidelity simulations and the flexibility available from the AM processing window, we are confident that it is no longer a significant challenge to fabricate complex components with a consistent microstructure using the alloy system introduced here.

We now briefly discuss our processing window (**Fig. A3**) and its validation.

1.3.1 Design of the processing window (independent of coupon geometry)

Our goal was to create a fine α - β lamellar microstructure through in-situ decomposition of the α' -phase. To this end, the process window needed to meet two essential requirements:

- (i) an appropriate cooling rate (≥ 400 °C/s) when the β -phase in each solidified layer was cooled from the single β -phase region, so achieving the formation of α' , and
- (ii) an appropriate stabilised temperature window across each layer (e.g. 600-750 °C) to allow the $\alpha' \rightarrow \alpha$ - β decomposition (excluding the last few layers)

As mentioned above, the influence of the wall thickness or coupon shape or build height can be simulated prior to the manufacture so that processing parameters can be adjusted.

We first considered scan speeds from 200-1200 mm/min and layer interval times from 0-120 s (see **Extended Data Table 2** for other parameters) and assessed the resulting average cooling rates at the centres of layers 1, 13 and 25 (25-layer coupon) from the molten state to 800 °C (the estimated M_s). The results are summarised in **Table A2**. The cooling rate satisfies the first requirement in each case, as expected.

Table A2 LMD conditions considered for establishing the processing window

Average cooling rate when cooled from the liquid state to 800 °C			
Manufacturing condition (scan speed – layer interval)	Dimensions of the 25- layer coupon (length × width × thickness)	Sample position	Average Cooling rate
1200 mm/min – 15 s	40 mm × 10 mm × 5 mm	Centre of layer 1	3,868 °C/s
800 mm/min – 0 s	40 mm × 10 mm × 5 mm	Centre of layer 1	2,356 °C/s
		Centre of layer 13	818 °C/s
		Centre of layer 25	646 °C/s
800 mm/min – 15 s	40 mm × 10 mm × 5 mm	Centre of layer 1	2,432 °C/s
		Centre of layer 13	1,444 °C/s
		Centre of layer 25	1,404 °C/s
800 mm/min – 60 s	40 mm × 10 mm × 5 mm	Centre of layer 1	2,881 °C/s
		Centre of layer 13	2,224 °C/s
		Centre of layer 25	2,401 °C/s
800 mm/min – 120 s	40 mm × 10 mm × 5 mm	Centre of layer 1	3,131 °C/s
		Centre of layer 13	2,619 °C/s
		Centre of layer 25	2,739 °C/s
600 mm/min – 15 s	40 mm × 10 mm × 5 mm	Centre of layer 1	1,855 °C/s
		Centre of layer 13	1,016 °C/s
		Centre of layer 25	1,008 °C/s
400 mm/min – 15 s	40 mm × 10 mm × 5 mm	Centre of layer 1	1,116 °C/s
200 mm/min – 15 s	40 mm × 10 mm × 5 mm	Centre of layer 1	450 °C/s

We then focused on monitoring the evolution of the average temperature at the centre of layer 1, since this will reveal the information from all subsequent 24 layers of the deposition. The average temperature was calculated over the scanning time for each layer (e.g. 35.85 s at 600 mm/min for each odd-number layer) plus the layer interval time (e.g. 15 s). The results are already shown in **Fig. A3**. For ease of comparison, we reproduce it below as **Fig. A5a**.

Decomposition of α' begins at 400 °C after an appropriate hold², yet complete decomposition requires a hold of ≥ 2 h at 800 °C^{2,3}. Fortunately, however, the effect of the thermal pulses (**Fig. 1B**) introduced during LMD is to significantly accelerate the decomposition process⁴.

On the other hand, there is a risk that the microstructure could experience severe coarsening at temperature of ≥ 800 °C, while below 480 °C (the minimum stress relief temperature), decomposition of the α' may not be complete. In addition, the stabilised temperature increases with increasing build height (see **Fig. A5b**), which means that the stabilised temperature of the first layer should not be set too high. Based on these considerations, we chose 480-800 °C in

Fig. A5a (the green zone) as our required thermal history bounds or processing window for experimental validation. The same green zone is also highlighted in **Fig. A5b**.

Fig. A5 (a) Evolution of the average temperature at the centre of layer 1 in a 25-layer coupon during LMD with scan speed (200-1200 mm/min) and layer interval time (0-120 s). **(b)** Evolution of the average temperature at the central points of layers 1, 8, 13 and 18 at two scan speeds (800 mm/min and 600 mm/min). The stabilised temperature increases with reducing layer interval time or scan speed or with increasing build height. Stabilisation occurs from the 8th or 9th layer of deposition. The green zone is the predicted processing window.

1.3.2 Experimental validation of the processing window

We fabricated test coupons of the Ti-0.35O-3Fe alloy both within and outside the green zone mapped in **Fig. A5**. The resulting tensile stress-strain curves are summarised in **Fig. A6**, where the horizontal broken line divides the tensile properties into two regions: outside (above) and within (below) the processing window. The results fully support our simulation-informed processing window and design concept. Specifically, we note:

- 1) Within the upper part of the green zone (600-800 °C), samples achieved $\epsilon_f = 21.9 \pm 2.2\%$ and $\sigma_{UTS} = 1034 \pm 9$ MPa (800 mm/min, 0 s) and $\epsilon_f = 14.0 \pm 0.7\%$ and $\sigma_{UTS} = 1085 \pm 17$ MPa (600 mm/min, 15 s). **These excellent tensile properties highlight the potential of Ti-O-Fe alloys as unexplored (for AM), ductile and strong novel α - β Ti alloys** (the Ti-0.5O-3Fe alloy can be similarly processed).
- 2) Within the lower part of the green zone (500-600 °C), the samples achieved $\epsilon_f = 9.0 \pm 0.8\%$ and $\sigma_{UTS} = 1157 \pm 3$ MPa (800 mm/min, 15 s).
- 3) Outside the green zone (< 450 °C), the ductility dropped to $2.2 \pm 0.6\%$ with very high $\sigma_{UTS} = 1303 \pm 18$ MPa (800 mm/min, 120 s; the case for 60 s is similar).

Fig. A6 Engineering stress-strain curves of the Ti-0.35O-3Fe alloy fabricated both within and outside the green zone in **Fig. A5**. The results fully support our simulation-informed processing window or green zone and design concept.

We have thus established a large processing window that accommodates **numerous LMD conditions** (e.g. scan speed: 200-1200 mm/min; layer interval: 0-120 s). We consider that the reviewer's concerns (limited ductility and printability) are hereby appropriately addressed.

Changes made to the manuscript: A section entitled "Selection of LMD parameters and the required thermal history bounds or processing window" has been added to the Extended Data File. **Fig. A5a** (as main text **Fig. 1C**), **Fig. A5b** (as Supplementary Fig. 4), **Fig. A6** (as Extended Data Fig. 2b) and **Table A2** (as Supplementary Table 3) have been added to the revised manuscript.

1.4 If the argument is that the alloys are low-cost alternative to grade23, it should be noted that most of the costs involved in additive come from processing operations and not the material costs. Therefore, arguably, the AM community is in need of alloys capable to tolerate large processing windows which is unlikely to be the case here.

Response: We concur and have revised our main text accordingly. On the other hand, we have drawn new emphasis that since our Ti-O-Fe alloys could be manufactured from powders made from off-grade sponge Ti, leveraging their intrinsically high Fe and O content, this work offers a potentially game-changing approach to revitalising large amounts of off-grade sponge Ti (see **4. Clarity and context**), helping to reduce the carbon footprint of Ti part production. Sponge Ti production is extremely energy-intensive at 423 GJ/ton⁵ (5-10 times that of the primary Al production). Off-grade sponge Ti accounts for 5-10% of the total sponge Ti production⁶⁻⁸ and so offers a potentially high environmental impact.

Large processing windows: we trust that our response to **Comment 1.3** above has answered this question. We have demonstrated a large processing window with high flexibility.

Changes made to the manuscript: We have included a statement related to the off-grade sponge titanium in the abstract, text and summary and added Supplementary Note 6.

1.5 It is not clear how the findings could be applied to sectors other than additive manufacturing, perhaps the authors could make a comment on that?

Response: Thank you for this important question. We outline below the potential implications or applications of the findings of this work inside and outside the titanium field.

Inside the titanium field

As discussed above (1.1 Advantages), with a **focus** on net-shape manufacturing, the findings are best applied to the metal AM sector. Without considering net-shape manufacturing, we envisage that our α - β Ti-O-Fe alloys could also be made with excellent tensile properties through PM and elaborate thermo-mechanical processing.

For example, ingots of Ti-O-Fe alloys can be made from hydride-dihydride **high-oxygen** Ti powder (**low value**) with Fe powder by cold isostatic pressing and sintering. Then, through β -field forging + multi-axial α - β field forging + extrusion/rolling, the pre-sintered Ti-O-Fe alloy ingots could be manufactured into billets or plates with fine equiaxed β grains (20-40 μm – much finer than that as-deposited here, main text **Fig. 1D-1G**). Finally, heat treatments could be explored to produce fine α - β lamellae within the fine equiaxed prior β grains. Such Ti-O-Fe microstructures are expected to possess attractive tensile properties. Machining will be the ultimate net-shape formation step. The process would normally lead to a high buy-to-fly ratio.

The aforementioned promise to utilise large amounts of off-grade sponge Ti due to excessive O and Fe contaminants is also relevant here and this point will be elaborated subsequently in our response to **Comment 4.2**.

Similar to oxygen, nitrogen (N) embrittlement also occurs in Ti⁹. As a result, N is tightly controlled to $\leq 0.05\%N$ in Ti alloys by conventional manufacturing, even though N is more potent than O as both α -phase stabiliser and α -phase strengthener. Based on this work, it may become promising to develop a new class of strong and ductile α - β Ti-N-Fe alloys with similar partitioning of N and Fe (or using a different β -stabiliser, based on DFT calculations and predictions – see below). This would help to further change the landscape of α - β Ti alloys and their manufacture and may lead to potential new interstitial engineering for Ti alloys.

From an α - β titanium alloy design perspective, the unique partitioning of oxygen and iron in the two phases (α and β) is fundamental to the success of these Ti-O-Fe alloys (a suitable β -phase volume fraction and the suppression of β -flecks are also important). This offers a different α - β titanium alloy design pathfinding approach. For example, systematic DFT calculations and predictions could be performed to map out the combination of O with each β -stabiliser (Mo, V, Cr, Fe, Mn, Ni, Co, Nb, Ta, and W) for similar partitioning in the α and β phases. These combinations could lead to the design of potentially attractive new α - β titanium alloys. The same approach could be applied to nitrogen in titanium just described.

Outside the titanium field

The physical metallurgy of zirconium (Zr) is similar to Ti. We envisage that the findings of this work would be applicable to the development of strong and ductile α - β Zr-O-Fe alloys using the off-grade sponge Zr or sponge Zr-O-Fe (sponge Zr is produced by the same Kroll process). Likewise, it may also be possible to develop a new class of α - β Zr-N-Fe alloys.

In addition, oxygen embrittlement occurs not only in HCP and BCC Ti, but also in other BCC metals (e.g. Nb¹⁰ and Mo¹¹), which presents a significant metallurgical challenge. However, here we have revealed a unique distribution of oxygen in these α - β Ti-O-Fe alloys (oxygen nano-heterogeneity in the α -phase, and a near oxygen-free β -phase), which leads to high tensile ductility ($21.9 \pm 2.2\%$). Similar to the alloy design approach just described based on DFT calculations and predictions, this work may provide a template to mitigate or suppress these oxygen embrittlement issues through the introduction of an appropriate second phase.

In a broader sense, the findings of this work could further stimulate researchers to consider revitalising alloying elements including interstitial elements that are less widely used in previous alloy designs (for both non-ferrous and ferrous alloys). It offers a potential pathfinding approach to various other new alloy systems. We note in closing that the reviewer's query here was very broad. Once again, we thank the reviewer for this very enlightening question.

Changes made to the manuscript: Several brief points based on the above discussion have been added to our revised main text. Furthermore, this section has been added as Supplementary Note 6 "Potential applications and implications" and referred to in the main text.

2. Data & methodology

The alloy fabrication from elemental powders will always present doubts on chemical homogeneity. The authors use a DED approach with a relative large spot size which apparently mitigates inhomogeneous mixing. The authors point of view of this topic would be useful.

Response: Thank you for another important comment. Indeed, the use of a relatively large laser spot size (1.5 mm) is known to mitigate chemical inhomogeneities.

Fig. A7 shows half of the melt pool shape predicted for our LMD process (laser power: 500 W; laser spot size: 1.5 mm; scan speed: 800 mm/min; layer interval: 15 s). The melt pool volume is **1.08 mm³** (major axis: 2.3 mm; minor axis: 1.5 mm; depth: 0.6 mm; $T_L = 1659$ °C

for Ti-0.35O-3.0Fe), consistent with Refs. [12-14] for similar LMD conditions. Each such **melt pool** thus requires **melting of ~6,250** Ti powder particles in our study ($D_{v50} = 69.09 \mu\text{m}$).

Fig. A7 Melt pool geometry. Laser power: 500 W; laser spot size: 1.5 mm; scan speed: 800 mm/min, layer interval time: 15 s, $T_L = 1658.5 \text{ }^\circ\text{C}$ for Ti-0.35O-3.0Fe. The resulting melt pool has a volume of 1.08 mm^3 (major axis: 2.3 mm; minor axis: 1.5 mm; depth: 0.6 mm). Each such melt pool requires melting of **~6,250** Ti powder particles ($D_{v50} = 69.09 \mu\text{m}$).

Due to the high melt pool temperature (2000-3000 $^\circ\text{C}$) and the dynamic nature of the melt pool, the composition within each small melt pool can quickly become homogeneous. For powder mixes containing 2-3 different powders, if the sample size is small, e.g., only a few hundred mixed powder particles, chemical inhomogeneity can be an important issue. However, as the sample size increases to ~6000 mixed powder particles, after prolonged mixing, it may be assumed that the average composition of each such "sample" (i.e. > 6000 mixed powder particles) will be similar.

Furthermore, the melt pool depth is 0.6 mm. Each solidified layer (0.2 mm thick) is thus remelted almost three times. In addition, the ~70% overlap (remelting) between every two abutting tracks further helps to facilitate chemical homogeneity.

The experimental measurements along the build height (see **Fig. A8** below) and the very similar microstructures observed from each batch of coupons (see **Fig. A9** and **Fig. A10** below) provide direct evidence for the expected (good) chemical homogeneities.

A small batch-to-batch variation in the Fe content was observed from 3.12 wt.% to 3.36 wt.%, as listed in **Table A3**. This is much smaller than the variations allowed for commercial Ti-6Al-4V [Ti-(5.5-6.5)Al-(3.5-4.5)V] and ATI 425® [Ti-(3.5-4.5)Al-(2.0-3.0)V-(1.2-1.8)Fe]. The O content was well controlled (**Table A3**).

Table A3 Design and experimental compositions of α - β Ti-O-Fe alloys

By design (wt.%)	Measured (wt.%)
Ti-0.15O-3.0Fe	Ti-0.14O-3.23Fe
Ti-0.35O-3.0Fe	Ti-0.34O-3.25Fe
Ti-0.50O-3.0Fe	Ti-0.50O-3.17Fe
Ti-0.70O-3.0Fe	Ti-0.67O-3.30Fe
Ti-0.35O-3.0Fe	Ti-0.35O-3.32Fe
Ti-0.35O-3.0Fe	Ti-0.34O-3.25Fe
Ti-0.35O-3.0Fe	Ti-0.32O-3.12Fe
Ti-0.35O-3.0Fe	Ti-0.35O-3.36Fe
Ti-0.35O-3.0Fe	Ti-0.35O-3.17Fe
Ti-0.35O-3.0Fe	Ti-0.33O-3.11Fe

Therefore, our view is that for simple alloys such as Ti-(0.35-0.50)O-3Fe alloys, where the powders do not have drastically different densities, it is practical to fabricate them with good chemical homogeneity using the LMD process with a large laser spot size (1.5 mm). We certainly concur that for critical components, the use of pre-alloyed powder may still be preferred, in which case the potential for batch-to-batch variations in the powder composition would need to be managed.

Changes made to the manuscript: A section entitled “Chemical homogeneity of as-fabricated samples” has been added to the Methods and also as “Supplementary Note 2 Chemical homogeneity and microstructure uniformity” based on this section.

800mm/min, 0s; Ti-0.32O-3.32Fe

Fig. A8 Microstructure (left column) and distribution of Fe (right column) at different build heights of the LMD-fabricated Ti-0.32O-3.32Fe. The coupon comprises 25 layers. Scan speed: 800 mm/min; layer interval time: 0 s (see Extended Data Table 2 for other conditions). The distribution of Fe was obtained using energy dispersive spectroscopy (EDS). **(a, b)** Top layers 23-24. **(c, d)** Middle layers 12-14. **(e, f)** Bottom layers 2-4.

Fig. A9 Comparison of the microstructures of three as-fabricated Ti-0.14O-3Fe tensile specimens along the build height (LMD conditions: 800 mm/min; 15 s). The same batch of coupons have similar microstructures, consistent with the predictions based on simulation.

Fig. A10 Comparison of the microstructures of three as-fabricated Ti-0.35O-3Fe tensile specimens along the build height (800 mm/min; 0 s). The same batch of coupons have similar microstructures, consistent with the predictions based on simulation.

3. Suggested improvements

the reviewer is puzzled by the title of the manuscript. In what way additive manufacturing enables the fabrication of these ternaries' microstructures? AM encompasses a wide range of fusion techniques and the findings seems relevant to a particular process which enables the formation of near-equilibrium microstructures.

Response: Thank you. We concur with your comment and have removed the word “enabled” from the title. Our original idea focused on the avoidance of β -flecks and the production of fine α - β lamellae in fine equiaxed prior- β grains, which are indeed relevant to the specific type of the AM process studied in this work.

4. Clarity and context

the physical metallurgy of titanium alloy is dealt very well and is very accessible. The applicability of the findings, supposedly materials well versed for structural room temperature applications, remains vague. If the alloy is enabled by “additive manufacturing” what are the relevant sectors that are being targeted?

Ti-6Al-4V, the benchmark alloy, is so amenable because it offers a combination of structural and non-structural properties that are relevant to aerospace, biomedical and marine sectors. How does the new range of alloy investigated in the present compare? At least a prediction should be made to highlight the impact of the work.

Response: Thank you for this important question. The processing window and additional tensile property data established above places us in a better position to answer this question. Since the elastic moduli of our α - β Ti-O-Fe alloys are very similar to that of the α - β Ti-6Al-4V (110-120 GPa), we can simply focus on tensile ductility (ϵ_f) and strength (σ_{UTS}), detailed in **Table A4** below. They may be divided into three categories:

- with outstanding tensile ductility ($\geq 20\%$) and high strength (1,000-1,100 MPa), or
- with excellent tensile ductility (13-20%) and high strength (1,050-1,100 MPa, required for wrought α - β ATI 425®), or
- with good tensile ductility (8-12%) and high strength ($\sigma_{UTS} = 1,150$ -1,200 MPa).

These property options allow us to compare with Ti-6Al-4V or ATI 425® for potential structural room temperature applications, which are predicted below.

Table A4 Tensile mechanical properties of the LMD-fabricated Ti-O-Fe alloys.

Alloy composition (wt.%)	Scan speed (mm/min)	Layer interval (s)	σ_{UTS} (MPa)	ϵ_f (%)
Ti-0.35O-3.17Fe	600	15	1085 ± 17	14.0 ± 0.7
Ti-0.35O-3.32Fe	800	0	1034 ± 9	21.9 ± 2.2
Ti-0.34O-3.25Fe	800	15	1157 ± 3	9.0 ± 0.8
Ti-0.34O-3.25Fe	800	15	1157 ± 3	9.0 ± 0.8
Ti-0.50O-3.17Fe	800	15	1194 ± 8	9.0 ± 0.5
Ti-0.67O-3.30Fe	800	15	1271 ± 6	3.0 ± 0.8

1) Biomedical sector

Our Ti-O-Fe alloys contain no toxic elements and so have high expected biocompatibility. Therefore, we expect that they would be suitable for making implantable medical devices as well as surgical and laboratory tools. We take bone fixture plates and reconstruction plates as examples, since these are two classes of medical devices in mass production. Their typical dimensions (thickness: 1.2-3 mm, width: 8-15 mm; length: 50-150 mm or longer) are close to our coupons. They are perforated plates (screw holes or slots). Due to the lack of sufficient ductility and strength, coupled with stress concentration at the edges of these holes, fracture of the mill-annealed Ti-6Al-4V bone fixation plates occurs from time to time. Our highly ductile ($\epsilon \geq 20\%$), strong ($UTS \geq 1000$ MPa), similarly stiff ($E = 110$ GPa) Ti-0.35O-3Fe plates with improved biocompatibility manufactured in net shape by AM represents a potentially exciting enhancement and/or replacement alloy technology.

2) Aerospace, marine, defence, chemical processing, pulp and paper production (where austenitic stainless steels only last for 2~3 months) sectors

We limit our predictions to non-fatigue critical applications at this point of time. Our Ti-(0.35-0.50)O-3Fe alloys are closer to ATI 425® [Ti-4Al-2V-(1.2-1.8)Fe-(0.2-0.3)O] than to Ti-6Al-4V. ATI 425® alloy performs similarly to Ti-6Al-4V and Ti-3Al-2.5V in marine environments and many media of the chemical process industry¹⁵. We may presume that the corrosion resistance of our Ti-(0.35-0.50)O-3Fe alloys would be suitable for some general marine applications.

Due to their excellent tensile properties in the as-fabricated state, we envisage that these simple Ti-O-Fe alloys, when manufactured in net or near-net shapes by LMD, will be attractive and competitive for a broad range of room temperature structural applications that currently use Ti-

6Al-4V or ATI 425® or Ti-3Al-2.5V across various sectors. In addition, since our as-fabricated Ti-0.35O-3Fe alloy has already reached the tensile properties of cold-rolled and hot-rolled ATI 425® for ballistic applications, it may be evaluated for ballistic applications as well. Our Ti-O-Fe alloys also offer some potential cost advantages (see below).

3) A potential game changer for the utilisation of off-grade sponge Ti and high-oxygen scrap Ti

Sponge titanium production is fundamental to the entire titanium industry. The unavoidable off-grade sponge Ti from the Kroll process accounts for 10–20% of the total sponge production⁶⁻⁸, due to excess O (0.3-0.5%) and Fe (0.4-1.5%)^{6,16} contamination. An average of 10% of the sponge production is reasonable.

Sponge titanium production is extremely energy-intensive (423 GJ/ton)⁵ which is about 5-10 times the energy consumption of the primary aluminium production. The substantial off-grade sponge Ti or sponge Ti-O-Fe is currently used as the raw ferro-titanium materials for the steel industry⁶⁻⁸ (and for the fireworks industry in some cases).

Our alloys introduced here could be conceptualists of sponge Ti-O-Fe alloys. Specifically, we suggest that the alloys and process developments reported in this work provide high potential for a game-changing approach that leverages off-grade sponge Ti or sponge Ti-O-Fe materials. These off-grade sponge Ti materials could be made into powder for AM to offer high tensile ductility and high strength in net or near-net shapes for advanced applications. This transformation from off-grade sponge into high-grade titanium alloys would contribute significantly to the reduction of the carbon footprint of titanium production. We suggest that this could open up a new chapter for the sponge titanium industry.

Additionally, high-oxygen scrap CP-Ti Grade 3 and Grade 4 could be used in the same way for much higher value creation.

Changes made to the manuscript: This section has been incorporated into “Supplementary Note 6 Potential applications and implications” and has been referred to in the manuscript.

Referee #2

1. The data presented are of high quality. The authors should comment on the potential sources of variability in the stress-strain curves. While some curves are repeatable, the Ti-3.23Fe-0.14O are not, and it would be informative to note the potential sources of this variability.

Response: Thank you for this comment, which allowed us to make a detailed assessment of the repeatability issue of our tensile property data.

We use **Table X1.3** of ASTM E8/E8M – 21 (Standard Test Methods for Tension Testing of Metallic Materials) as a starting point. In this table (reproduced below), X is the mean yield strength ($\sigma_{0.2}$) in MPa, S_r is the repeatability standard deviation in MPa, and r is the **95% repeatability limit** in MPa ($r = 2.8 S_r$, by the statistical theory¹⁷). S_R and R relate to reproducibility from different operators in different labs.

In **Table X1.3**, SAE 51410 (a Cr-Mn-Si steel) has similar tensile strengths to our LMD-fabricated Ti-O-Fe alloys, with an r value of 24.8 MPa. This means that if the absolute difference in yield strength ($\sigma_{0.2}$) between any two repeatability tests is less than 24.8 MPa for the mean value of $\sigma_{0.2} = 967.5$ MPa¹⁸, then the repeatability is acceptable (repeatability tests: same operator, equipment and environment).

TABLE X1.3 Precision Statistics—0.2 % Yield Strength, MPa [ksi]

Material	X	s_r	$s_r / X, \%$	s_R	$s_R / X, \%$	r	R
EC-H19	158.4 [22.98]	3.3 [0.47]	2.06	3.3 [0.48]	2.07	9.2 [1.33]	9.2 [1.33]
2024-T351	362.9 [52.64]	5.1 [0.74]	1.41	5.4 [0.79]	1.49	14.3 [2.08]	15.2 [2.20]
ASTM A105	402.4 [58.36]	5.7 [0.83]	1.42	9.9 [1.44]	2.47	15.9 [2.31]	27.8 [4.03]
AISI 316	481.1 [69.78]	6.6 [0.95]	1.36	19.5 [2.83]	4.06	18.1 [2.63]	54.7 [7.93]
Inconel 600	268.3 [38.91]	2.5 [0.36]	0.93	5.8 [0.85]	2.17	7.0 [1.01]	16.3 [2.37]
SAE 51410	967.5 [140.33]	8.9 [1.29]	0.92	15.9 [2.30]	1.64	24.8 [3.60]	44.5 [6.45]
		Averages:	1.35		2.32		

The ratio of S_r to X typically falls within 1-2% for **quality tensile tests** (see **Table X1.3**). To put this in context, both EOS and SLM Solutions are premier industrial 3D printing technology providers worldwide. Their latest (2022) Material Data Sheet for AM Ti-6Al-4V allows the ratio of S_r to X to **be greater than 3%** (3-5%)^{19,20}.

Therefore, as an estimate, we can use $r = 2.8 \times (0.01-0.02)X$. We take $r = 2.8 \times 0.02X = 0.056X$. The stress-strain curves of our Ti-0.14O-3.23Fe alloy (**Extended Data Fig. 2a**) give $\sigma_{0.2} = 744 \pm 19$ MPa (i.e. $X = 744$ and $S_r = 19$) and $\sigma_{UTS} = 886 \pm 19$ MPa (i.e. $X = 886$ and $S_r = 19$). Therefore, we have $r(\sigma_{0.2}) = 0.056 \times 744 = 41.7$ MPa and $r(\sigma_{UTS}) = 0.056 \times 886 = 49.6$ MPa.

The maximum absolute difference in $\sigma_{0.2}$ for this alloy is 32 MPa $< r = 41.7$ MPa, while the maximum absolute difference in σ_{UTS} is 36 MPa $< r = 49.6$ MPa. **Hence, although the stress-strain curves (Extended Data Fig. 2a) appear *prima-facie* as less repeatable for this alloy, they are quality tensile property data when benchmarked against ASTM E8/E8M – 21.**

The underlying reason for the wider than might be expected discrepancy in stress-strain curves for this Ti-0.14O-3.23Fe alloy is not due to chemical inhomogeneities (see Supplementary Note 2 “Chemical homogeneity and microstructure uniformity”). To further substantiate this point, we have manufactured Ti-185 (Ti-1Al-8V-5Fe) alloy using the same LMD system with high-quality pre-alloyed Ti-185 powder containing negligible internal porosity and gas. This was purchased from a commercial supplier (<http://www.slm metal.com/en/>), where the powder was produced using a high-speed (17,000 rpm) plasma rotating electrode process (PREP). However, a similar discrepancy in tensile stress-strain curves was observed, see **Fig. B1** below.

Fig. B1 Tensile stress-strain curves of Ti-1Al-8V-5Fe (wt.%) fabricated using high-quality pre-alloyed PREP spherical powder and the same LMD system used for Ti-0.14O-3.23Fe. A larger than expected discrepancy in stress-strain curves was observed, but all the curves are still within the repeatability limit, i.e. they represent quality tensile tests.

Therefore, we attribute the variation in the tensile properties of the Ti-0.14O-3.23Fe alloy **mainly to its inhomogeneous and coarse grain structure due to its low O content**, since O promotes the columnar-to-equiaxed transition in these alloys. As observed from our main text

Fig. 1D-1G, the Ti-0.14O-3.23Fe alloy has the coarsest, most irregular prior- β grains (main text **Fig. 1D**) of all the Ti-(0.14-0.67)O-3Fe alloys fabricated in our study. This leads to a wider-than-expected discrepancy in tensile properties for this alloy. No lack-of-fusion defects were observed on the fracture surface of this alloy. The build quality was consistent and high. The residual stress was considered similar for all samples.

Changes made to the manuscript: A concise section entitled “Repeatability of tensile stress-strain curves” has been added to the **Methods**. This section has been added as “Supplementary Note 7 Repeatability of tensile stress-strain curves”.

2. The orientation of the extracted mechanical test specimens should be specified.

Response: **Fig. B2** below has now been added as **Supplementary Fig. 5** to display the orientation of each extracted specimen and mentioned in the Methods. The track lines on each coupon surface in **Fig. B2a** show the scan paths for the last layer deposition. All coupons were built parallel to the substrate at equal intervals.

Fig. B2 As-fabricated coupons by laser metal deposition (LMD) for tensile testing. (a) As-built rectangular coupons of 40 mm \times 10 mm \times 5 mm at the layer thickness of 200 μ m. (b) Schematic extraction of tensile specimens (12 mm \times 3 mm \times 2 mm as per Australian Standard AS 1391-2007) from as-built coupons. Each tensile specimen was extracted from the middle nine layers of the coupon. The temperature variation was within 30 $^{\circ}$ C across these nine layers by simulation, ensuring consistent microstructures. At least five coupons were printed for each condition.

3. In directed energy deposition, the microstructure changes from near the baseplate to further up in the build, with this transient existing several mm into the height of the build. Were the samples all extracted from the same height from analogous builds? Or what method was used to check that the samples had similar microstructures?

Response: Thank you for this important question.

Sample extraction position: Yes, all tensile samples were extracted from the middle nine layers of each coupon, as shown in **Fig. B2b** above, to ensure similar microstructures based on detailed simulations and experimental validation. The reason is elucidated below.

Methods to check similar microstructures

(i) Detailed simulation using *Simufact welding* (high-fidelity simulations)^{4,21}. **Fig. B3** (next page) shows the evolution of the average temperature at the central points of layers 1, 8, 13 and 18 in a 25-layer build (coupon) with various LMD parameters. Each data point in **Fig. B3** was averaged from about 2000 to 2500 temperature signals based on the layer scanning time, layer interval time and time resolution (0.02 s). Temperature stabilisation occurred from the 8th or 9th layer of deposition. Although the average temperature increased with increasing build height, the increase across every 8 layers after layer 9 was limited to ~30 °C (insignificant for microstructural evolution). **This is the main reason for extracting each tensile sample from the middle nine layers.**

(ii) Microstructural observations. We have examined the microstructures for each composition or each fabrication condition. **Fig. B4** and **Fig. B5** below display the microstructures of *six tensile samples* from the bottom to the top. Similar microstructures were confirmed for the same composition and fabrication conditions, consistent with predictions by simulations.

Changes made to the manuscript: A paragraph entitled “Extraction of tensile specimens and microstructural uniformity” has been added to the **Methods**.

Fig. B3 Temperature evolution in coupons during laser metal deposition by simulation. All samples comprise 25 layers. The simulation focuses on the evolution of the average temperature at the central points of layers 1, 8, 13 and 18 in each coupon with layer interval time. Stabilisation occurs from about the 9th layer of deposition.

Fig. B4 Comparison of the microstructures of three as-fabricated Ti-0.14O-3Fe tensile specimens along the build height (scan speed: 800 mm/min; layer interval: 15 s). The same batch of coupons have similar microstructures, consistent with predictions by simulation.

Fig. B5 Comparison of the microstructures of three as-fabricated Ti-0.35O-3Fe tensile specimens along the build height (scan speed: 800 mm/min; layer interval: 0 s). The same batch of coupons have similar microstructures, consistent with predictions by simulation.

4.1 For the alloys examined, it is implied that the additive manufacturing processing was advantageous for fabricating the observed microstructure and achieving the measured properties. However, no discussion is provided on the selection of processing parameters.

The authors did simulate the additive manufacturing process, but indicated in the supplemental document that, “Based on simulations and our experiences, the LMD processing parameters for alloy fabrication were chosen to be laser power of 500 W, laser spot size of 1.5 mm, scan speed of 800 mm/min, layer thickness of 0.2 mm, and scan spacing of 1.05 mm.” The authors should detail what metrics were used to determine these parameters. Were they bead shape, reduced thermal cycling, or some critical cooling rate?

Response: Thank you for these questions. Indeed, there were significant efforts behind these selections. Our goal was to obtain (i) fast and consistent high-quality fusion (minimal lack-of-fusion defects and keyhole pores), and (ii) fine α - β lamellae through decomposition of the α' -martensite after solidification. **Two metrics** were used to co-determine them, which are elaborated below.

Metrics I: Fast and consistent high-quality fusion, including chemical homogeneity

As the expert reviewer is no doubt aware, there are multiple parameters that can be selected for quality deposition. The energy density E_d is defined as $E_d = P/(d \times v)$, where P is laser power, d is laser spot size, and v is laser scan speed. The range of E_d (J/mm^2) for obtaining high-quality LMD of Ti-6Al-4V has been experimentally determined to be 13-36 J/mm^2 by other researchers²², consistent with our experiences (optimum range of E_d : **25-35** J/mm^2).

In our experiments, a laser spot size of 1.5 mm was chosen to ensure that the melt pool volume was sufficiently large (≥ 1.0 mm^3), for two reasons. **Fig. B6** displays half of a melt pool produced with the laser spot size of 1.5 mm, laser power of 500 W, scan speed of 800 mm/min, layer thickness of 200 μm and layer interval of 15 s. The volume of this semi-ellipsoidal melt pool is 1.08 mm^3 (major axis: 2.3 mm; minor axis: 1.5 mm; depth: 0.6 mm), where the Ti-0.35O-3.0Fe alloy has a liquidus temperature of 1659 $^\circ C$.

This melt pool size (1.08 mm^3) is consistent with the literature data under similar LMD conditions⁸⁻¹⁰. It requires **melting of ~6,250** Ti particles ($D_{v50} = 69.09$ μm). This helps to ensure good chemical homogeneity and quality deposition. Furthermore, with a 70% overlap used in this work between two abutting melt pools, such a melt pool size helps achieve a more stabilised temperature profile (**Fig. B3**), which is important for the decomposition of the α' -martensite.

Fig. B6 Melt pool geometry and volume. Laser power: 500 W; laser spot size: 1.5 mm; scan speed: 800 mm/min, layer interval time: 15 s. The liquidus temperature of Ti-0.35O-3.0Fe is 1658.5 $^\circ C$. The resulting semi-ellipsoidal melt pool has a volume of 1.08 mm^3 Each melt pool requires melting of **~6,250** Ti powder particles with $D_{v50} = 69.09$ μm .

Apart from the laser spot size d , a variety of P/v combinations can satisfy $E_d = 25\text{-}35 \text{ J/mm}^2$ through $E_d = P/(d \times v)$. The laser scan speed is typically selected in the range of 400-1200 mm/min. Another important consideration to ensure sufficient fusion is that of the laser dwell time, defined as d/v . With $E_d = 25\text{-}35 \text{ J/mm}^2$ for Ti alloys, our experience is that d/v should be around 0.1 s. Based on these considerations, our preferred scan speed was 800 mm/min with a laser power of 500 W, leading to $E_d = 25 \text{ J/mm}^2$. The scan speed of 600 mm/min ($E_d = 33.3 \text{ J/mm}^2$) or 1000 mm/min ($E_d = 20 \text{ J/mm}^2$) can also be considered for $P = 500 \text{ W}$. Further increasing the scan speed to 1200 mm/min would require an accompanying increase in laser power.

The layer thickness of 200 μm and the powder flow rate of 1.7 g/min were determined together through various preliminary trials, focusing on ensuring (i) a consistent weld bead with an aspect ratio of around 3-4 to 1, (ii) a nearly lack-of-fusion-free build (through micro-CT and fracture surface examination), and (iii) a consistent microstructure.

The scan spacing of 1.05 mm was selected based on the simulated melt pool minor axis of 1.5 mm in order to ensure a linear overlap of 70% for sufficient fusion between two abutting tracks.

Metrics II: Obtaining fine α - β lamellae dispersed in fine equiaxed prior- β grains after solidification

Our above-selected LMD processing parameters must also meet the following two criteria in order to obtain fine α - β lamellae through the in-situ decomposition of the α' -martensite:

- (i) an appropriate cooling rate ($\geq 400 \text{ }^\circ\text{C/s}$) when the β -phase in each solidified layer is cooled from the single β -phase region for the formation of α' , and
- (ii) an appropriate stabilised temperature window across each layer (e.g. $\geq 480 \text{ }^\circ\text{C}$, preferably from 600 to 750 $^\circ\text{C}$) to facilitate the $\alpha' \rightarrow \alpha$ - β decomposition (excluding the last few layers).

If the selected parameters failed to satisfy these criteria, re-selection by simulation was necessary until they did. **Table B1** summarises the predicted average cooling rates at the centres of layers 1, 13 and 25 in a 25-layer coupon during cooling from the liquid state to 800 $^\circ\text{C}$ (the assumed M_s temperature) for various processing parameters (scan speed: 200-1200 mm/min; layer interval: 0-120 s). They all satisfy the required cooling condition to form α' -martensite.

Table B1 Average cooling rate at the centres of layers 1, 13 and 25 in a 25-layer coupon during cooling from the liquid state to 800 °C

Average cooling rate when cooled from the liquid state to 800 °C			
Manufacturing condition (scan speed – layer interval)	Dimensions of the 25-layer coupon (length × width × thickness)	Sample position	Average Cooling rate
1200 mm/min – 15 s	40 mm × 10 mm × 5 mm	Centre of layer 1	3,868 °C/s
800 mm/min – 0 s	40 mm × 10 mm × 5 mm	Centre of layer 1	2,356 °C/s
		Centre of layer 13	818 °C/s
		Centre of layer 25	646 °C/s
800 mm/min – 15 s	40 mm × 10 mm × 5 mm	Centre of layer 1	2,432 °C/s
		Centre of layer 13	1,444 °C/s
		Centre of layer 25	1,404 °C/s
800 mm/min – 60 s	40 mm × 10 mm × 5 mm	Centre of layer 1	2,881 °C/s
		Centre of layer 13	2,224 °C/s
		Centre of layer 25	2,401 °C/s
800 mm/min – 120 s	40 mm × 10 mm × 5 mm	Centre of layer 1	3,131 °C/s
		Centre of layer 13	2,619 °C/s
		Centre of layer 25	2,739 °C/s
600 mm/min – 15 s	40 mm × 10 mm × 5 mm	Centre of layer 1	1,855 °C/s
		Centre of layer 13	1,016 °C/s
		Centre of layer 25	1,008 °C/s
400 mm/min – 15 s	40 mm × 10 mm × 5 mm	Centre of layer 1	1,116 °C/s
200 mm/min – 15 s	40 mm × 10 mm × 5 mm	Centre of layer 1	450 °C/s
Cu-mould casting	Wall thickness: 5 mm	Centre of sample	30 °C/s ²³

On the other hand, our systematic simulations, summarised in main text **Fig. 1C** (reproduced as **Fig. B7** below in our response to the next comment), have indicated that the parameters listed in **Table B1** can lead to a wide stabilised temperature window from 480 °C to 800 °C in the build. Therefore, they satisfy the second criterion above. Please refer to **Fig. B7** below.

Changes made to the manuscript: A long section titled “Selection of LMD parameters and the required thermal history bounds or processing window” has been added to the **Methods**.

4.2 What thermal history bounds are required for achieving the observed microstructures? Was the optimization based on creating dense samples or achieving some optimal temperature profile?

Response: Thank you for these two critical questions, which have encouraged us to assess in detail the required thermal history bounds.

As just described, our goal was to obtain (i) fast and consistent high-quality fusion (with minimal lack-of-fusion defects and keyhole pores), and (ii) fine α - β lamellae through decomposition of the α' after solidification. The two metrics elaborated above were also used to determine the required thermal history bounds for achieving our goal.

Earlier, **Fig. B3** showed the evolution of the average temperature at the central points of layers 1, 8, 13, and 18 (all coupons consist of 25 layers). We now focus on the central point of layer 1 since this position will reveal the information from the deposition of all subsequent 24 layers (25-layer coupons). **Fig. B7** shows the evolution of the average temperature at the centre of layer 1 for all the parameters considered in **Table B1** (scan speed: 200-1200 mm/min, layer interval time: 0-120 s —the most practical range of the LMD operations).

Fig. B7 Required thermal history bounds (the green zone), predicted by simulations. Each curve in the plot shows the evolution of the average temperature at the centre of layer 1 in a 25-layer coupon with respect to different combinations of laser scan speed (200-1200 mm/min) and layer-to-layer interval time (0-120 s). See **Extended Data Table 2** for other parameters. Temperature stabilisation occurs from the 8th or 9th layer of deposition.

In general, decomposition of the α' -martensite commences at ~ 400 °C (long holding is needed at this temperature)², while complete decomposition of the α' usually requires isothermal exposure to 800 °C for ≥ 2 h, depending on wall thickness^{2,3}. However, the

effect of the thermal pulses (**Fig. 1B** – main text) generated during LMD can markedly shorten this decomposition process⁴.

On the other hand, there is a risk that the microstructure could experience severe coarsening at ≥ 800 °C, while below 480 °C (the minimum stress relief temperature), decomposition of the α' may not be complete. In addition, as shown earlier in **Fig. B3**, the stabilised temperature increases with increasing build height, which means that the stabilised temperature of the first layer should not be set too high. Based on these considerations, we chose 480-800 °C – **the green zone** in **Fig. B7** as our required thermal history bounds for experimental validation.

To validate the thermal history bounds delimited in **Fig. B7** (**green zone**), we fabricated tensile coupons of the Ti-0.35O-3Fe alloy both within and outside this green zone and tested their tensile properties. The results are summarised in **Fig. B8** below, where the horizontal broken line divides the tensile properties into two regions: outside (above) and within (below) the thermal history bounds or processing window (the green zone in **Fig. B7**). These results fully support our simulation-informed thermal history bounds and design concept.

Fig. B8 Engineering stress-strain curves of the as-fabricated Ti-0.35O-3Fe alloy. The required thermal history bounds delimited in **Fig. B7** (green zone) are validated and established.

Within the required thermal history bounds, the as-fabricated Ti-0.35O-3Fe alloy achieved excellent combinations of tensile ductility (ϵ_f) and ultimate tensile strength (σ_{UTS}):

$\epsilon_f = 21.9 \pm 2.2\%$ and $\sigma_{UTS} = 1034 \pm 9$ MPa with 800 mm/min and 0 s

$\epsilon_f = 14.0 \pm 0.7\%$ and $\sigma_{UTS} = 1085 \pm 17$ MPa with 600 mm/min and 15 s

$\epsilon_f = 9.0 \pm 0.8\%$ and $\sigma_{UTS} = 1157 \pm 3$ MPa with 800 mm/min and 15 s

In contrast, outside the required thermal history bounds, the tensile ductility dropped sharply with very high σ_{UTS} , resulting in unbalanced and undesired combinations.

$\epsilon_f = 2.2 \pm 0.6$ with $\sigma_{UTS} = 1303 \pm 18$ MPa with 800 mm/min and 120 s

$\epsilon_f = 3.2 \pm 0.3$ with $\sigma_{UTS} = 1268 \pm 10$ MPa with 800 mm/min and 60 s

We have thus **established** the required thermal history bounds or processing window, which can accommodate numerous combinations of LMD parameters, as long as the resulting stabilised temperature falls into this large green zone in **Fig. B7**. This offers high flexibility.

The influence of the wall thickness, coupon shape, or build height can be easily predicted nowadays using high-fidelity simulations prior to the manufacture so that the LMD processing parameters can be adjusted to generate a consistent microstructure throughout the build. To this end, it is important to have flexible thermal history bounds.

Changes made to the manuscript: A section entitled “Selection of LMD parameters and the required thermal history bounds or processing window” has been added to the **Methods** based on this discussion.

5. The team should comment on the applicability of other manufacturing processes for this alloy. Is additive manufacturing expected to be the only or best method for these alloys? Why (in terms of thermal history/processing steps)?

Response: Thank you. Yes, we conclude that fusion-based metal AM is the best net-shape manufacturing method for these alloys that can ensure significant tensile properties. In the following, we first comment on bulk solidification-based manufacturing processes and then on non-solidification-based processes.

1) Bulk solidification-based manufacturing processes

To address this comment, we prepared small ingot samples (120 mm × 12 mm × 5 mm – length × width × thickness) of the Ti-0.35O-3.0Fe alloy (measured: Ti-0.33O-3.11Fe), using a vacuum arc melter, followed by water-cooled Cu-mould casting (see **Methods**).

Fig. B9 shows the resulting as-cast microstructure (lamellar α - β type). A significant presence of the Fe-stabilized β -flecks containing 5.6%Fe was observed at the number density of ~ 400 β -flecks/cm², **Fig. B9 a-d**, despite the use of water-cooled Cu-moulds and the small sample size (32 g/sample). This demonstrates the severity of the β -fleck formation issue in these simple alloys when prepared using ingot metallurgy.

β -flecks can reduce the tensile ductility **by > 50%** (from 10-13% to 5-6%) and the low-cycle fatigue strength **by > 90%** in the case of Ti-1023 (Ti-10V-2Fe-3Al)¹. They are difficult to eliminate in large ingots. Therefore, we may conclude that bulk solidification-based conventional manufacturing techniques are not suitable for these Ti-O-Fe alloys. **In contrast**, β -flecks were completely absent in all our LMD-fabricated Ti-O-Fe alloys, even a higher Fe content.

Fig. B9 Microstructure of the as-cast reference alloy Ti-0.35O-3.0Fe (measured: Ti-0.33O-3.11Fe). **(a)** An overview of the microstructure comprising α - β lamellae and β -flecks. **(b)** A closer view of a selected β -fleck. **(c)** EDS mapping of Fe in (b) to confirm the Fe-enriched β -fleck. **(d)** A selected β -fleck phase for EDS line scan analysis. Also shown is the as-cast porosity. **(e)** EDS line scan further confirms the high Fe content in the β -fleck shown in (d). **(f)** A BSE image showing the α - β lamellae in this as-cast alloy.

Fig. B10 compares the tensile properties of the as-cast and LMD-fabricated Ti-0.35O-3.0Fe alloy. The tensile ductility of the as-cast alloy is more than 50% lower for similar tensile strength.

Fig. B10 Comparison of tensile properties of water-cooled copper-mould cast Ti-0.33O-3.11Fe (Ti-0.35O-3Fe by design) alloy and LMD-fabricated Ti-0.33O-3.11Fe alloy with two sets of LMD conditions. The as-cast Ti-0.33O-3.11Fe is much less ductile (50-70% lower) than the LMD-fabricated alloy for similar tensile strengths.

2) Non-solidification-based manufacturing processes

Sintering-based powder metallurgy (PM) processes, including (i) conventional powder metallurgy (C-PM: press and sinter), (ii) metal injection moulding (MIM, small parts, ≤ 65 mm in most cases) and (iii) hot isostatic pressing (HIP) all represent examples of processing technologies that **can also avoid** β -fleck formation and produce lamellar α - β microstructures, close to equilibrium. However, C-PM and MIM usually leads to high residual porosity (~ 2 vol.%, pressureless sintering), coarse β grains (1200-1300 °C for 1-4 hours), and coarse α - β lamellae (slow furnace cooling). They are not expected to exhibit high tensile properties. HIP is a possible option but is not comparable to AM for net shape formation. In addition, our HIP experiences with Ti-6Al-4V indicate that even with a low HIP temperature (820 °C) and a high

HIP pressure (200 MPa), HIP still yields thick α -laths ($\sim 2 \mu\text{m}$ thick), resulting in much lower tensile strengths than those available in LMD-fabricated materials.

Without considering net-shape manufacturing (high buy-to-fly ratio), we envisage that our α - β Ti-O-Fe alloys can be manufactured with excellent tensile properties through the combination of PM, elaborate thermo-mechanical processing and machining.

For example, ingots of Ti-O-Fe alloys can be made from hydride-dihydride (HDH) **high-oxygen** Ti powder (**low value**) with Fe powder by cold isostatic pressing and sintering. Then, through β -field forging + multi-axial α - β field forging + extrusion/rolling, the pre-sintered Ti-O-Fe alloy ingots could be manufactured into billets or plates with fine equiaxed β grains (20-40 μm – much finer than that as-deposited here, main text **Fig. 1D-1G**). Finally, heat treatments could be explored to produce fine α - β lamellae within the fine equiaxed prior β grains. Such Ti-O-Fe microstructures are expected to possess attractive tensile properties. Machining will be the ultimate net-shape formation step. The allowed use of high-oxygen HDH Ti powders based on this work could be a significant advantage.

Therefore, our conclusion is that AM remains the essential method of choice for the net-shape manufacture of these alloys, with outstanding tensile properties.

Changes made to the manuscript: “Supplementary Note 3 Advantages of AM in producing the designed Ti-O-Fe alloys” and “Supplementary Note 6 Potential applications and implications” have covered the discussion of this section.

6. The authors talk about the absence of beta-flecks. For the reader, it would be good to mention the expected size of these and concentration in the related alloys.

Response: Based on the literature data, the width of the β -flecks in large Ti alloy ingots is estimated to be **1-4.4 mm**²⁴ (based on observations from billet samples by assuming a uniform reduction during billet conversion), while the length can be up to **a few centimetres**. They mainly form in the late-stage solidification when the solid fraction is greater than 0.8²⁵. The β -flecks observed in our copper-mould-cast 5-mm thick Ti-0.33O-3.11Fe sample were 50-150 μm long and 30-60 μm wide (**Fig. B9**). We have added (**up to centimetres in size, Supplementary Note 1**) to our revised main text.

7. On page 6-7, the authors state, “The strengthening potency of the Fe is inferred by comparing the ultimate tensile strength (UTS) values of Ti-3.23Fe-0.14O and CP Ti-0.062Fe 0.16O, demonstrating ~105 MPa/1.0wt.% Fe, close to the reported 75 MPa/1.0wt.% Fe.” Is the “75 MPa/1.0wt.% Fe” for pure Ti? Is this a theoretical or experimental result that is referenced in 31?

Response: Thank you for these questions. Based on the notes below from Ref. [26] (Ref. [1] in the main text), we can now confirm that the quoted value is an experimental result.

Page 15, Ref. [26]

“Table 4 presents the physical and mechanical properties of a solid α -solution, as determined during the tension of annealed titanium alloys containing alloying elements within the range of solubility in α -titanium. The properties of a solid β -solution were determined on alloys quenched from the β -region and having a mechanically stable β -phase, i.e., the β -phase which does not undergo any transformations distorting the true properties of the phase during the tension test.”

Changes made to the manuscript: we have added “experimental value of” and “experimental range” to clarify them in the revised manuscript.

8. More details should be added on the thermodynamic calculations the team used. The CALPHAD method was mentioned several times, and only in its 4th mention (page 10 in the main document, and in the supplemental document) is it stated that equilibrium calculations were used. As additive manufacturing is a far-from-equilibrium process, the authors should discuss the applicability and limitations of these equilibrium calculations.

Response: The use of the CALPHAD method in this work was to approximate the isothermal holding stage, e.g. the predicted stabilised temperature (650 °C or 800 °C) in **Fig. B7** or main text **Fig. 1C**. The isothermal holding time at each predicted stabilised temperature is up to ~**14 min** in our build. It was calculated as follows: the deposition time for each odd-number layer (e.g. 35.85 s at 600 mm/min) + the deposition time for each even-number layer (e.g. 37.71 s at 600 mm/min) + each layer-to-layer interval time (15 s). Therefore, the CALPHAD approach was not used to predict the far-from-equilibrium cooling process. Its use was limited to the approximate isothermal holding stage, and the predictions were used to help understand the microstructural evolution during this isothermal holding stage. We have added a section “The CALPHAD method” to the **Methods** to clarify this point and mentioned this in the revised manuscript.

9. The authors state that they have a large volume fraction of “virtually O-free continuous beta phase” – but how does this compare to the O in the beta phase in Ti-6Al-4V with oxygen for example?

Response: To address this comment, we first conducted similar Atom Probe Tomography (APT) characterisation of the Ti-6Al-4V-0.22O-0.20Fe alloy sample fabricated under **the same** LMD condition as our Ti-O-Fe alloys.

Fig. B11 displays the APT 3D reconstruction data for the distribution of Ti, Al, V, Fe and O in an α - β - α region in this Ti-6Al-4V-0.22O-0.20Fe alloy. There is a significant presence of O in the β -phase in this LMD-fabricated Ti-6Al-4V: **the O content in the β -phase is about half of the O content in the α -phase**, fundamentally different from the β -phase in our as-fabricated Ti-O-Fe alloys.

Fig. B11 APT 3D reconstruction data for the distribution of Ti, Al, V, Fe and O in an α - β - α region from the Ti-6Al-4V-0.22O-0.20Fe alloy, which was fabricated with the same LMD parameters (800 mm/min-15s) used for the Ti-0.35O-3Fe alloy. The two black isoconcentration surfaces of 10 at.% V represent the α - β phase boundaries. The downward vertical arrow in (a) indicates the direction of the 25-nm long range for the APT profiles shown in the right column. The Fe- and V-enriched region (between the two black isoconcentration surfaces) was the β -phase, which lies between two α phases. Refer to **Supplementary Table 4** for the quantitative results obtained from this APT analysis.

To further clarify the partitioning of O in the α and β phases in Ti-6Al-4V, we conducted a detailed literature review. The data are summarised in **Table B2** (next page). Our APT analysis is **in good agreement with the literature data**, which revealed that **it is common for the β -phase in Ti-6Al-4V to contain 40-80% of the O level in the α -phase, irrespective of manufacturing method.**

Fig. B12a plots the ratio of the O content in the β -phase ($C_{\beta-O}$) to the O content in the α -phase ($C_{\alpha-O}$) in Ti-6Al-4V versus manufacturing method. Also plotted are the ratios of $C_{\beta-O}/C_{\alpha-O}$ for our LMD-fabricated Ti-0.34O-3.25Fe and Ti-0.14O-3.23Fe alloys. The difference is striking and is fundamental to our explanation of the significant tensile strength-ductility combinations observed with our Ti-O-Fe alloys (see discussion below on Deformation Mechanisms).

Similarly, **Fig. B12b** plots the ratio of the Fe content in the β -phase ($C_{\beta-Fe}$) to the Fe content in the α -phase ($C_{\alpha-Fe}$) in Ti-6Al-4V and our Ti-O-Fe alloys versus manufacturing methods. The ratio of $C_{\beta-Fe}/C_{\alpha-Fe}$ in our Ti-O-Fe alloys is about **an order of magnitude higher** than that in Ti-6Al-4V. This may be regarded as another feature of the β -phase in our Ti-O-Fe alloys.

Changes made to the manuscript: we have added “Supplementary Note 4 Oxygen and iron in LMD-fabricated Ti-O-Fe and Ti-6Al-4V alloys” and referred to it in the revised manuscript

Fig. B12 APT data on the partitioning of O and Fe in α - β Ti-6Al-4V fabricated by different processes versus in two α - β Ti-O-Fe alloys fabricated by LMD in this work. **(a)** O and **(b)** Fe. The two blue squares in each plot are for the reference Ti-6Al-4V alloy fabricated in this work by LMD. Data source: **Table B2**. $C_{\alpha-O}$: O in α ; $C_{\beta-O}$: O in β ; $C_{\alpha-Fe}$: Fe in α ; $C_{\beta-Fe}$: Fe in β .

Table B2 APT results on the distribution of O and Fe in the α and β phases of Ti-6Al-4V manufactured by different methods ($C_{\alpha-O}$: O in α ; $C_{\beta-O}$: O in β ; $C_{\alpha-Fe}$: Fe in α ; $C_{\beta-Fe}$: Fe in β)

Manufacture method	$C_{\alpha-O}$ (at.%)	$C_{\beta-O}$ (at.%)	$C_{\beta-O}/$ $C_{\alpha-O}$	$C_{\alpha-Fe}$ (at.%)	$C_{\beta-Fe}$ (at.%)	$C_{\beta-Fe}/$ $C_{\alpha-Fe}$	Ref.
Laser powder bed fusion (LPBF)	0.5	0.4	0.8	0.2	2.5	12.5	27
	0.9	0.3	0.33	0.2	3.3	16.5	
			~0.2			~70	28
			~0.4				
	0.5	0.09	0.18	0.05	2.8	56.0	29
Electron beam powder bed fusion (EB-PBF)	0.41	0.24	0.59	0.16	3.28	20.0	30
	0.78	0.23	0.29				31
	0.41	0.24	0.58	0.16	3.28	20.5	32
	0.42	0.20	0.48	0.16	3.34	20.9	
LPBF + heat treatment at 400 °C	0.4	0.1	0.25	0.1	5.0	50.0	27
LPBF + heat treatment at 530 °C	0.7	0.2	0.29	0.1	7.7	77.0	
Laser metal deposition (LMD)	1.05	0.74	0.70				31
Thermo-mechanical reversals between 400-650 °C – 45 cycles	0.4	0.1	0.25	0.1	2.0	20.0	29
Thermo-mechanical reversals between 400-650 °C – 75 cycles	0.4	0.1	0.25	0.06	1.7	28.3	
Arc-melting + beta forging + alpha-beta forging + ageing	0.29 (wt.%)	0.11 (wt.%)	0.38	0.06 (wt.%)	3.91 (wt.%)	65.2	33
LMD (Ti-6Al-4V)	0.56	0.27	0.482	0.08	1.97	24.63	This study
	0.64	0.27	0.422	0.09	1.97	21.89	
LMD (Ti-0.14O-3Fe)	0.9619	0.0659	0.068	0.0611	10.790	176.6	
LMD (Ti-0.34O-3Fe)	1.3895	0.0314	0.023	0.0211	11.355	538.13	

10. A key limitation of this study is that while an alloy is presented, the authors do not emphasize the mechanisms for the strength and ductility. They do provide a list of potential reasons for the observed properties in their summary paragraph, but these are not discussed at length in the body of the document, and they do not dive deeply into investigating these **deformation mechanisms** to identify what about the microstructure or chemistry results in the observed properties. For example, the authors could perform **control studies to vary the microstructure, but not chemistry**, to see how changing the size of the microstructural features impacts the properties.

Response: Thank you for this comment and suggestion. In the following, we first focus on the suggested “control studies”. Then we discuss our TEM results and other ‘deep dive’ efforts to elucidate the deformation mechanisms. This is a long section. We divide it into sections.

10.1 Control studies to vary the microstructure, but not chemistry

The suggested control studies helped us to identify the required thermal history bounds or establish the required processing window. Most of the results have been presented and discussed in the preceding sections. Here, we briefly recapitulate the main points.

We chose to focus on the Ti-0.35O-3Fe alloy (**fixed chemistry**) for our control studies to identify how changing the size of the microstructural features impacts the properties. The experiments we conducted include systematic changes in cooling rate (**Table B1**) through changing LMD conditions, both within and outside thermal history bounds in **Fig. B7**. The experimental results have been shown in **Fig. B8**. To facilitate the present discussion, we reproduce **Fig. B3** and **Fig. B8** below as **Fig. B13(a, b)**.

As shown in **Fig. B13b**, the resulting tensile properties of the same Ti-0.35O-3Fe alloy fabricated within and outside the required thermal bounds are drastically different. The representative microstructure of each sample is shown in **Fig. B14(a-e)**. Based on the scan speed and layer interval time or the predicted cooling rate at the central point of layer 13 listed in **Table B1**, it is easy to conclude that the processing condition of 800 mm/min – 120 s will lead to the finest microstructure (**Fig. B14d**), followed by the conditions of 800 mm/min – 60 s (**Fig. B14c**); 800 mm/min – 15 s (**Fig. B14b**), 600 mm/min – 15 s (**Fig. B14e**), and 800 mm/min – 0 s (**Fig. B14a**). The microstructures shown in **Fig. 14(a-e)** confirmed this trend. The decrease in tensile strength from 1303 ± 18 to 1034 ± 9 MPa and the increase in tensile ductility from 2.2 ± 0.6 % to 21.9 ± 2.2 % follow the same sequence. The strength-ductility trade-off rule is still followed (i.e. increasing strength leads to ductility loss).

Therefore, the conclusion is clear from these control studies and those shown in the main text **Fig. 1D-K** and **Fig. 2** — **it is the combination** of the composition and the AM process that results in the success of these alloys. This combination leads to the unique integration of multi-scale (micro-nano-atomic) microstructural features that are responsible for the outstanding tensile properties obtained.

The deformation mechanisms will be discussed in the next section.

Changes made to the manuscript: We have emphasised in the revised manuscript that it is the *combination* of the composition and the AM process that results in the attractive tensile properties of these Ti-O-Fe alloys. Also, a section entitled “Selection of LMD parameters and the required thermal history bounds or processing window” has been added to the **Methods**.

Fig. B13 Engineering stress-strain curves of the as-fabricated Ti-0.35O-3Fe alloy for control studies to vary the microstructure, but not chemistry, through changing the LMD processing conditions, both within and outside the required thermal history bounds in (a) – the green zone.

Fig. B14 Control studies that vary the microstructure of the Ti-0.35O-3Fe alloy by varying the manufacturing conditions within and outside the required thermal history bounds delimited in **Fig. B13a** (the green zone). (a-e) LMD with different scan speeds and layer interval times as indicated in each micrograph.

10.2 Deformation mechanisms

We sought to elucidate the deformation mechanisms based on experimental observations obtained from (i) the HAADF-STEM analysis (HAADF: high-angle angular dark field; STEM: scanning transmission electron microscopy); (ii) integrated differential phase contrast (iDPC) STEM analysis; (iii) transmission electron microscopy (TEM) analysis; and (iv) fractography analysis by scanning electron microscopy (SEM).

10.2.1 The original HAADF-STEM and iDPC-STEM analysis

We first discuss in more detail the results presented in the main text **Fig. 3E-F**, reproduced below. In **Fig. 3E**, a dislocation core was imaged, delineated by the red-blue intersection in the green inset. This inset reveals the strain condition at the defect (dislocation) and the surrounding α -phase lattice by Geographic Phase Analysis (GPA—a digital signal processing method used to analyse the strain from high-resolution STEM images). The right-hand-side enlarged HAADF-STEM image reveals the strong presence of O interstitials (marked as green spheres) in the dislocation, similar to Cottrell atmospheres in BCC crystals³⁴. They impede dislocation movement, which strengthens the α -phase but also makes the α -phase less ductile.

No oxygen atmospheres were observed in the β -phase, which is virtually free of O. The distribution of the Fe atoms was uneven in the β -phase, **Fig. 3F**, where bright spots correspond to more Fe atoms in the local β -Ti lattice. This uneven distribution of the Fe atoms is, however, not a strong impediment to dislocation movement. This is corroborated by the TEM observations shown below.

Fig. 3 (E) A HAADF-STEM image of a dislocation inhibited by an O interstitial array. Geographic Phase Analysis (top-left corner inset) shows the strain condition of the defect and the surrounding region. The dislocation core is defined by the red-blue intersection in the inset. The iDPC-STEM image (bottom-right corner inset) shows the O interstitial array. The right-hand enlarged HAADF-STEM image reveals the strong presence of O interstitials (extracted from the iDPC image and marked as green balls) in the dislocation, impeding dislocation movement. **(F)** A HAADF-STEM image of a β phase region, highlighting the non-uniform distribution of Fe, revealed by the uneven Z-contrast, where the zone axis is $[110]_{\beta}$. Bright contrast represents heavy elements (more Fe atoms in the local β -Ti lattice) while dark contrast corresponds to light elements (fewer Fe atoms).

10.2.2 TEM analysis and fractography analysis

To gain more insight into the potential deformation mechanisms, we performed detailed TEM analyses of several areas near the tensile fracture surfaces of Ti-0.34O-3.25Fe ($\epsilon = 9.0 \pm 0.8$, UTS = 1157 ± 3) and Ti-0.67O-3.30Fe ($\epsilon = 3.0 \pm 0.8\%$, UTS = 1271 ± 6). The observations are shown below:

Ti-0.34O-3.25Fe: **Fig. B15 a-c** focusing on the α -phase

Fig. B15 d-f focusing on the β -phase

Ti-0.67O-3.30Fe: **Fig. B16 a-c** focusing on the α -phase

Fig. B16 d-f focusing on the β -phase.

Dislocation multiplication in the vicinity of the tensile crack tips plays a vital role in determining whether the materials such as Ti alloys exhibit brittle or ductile behaviour during fracture³⁵. This is because sufficient dislocation activity at the tip of a tensile crack (e.g. through local slip) can blunten the crack tip and relieve the stress concentration³⁶.

Dislocation tangles developed at the centre of the α -phase laths in the Ti-0.34O-3.25Fe alloy, **Fig. B15 a-c**, while extensive dislocation multiplication was apparent in the β -phase, **Fig. B15 d-f**. The former corresponds to the O nano-heterogeneity in the α -phase (much lower O in the interior of the α -lath, main text **Fig. 1C**), while the latter corresponds to the virtually O-free nature and low impediment of the Fe atoms in the β -phase. We propose that these dislocation activities would, in turn, reduce the local stress intensity factor K (proportional to the applied tensile stress σ for sharp cracks) and inhibit the tendency for the fracture of the α -phase, or the α/β interface, or the prior- β GBs, resulting in good tensile ductility ($\epsilon_f = 9.0 \pm 0.8\%$).

The above observations confirm that the virtually O-free β -phase and the O nano-heterogeneity in the α -phase can effectively combine to allow significant plastic deformation. The fracture surface features (**Fig. B17 a-b**) support the above understanding. For example, regions of large, deep dimples are indicative of extensive dislocation multiplication, which we propose correspond to the ductile β -phase. On the other hand, facet-like regions that contain small, shallow dimples are likely due to the fracture of the high-oxygen α -phase (less ductile), or the prior- β GB regions (less ductile), or the high-oxygen α -phase/ β -phase interfacial regions (the α -lath rim near the α/β interface contains much higher oxygen due to the O nano-heterogeneity, main text **Fig. 1C**).

Fig. B15 TEM observations of dislocations in the α -phase (a-c) and β -phase (d-f) of the Ti-0.34O-3.25Fe alloy after tensile fracture ($\epsilon = 9.0 \pm 0.8\%$, $\sigma_{UTS} = 1157 \pm 3$ MPa). The TEM specimen was prepared from a region close to the tensile fracture surface. (a, b, d, e) Bright field images, where (b) and (e) are closer views. (c, f) Dark field images. Dislocations are clearly seen in (c) and (f). Dislocation tangles developed at the centre of the α -phase laths, **Fig. B15 a-c**, while extensive dislocation multiplication was apparent in the β -phase, **Fig. B15 d-f**.

Fig. B16 TEM observations of dislocations in the α phase (a-c) and β phase (d-f) of the Ti-0.67O-3.30Fe alloy after tensile testing ($\epsilon = 3.0 \pm 0.8\%$, $\sigma_{UTS} = 1271 \pm 6$). The TEM specimen was prepared from a region close to the tensile fracture surface. (a, b, d, e) Bright field images, where (b) and (e) are closer views. (c, f) Dark field images. Dislocations are clearly seen in (c) and (f). Dislocation multiplication was observed in the β -phase, but to a lesser extent versus the β -phase in the Ti-0.34O-3.25Fe alloy, corresponding to the overall limited plastic deformation. Only a small number of dislocations were observed in its α -phase, due to the strong pinning effect by the excess of O atoms.

The Ti-0.67O-3.30Fe alloy fractured at $\varepsilon_f = 3.0 \pm 0.8\%$. Only a small number of dislocations were observed in its α -phase, **Fig. B16 a-c**, indicative of a low capacity for dislocation multiplication due to the strong pinning effect by the excess of O atoms. Dislocation multiplication was observed in the β -phase, **Fig. B16 d-f**, but to a lesser extent versus the β -phase in the Ti-0.34O-3.25Fe alloy, corresponding to the overall limited plastic deformation ($\varepsilon_f = 3.0 \pm 0.8\%$). These insufficient dislocation activities imply limited stress concentration relieving at the tips of the tensile cracks in the microstructure. On the other hand, the local stress intensity factor K increases with increasing applied tensile stress σ as tensile deformation continues. Local fracture will occur once K reaches the critical value K_c for the high-oxygen α -phase rim, or the prior- β GBs, or the high-oxygen α -phase/ β interfacial regions, leading to low tensile ductility.

The fracture surface features of this alloy (Ti-0.67O-3.30Fe), **Fig. B17 c-d**, support the above understanding. Facets with small and shallow dimples are prevalent on the fracture surface, **Fig. B17c**. Furthermore, the facet size is well comparable to the prior- β grain size of this alloy. **Fig. B17d** shows discernible intergranular fracture features. The high tensile stress (1271 ± 6 MPa) applied to this alloy leads to a high local K value, which exceeds the critical K_c value for the prior- β GBs, or the high-O α -phase, or the interfacial regions between the high-O α -phase and the β -phase. Local fracture then occurs. The fraction of fracture through each microstructural feature depends on the K_c value of each feature and the applied tensile stress.

The principles discussed above apply to the deformation and tensile fracture behavior of all other Ti-O-Fe alloys fabricated in this work, both inside and outside the required thermal history bounds delineated in **Fig. B7** (the green zone).

Changes made to the manuscript: **Figs. R15 - R17** have been added as **Extended Data Figs. 6-8**. In addition, the above discussion has been incorporated into the revised main text to elaborate on the deformation mechanism.

Fig. B17 Fracture surface features of LMD-fabricated alloys Ti-0.34O-3.25Fe (a, b) and Ti-0.67O-3.30Fe (c, d). (b): closer views from (a). (d): closer views from (c).

11. This points to other comments in this review – is the composition or the additive manufacturing process the key to the success of these alloys? Is it the combination? If additive manufacturing processing is needed, the authors should expand on the phase transformations/phase formations that occur, as well as the critical processing conditions needed for these alloys.

Response: Thank you for these important questions. As we have concluded in our response to **Comment 10.1**, it is indeed **the combination** of the composition and the AM process that results in the success of these alloys. Yes, AM processing is essential — the thermal history bounds required for the fabrication of these Ti-O-Fe alloys have been identified in **Fig. B7** as the green zone, and validated via the suggested control studies. Indeed, the establishment of these critical processing conditions is realised by expanding on the $\alpha' \rightarrow \alpha + \beta$ phase decomposition that occurs in the build. Thank you again for the very insightful suggestions.

Changes made to the manuscript: this response has been incorporated into the **Methods** under “Selection of LMD parameters and the required thermal history bounds or processing window”.

12.1 Fig. 1A: are all 5 of these samples the same composition? Why is the scan pattern of the 4th sample different from the rest? The authors should note which samples these are.

Response: Yes, those five samples shown in the original Fig. 1A all had the same composition. Each composition was printed into 5-10 identical samples for characterisation. We have added a note to the caption of Fig. 1A: **The five coupons in (A) have the same composition of Ti-0.34O-3.25Fe.**

Since we used a bidirectional scan strategy, each odd-number layer and each even-number layer will look different. Therefore, the top surface will look different, depending on whether it is finished as an even-number layer, or an odd-number layer. Those five samples shown in our original Fig. 1A were our first prints to highlight the actual physical difference in the scan path between even-number and odd-number layers. After that, all the samples were finished with the same last odd layer scan, as shown earlier in Fig. B2a, reproduced below.

Changes made to the manuscript: We have replaced our original Fig. 1A with half of the Fig. B2 a below. In addition, we have included in our **Extended Data Table 2** the actual physical difference in the scan path between even-number and odd-number layers, as shown below.

Fig. B2 LMD-fabricated coupons for tensile testing. (a) As-built rectangular coupons of 40 mm × 10 mm × 5 mm at the layer thickness of 200 μm.

12.2 Fig. 1B: in the simulation data, why are there many peaks followed by one peak? Even for a bimodal scan, the laser should pass nearby the same point multiple times during scanning a subsequent layer.

Response: Thank you for this interesting question. Please refer to Fig. B18 (next page).

Each odd-number layer has **nine** scan paths parallel to the length direction of the sample across its width. As a result, **nine peaks** or nine thermal pulses corresponding to these nine scan paths will appear in the previously solidified layers. Please refer to the nine peaks in Layer 17 shown in **Fig. B18 (blue)**.

In contrast, each even-number layer has 38 short scan paths perpendicular to the length direction of the sample. However, each of these short scan paths is not thermally strong enough to result in a detectable thermal pulse in the previously solidified layers. Collectively, they only lead to a limited number of thermal pulses, as shown in Layer 16 in **Fig. B18**. These multi-thermal pulses are important to enable in-situ decomposition of the α' -phase. We have added a note to the caption of our **Fig. B18** about the origin of the multi-peaks.

Fig. B18 Bi-directional scan strategy for LMD (scan speed: 800 mm/min; layer interval: 15 s, see **Extended Data Table 2** for other conditions). The sample has 25 layers. (a) Temperature evolution at the central point of layer 13 under the bidirectional scan strategy. The multiple peaks result from bidirectional scans. (b) A closer view of (a) from 630 to 735 s when layers 16 and 17 were deposited. β_{tr} : β transus; M_s : martensite formation temperature for Ti-6Al-4V. Each odd-number layer has nine scan paths. As a result, nine corresponding peaks or thermal pulses appear. Each even-number layer has 38 short scan paths. Since each short scan path was not hot enough to result in a clear thermal pulse in previous layers, collectively, only a limited number of thermal pulses were observed from the deposition of layer 16.

Changes made to the manuscript: A short paragraph describing the cause of these thermal pulses has been added to the **Methods** under “Thermal History Simulation”. In addition, **Fig. B18** has been added as **Supplementary Fig. 3** to show the correspondence.

12.3 Fig. 4: the labels are cut off (O, HCP, BCC, 15% are all going to a second line and cut off in the presented images).

Response: Thank you. The labels have been corrected

12.4 Extended Data Fig. 3: y-axis should be labelled stress, not strength.

Response: Thank you. We have rectified this incorrect term (now **Extended Data Fig. 2**).

12.5 Extended Data Fig. 4: there’s an extra comma at the end of the caption, and a scale bar would be beneficial in the left figure (although it can be implied from the right figure).

Response: Thank you. We have removed the extra comma and added a scale bar (now **Extended Data Fig. 4a**).

Editor's comments or suggestions

1. Fully establishing the advantage of additive manufacturing (AM) in producing the alloys and its effect on alloy microstructure (Reviewers #1 and #2)

Supplementary Note 3 “Advantages of AM in producing the designed Ti-O-Fe alloys” has been added to address this issue in detail, which explains how the dynamic nature of the AM process delivers a unique alloy microstructure. This is based on new experimental and simulation data. All the key points have been incorporated into the main text.

2. Clarifications on the setup parameter selection as well as the CALPHAD simulations (Reviewer #2)

Setup parameter selection: A new section entitled "Selection of LMD parameters and processing window" has been added to the **Methods**, which details the selection of the setup parameters, and the determination of the required thermal history bounds or processing window.

The CALPHAD simulations: A new section entitled "The CALPHAD (CALculation of PHase Diagrams) method" has been added to the **Methods**, which clarifies that the CALPHAD method was applied to the isothermal holding stage of the LMD process, rather than the fast cooling process.

3. Additional discussion adding context to highlight the novelty of the work and its significance to the additive manufacturing community (Reviewer #1)

Novelty of the work: The two prime novelty points of this work are:

- 1) our integration between alloy design and simulation-based AM process design to create a new class of α - β Ti-(0.35-0.50)O-3Fe alloys with outstanding tensile properties over a large processing window using the inexpensive and abundant alloying elements O and Fe, and
- 2) our discovery of oxygen nano-heterogeneity in the α -phase, and a near oxygen-free β -phase — a unique combination that enables the outstanding tensile properties — demonstrated via state-of-the-art atomic-scale characterisation techniques.

Significance to the additive manufacturing community: we have emphasised the following significance in our revised main text:

- 1) Our study could serve as a pathfinder for new directions in alloy development that harness the power of our approach of coupling alloy design and AM process design to address metallurgical challenges faced by conventional manufacturing. Critically, we have demonstrated a large AM processing window for these exciting new alloys. Their simple chemistry and high printability set the scene for a significant disruption in the metallurgy and manufacturing of α - β titanium alloys.
- 2) Our work offers a promising game-changing approach to revitalising large amounts of off-grade sponge Ti (due to excess O and Fe contamination – sponge Ti-O-Fe) and scrap high-oxygen Ti. These materials can be atomised into powder for AM to produce Ti-O-Fe alloys with outstanding tensile properties for advanced applications. Sponge Ti production is extremely energy-intensive (5-10 times that of primary Al production). Their utilisation for AM represents significant potential economic and environmental benefits. The same could be expected for similarly produced off-grade sponge Zr.
- 3) Oxygen embrittlement occurs not only in HCP and BCC Ti, but also in other BCC metals (e.g. Nb and Mo), presenting a significant metallurgical challenge. However, here we reveal a unique distribution of O in these AM-fabricated α - β Ti-O-Fe alloys. The same concept could provide a template for future interstitial engineering by AM.
- 4) A potential pathfinding approach to various other new alloy systems, including strong and ductile AM-fabricated α - β Zr-O-Fe alloys, α - β Ti-N-Fe or Zr-N-Fe alloys (Supplementary Note 6). This would help to further change the landscape of α - β Ti alloys and their manufacture.

Changes made to the manuscript: We have incorporated the above key points into our revised main text and have also added Supplementary Note 6 “Potential applications and implications”.

References

- 1 Zeng, W. & Zhou, Y. Effect of beta flecks on mechanical properties of Ti–10V–2Fe–3Al alloy. *Mater. Sci. Eng. A* **260**, 203-211 (1999).
- 2 Xu, W. *et al.* Additive manufacturing of strong and ductile Ti–6Al–4V by selective laser melting via in situ martensite decomposition. *Acta Mater.* **85**, 74-84, (2015).
- 3 Cao, S. *et al.* Role of martensite decomposition in tensile properties of selective laser melted Ti-6Al-4V. *J. Alloys Compd.* **744**, 357-363, (2018).
- 4 Song, T. *et al.* Simulation-informed laser metal powder deposition of Ti-6Al-4V with ultrafine α - β lamellar structures for desired tensile properties. *Addit. Manuf.* **46**, 102139 (2021).
- 5 Gao, F. *et al.* Environmental impacts analysis of titanium sponge production using Kroll process in China. *J. Clean. Prod.* **174**, 771-779 (2018).
- 6 Marui, Y., Kinoshita, T. & Takahashi, K. Development of a titanium material by utilizing off-grade titanium sponge. *SAE Tech. Pap.* **32**, 1816, (2002).
- 7 Takeda, O., Ouchi, T. & Okabe, T. H. Recent progress in titanium extraction and recycling. *Metall. Mater. Trans. B* **51**, 1315-1328, (2020).
- 8 Taninouchi, Y.-k., Hamanaka, Y. & Okabe, T. H. in Proceedings of the 13th World Conference on Titanium 165-170 (2016).
- 9 Polmear, I., StJohn, D., Nie, J.-F. & Qian, M. *Light Alloys: Metallurgy of the Light Metals.* (Butterworth-Heinemann, 2017).
- 10 Yang, P.-J. *et al.* Mechanism of hardening and damage initiation in oxygen embrittlement of body-centred-cubic niobium. *Acta Mater.* **168**, 331-342 (2019).
- 11 Wang, Z.-Q. *et al.* Suppressing effect of carbon on oxygen-induced embrittlement in molybdenum grain boundary. *Comput. Mater. Sci.* **198**, 110676 (2021).
- 12 Wu, Q. *et al.* Effect of molten pool size on microstructure and tensile properties of wire arc additive manufacturing of Ti-6Al-4V alloy. *Materials* **10**, 749 (2017).
- 13 Mok, S. H., Bi, G., Folkes, J. & Pashby, I. Deposition of Ti–6Al–4V using a high power diode laser and wire, Part I: Investigation on the process characteristics. *Surf. Coat. Technol.* **202**, 3933-3939, (2008).
- 14 Tran, H. S. *et al.* 3D thermal finite element analysis of laser cladding processed Ti-6Al-4V part with microstructural correlations. *Mater. Des.* **128**, 130-142, (2017).
- 15 ATI 425[®] Alloy (Grade 38) Technical Data Sheet http://atimaterials.com/Products/Documents/datasheets/titanium/alloyed/ati_425_alloy_tds_en_v5.pdf (Allegheny Technologies Incorporated, 2013)
- 16 Osipenko, A. B. Development of technology of obtaining raw materials for titanium alloys made of off-grade titanium sponge. *East. -Eur. J. Enterp.* **4**, 28-32, (2015).
- 17 Repeatability and reproducibility. *Stats Book* http://pcool.dyndns.org:8080/statsbook/?page_id=835.
- 18 *Determination of Precision of Analytical Methods* (AOCS Procedure M 1-92, 2017)

- 19 *Material Data Sheet* – EOS Titanium Ti64 M290 (EOS GmbH Electro Optical Systems, 2022)
- 20 *Material Data Sheet* – Ti6Al4V Grade 23 ELI (SLM Solutions, 2022).
- 21 Amirabdollahian, S. *et al.* Towards controlling intrinsic heat treatment of maraging steel during laser directed energy deposition. *Scr. Mater.* **201**, 113973, (2021).
- 22 Ogunlana, M. O. & Akinlabi, E. T. in Proceedings of the world congress on engineering and computer science (2016).
- 23 Kozieł, T. Estimation of cooling rates in suction casting and copper-mould casting processes. *Arch. Metall. Mater.* **60**, 767--771 (2015).
- 24 Shamblen, C. E. Minimizing beta flecks in the Ti-17 alloy. *Metall. Mater. Trans. B* **28**, 899-903 (1997).
- 25 Mitchell, A., Kawakami, A. & Cockcroft, S. Beta fleck and segregation in titanium alloy ingots. *High Temp. Mater. Process.* **25**, 337-349 (2006).
- 26 Moiseyev, V. N. *Titanium Alloys: Russian Aircraft and Aerospace Applications*. (CRC Press, 2005).
- 27 Haubrich, J. *et al.* The role of lattice defects, element partitioning and intrinsic heat effects on the microstructure in selective laser melted Ti-6Al-4V. *Acta Mater.* **167**, 136-148, (2019).
- 28 Zhang, J. *et al.* Designing against phase and property heterogeneities in additively manufactured titanium alloys. *Nat. Commun.* **13**, 4660, (2022).
- 29 Kumar, S. *et al.* Role of thermo-mechanical gyrations on the α/β interface stability in a Ti6Al4V AM alloy. *Scr. Mater.* **204**, 114134, (2021).
- 30 Tan, X. *et al.* Graded microstructure and mechanical properties of additive manufactured Ti-6Al-4V via electron beam melting. *Acta Mater.* **97**, 1-16, (2015).
- 31 Sridharan, N. *et al.* On the potential mechanisms of β to $\alpha' + \beta$ decomposition in two phase titanium alloys during additive manufacturing: a combined transmission Kikuchi diffraction and 3D atom probe study. *J. Mater. Sci.* **55**, 1715-1726, (2020).
- 32 Tan, X. *et al.* Revealing martensitic transformation and α/β interface evolution in electron beam melting three-dimensional-printed Ti-6Al-4V. *Sci. Rep.* **6**, 26039, (2016).
- 33 Martin, T. L. *et al.* Insights into microstructural interfaces in aerospace alloys characterised by atom probe tomography. *Mater. Sci. Technol.* **32**, 232-241, (2016).
- 34 Cottrell, A.H. & Bilby, B.A. Dislocation theory of yielding and strain ageing of iron. *Proceedings of the Physical Society. Section A*, **62**(1), 49-62, (1949).
- 35 Hull, D. & Bacon, D. J. *Introduction to Dislocations*. (Elsevier, 2011).
- 36 Anderson, P. M., Hirth, J. P. & Lothe, J. *Theory of Dislocations*. (Cambridge University Press, 2017).

Reviewer Reports on the First Revision:

Referees' comments:

Referee #1 (Remarks to the Author):

Dear authors, thank you for addressing all the review with such thorough explanation.

Originality and significance

Thank you for expanding on the 4 points that demonstrate the usefulness of additive manufacturing versus other conventional techniques. The ingot experiment proves quite well that ingot metallurgy does not seem suitable for this alloy. Can the authors substantiate by providing a range of thermal where β -flecks are avoided? Presumably this relates to the cooling rate during solidification so it is surprising to see that β -flecks form in small ingots but are prevented in builds where laser traverse speed as low as 200mm/s. Does this relate to a critical melt pool size beyond which flecks start to form? If so, how the CALPHAD method is still applicable?

I accept the general point of F3 although the authors should emphasise that DED is not a net-shape technology at least when it comes to complex shapes/fine features where extensive machine is required.

Thank you for addressing the comments on the printability. The link between processing and microstructure evolution is well expressed presented temperature vs. layers charts and highlighted a critical band of temperatures. However, the authors should demonstrate that "simufact welding" can accurately predict melt pool size and thermal cooling under such a vast range of process parameters, otherwise a difference in ductility could be due to different amounts of porosity in the builds.

Thanks for expanding the section on applicability -very interesting and looking forward to seeing more alloys developed using this approach.

Data & methodology

Thank you for the detailed explanation

Clarity and context

Thank you for the interesting comments.

Referee #2 (Remarks to the Author):

The authors sufficiently addressed the reviewer comments.

The writing in the supplementary notes is a bit informal (e.g., "This is remarkable and exciting"), and it is suggested to make the writing more factual.

I would suggest the authors reduce the number of significant figures in the tables in the supplementary notes.

The following sentence in supplementary notes is unclear: "They are not expected to exhibit outstanding tensile properties, but they can still be used for low-end applications."

Author Rebuttals to First Revision:

Referee #1 (Remarks to the Author):

Comment: Thank you for expanding on the 4 points that demonstrate the usefulness of additive manufacturing versus other conventional techniques. The ingot experiment proves quite well that ingot metallurgy does not seem suitable for this alloy. Can the authors substantiate by providing a range of thermal where β -flecks are avoided? Presumably this relates to the cooling rate during solidification so it is surprising to see that β -flecks form in small ingots but are prevented in builds where laser traverse speed as low as 200mm/s.

Response: Thank you for these further important comments. We have determined, through experiments and simulations, the approximate thermal conditions that helped to avoid the Fe-stabilised β -flecks in our Ti-O-Fe alloys (we are preparing a separate paper on this).

Yes, the solidification cooling rate T is an important and possibly decisive factor, as it may affect solute trapping depending on *the actual solid-liquid (S-L) interface velocity* V [1, 2], where V is related to the undercooling ΔT at the interface and a few other kinetic factors. Unfortunately, the relationship between ΔT and T has not been well established. In general, for small liquid volumes, it is convenient to use T [1] because T can be measured or estimated directly. For example, for 75-1000 μm Cu-22Sn alloy droplets, the complete trapping of solute Sn occurred at $T \geq 3.5 \times 10^4$ $^{\circ}\text{C/s}$ [3] (the decisive factor in this case). For the closely relevant Ti-1Al-8V-5Fe (Ti-185) alloy, it was found that plasma atomized spherical Ti-185 powders (50-150 μm) exhibited a uniform distribution of Fe (no β -flecks) [4], indicative of complete solute (Fe) trapping. The T of the 150 μm Ti-185 droplets is $\sim 1.3 \times 10^5$ $^{\circ}\text{C/s}$ (estimated using Eqs. 3 and 4 of Ref. [5]). Again, the high T should be the decisive factor for the avoidance of β -flecks in this case because Ti-185 is prone to the formation of Fe-stabilised β -flecks in ingot metallurgy due to its 5%Fe.

In the following, we first determine the T of our Cu-mould cast Ti-0.35O-3Fe alloy and then, through simulations, determine the approximate slowest or minimum T of our DED-fabricated alloy samples (all samples are free of Fe-stabilised β -flecks). Here, we draw attention to the fact that all of our experimental results used laser traverse speeds of ≥ 400 **mm/min**, because 200 mm/min is too slow for real production. Also, a much lower laser power should be used to

ensure the desired range of laser energy density ($E_d = 25\text{-}35 \text{ J/mm}^2$, **Table R1**). The curve in **Fig. 1c** related to **200 mm/min** was obtained by simulations.

1. Cooling rate in Cu-mould casting and the formation of Fe-stabilised β -flecks

The formation of Fe-stabilised β -flecks arises from the excessive accumulation of Fe in the remaining liquid (often when the solid fraction $f_s \geq 0.8$ [6]) due to (i) the continuous rejection of the Fe as solidification continues or f_s increases and (ii) the insufficient back-diffusion of Fe from the remaining liquid into the solid (i.e. the prior- β grains).

Our Cu-mould cast Ti-0.35O-3Fe ingots were 5 mm thick. Experiments have shown that the T in the central region of a 5-mm diameter Cu-mould cast ingot is in the range of about 10-80 $^\circ\text{C/s}$ (600-4800 $^\circ\text{C/min}$) for both ferrous (Fe-25Ni) and non-ferrous (Al-33Cu) alloys [7, 8], although the surface can reach $10^4 \text{ }^\circ\text{C/s}$ [7]. There were pores in the central region of our as-cast Ti-0.35O-3Fe alloy (Extended Data Fig. 1d). This provides a unique opportunity to study its T . **Fig. R1** shows the dendritic prior- β grains observed in a pore cavity on the fracture surface of this alloy. The secondary dendrite arm spacing (λ_2) is measured to be $13.83 \pm 0.81 \text{ } \mu\text{m}$ (excluding tertiary dendrites).

Fig. R1 Dendritic prior- β grains in a pore cavity near the central region of a Cu-mould cast Ti-0.35O-3Fe alloy ingot sample. The image is taken from the tensile fracture surface of the alloy.

An experimental relationship between the secondary dendrite arm spacing λ_2 and the solidification cooling rate T has been established for Ti-6Al-4V at cooling rates below 2430 °C/s [9]

$$A_2 = 108.92T^{-0.46} \quad (1)$$

As an approximate estimate, Eq. (1) predicts $T = 89$ °C/s for our Cu-mould cast 5-mm thick Ti-0.35O-3Fe ingot, close to the reported approximate upper limit (80 °C/s) in the central region of the 5-mm thick Cu-mould cast ingots for both Fe-25Ni and Al-33Cu [7, 8].

It is generally accepted that “For cooling rates up to 10^3 K/s, local equilibrium with compositional partitioning between the liquid and solid phases at the solidification interface is maintained. The interface undercooling is small. However, when the cooling rate increases above 10^3 K/s, nonequilibrium solidification occurs” [10], namely solute trapping occurs above 10^3 °C/s. Therefore, at the cooling rate of $T = 89$ °C/s, no solute trapping is expected. Consequently, a significant accumulation of Fe is expected to occur in the remaining liquid. **This is the key reason** for the formation of the Fe-stabilised β -flecks in our Cu-mould cast Ti-0.35O-3Fe ingots.

It should be emphasised that the solidification of an ingot always starts from the mould walls and proceeds towards the centre. This sequential solidification process leads to continuous accumulation of Fe in the remaining liquid, conducive to the formation of β -flecks. This is *another* important contributing factor (melt volume is therefore important, which in turn affects the T). Conversely, in the DED process, this cumulative effect is much weaker due to the small melt pool (1.08 mm^3), which corresponds to a much faster T ($> \sim 2 \times 10^3$ °C/s, see below).

2. Approximate minimum T to help avoid β -flecks in the DED process used in this study

Before studying our Ti-O-Fe alloys, we first investigated the β -fleck issue in three binary Ti-Fe alloys: **Ti-3Fe**, **Ti-5Fe** and **Ti-7Fe** (all containing 0.13-0.14 O after DED). **Table R1** lists the predicted cooling rates in Ti-3Fe rectangular coupons built on a 10 mm-thick Ti-6Al-4V plate under different DED conditions. Each sample has 25 layers. Due to the lack of similar data for Ti-Fe alloys, we used the temperature-dependent thermophysical data of Ti-6Al-4V. Room temperature thermophysical property data cannot be used because these properties vary considerably between T_{room} and T_{liquidus} (e.g. 4-6 times) [11]. As expected, T decreases with increasing laser energy density but is influenced by the layer interval time. In each 25-layer

build, the centres of the layers **18**, **20** and **22** undergo slower cooling rates, with the slowest **T** being predicted to be **1994 °C/s**.

Table R1 Predicted solidification cooling rates at the centres of different layers in 25-layer builds ($40 \times 10 \times 5 \text{ mm}^3$) when cooled from the third (last) remelting state to 1500 °C (solute enrichment lowers the T_{solidus} of the Ti-3Fe alloy in the final liquid to $\sim 1500 \text{ °C}$).

DED condition (laser power – scan speed – layer interval)	Laser energy density E_d (J/mm^2) ^a	Solidification cooling rate T at the centre of layer n ($^{\circ}\text{C/s}$) – each sample has 25 layers					
		1	13	18	20	22	25
500 W–1200 mm/min–15 s	16.7	7054	6961	3251	3236	3527	12712
500 W–800 mm/min–60 s	25	7068	6507	3089	3074	3315	12551
500 W–800 mm/min–120 s	25	7211	6890	3223	3278	3478	13291
500 W–800 mm/min–0 s	25	6306	5311	2509	2492	2396	9981
500 W–800 mm/min–15 s	25	6151	5914	2796	2784	2729	11033
500 W–600 mm/min–15 s	33.3	5881	4968	2391	2462	2339	9695
500 W–400 mm/min–15 s	50	4532	4069	2080	2018	1994	6936

The optimum range of E_d for determined for Ti-(0.35-0.50)O-3Fe alloys is 25-35 J/mm^2 (Extended Data).

Table R2 DED schedules for Ti-3Fe, Ti-5Fe and Ti-7Fe alloys

Laser power (W)	Laser spot size (mm)	Traverse speed (mm/min)	Layer interval time (s)	Energy density E_d (J/mm^2)	Powder flow rate (g/min)	Step over (mm)	Overlap (%)
500	1.5	1200	15	16.7	1.7	1.05	70
500	1.5	800	15	25	1.7	1.05	70
500	1.5	400	15	50	1.7	1.05	70

To evaluate the influence of the cooling rate on the formation of Fe-stabilised β -flecks, we printed rectangular coupons ($40 \times 10 \times 5 \text{ mm}^3$; thickness: 5 mm) of Ti-3Fe, Ti-5Fe and Ti-7Fe alloys. Each composition was printed with three sets of DED conditions, listed in **Table R2**. Each printed sample was examined layer by layer. **No β -flecks** were observed in any of these alloy samples.

We first discuss the Ti-7Fe alloy (**Fig. R2**) and then briefly discuss the Ti-3Fe and Ti-5Fe alloys (**Fig. R3**). All micrographs were taken from around the 18th layer of each sample. The highest energy density (50 J/mm^2 , $v = 400 \text{ mm/min}$) produced the coarsest α - β microstructure, coupled with acicular secondary α in the remaining Fe-containing β -phase (**Fig. R2a-b**), due

to the pronounced effect of the thermal cycles. The desired laser energy density (25 J/mm²) resulted in ultrafine (~100 nm thick) α - β lamellae (**Fig. R2c-d**). Unsurprisingly, the lowest energy density (16.7 J/mm², $v = 1200$ mm/min) entailed some defects (black dots in **Fig. R2e-f**). **Fig. R3** shows a brief view of the microstructures of the Ti-3Fe and Ti-5Fe alloys printed under three DED conditions. At higher magnifications, they all consist of ultrafine (100-250 nm thick) α - β lamellae.

In summary, the DED conditions investigated in **Table R1** were all successful in avoiding the β -flecks, including in the Ti-7Fe alloy. **All these DED conditions fall in the green zone of Fig. 1c.** The **approximate slowest T** identified by simulations for these alloy samples is **~2000 °C/s**.

Note that this approximate slowest solidification cooling rate (**~2000 °C/s**) and those predicted for the layers of 18, 20 and 22 in each sample (2000 – 3500 °C/s, **Table R1**) are clearly slower than the cooling rates (10⁴-10⁵ °C/s) indicated earlier for complete solute trapping. Therefore, these cooling rates allow only partial solute (Fe) trapping to occur, reducing Fe accumulation in the remaining liquid, which helps avoid the formation of Fe-stabilised β -flecks.

As emphasized earlier, the small melt pool (1.08 mm³) in the DED process limits Fe accumulation in the remaining liquid ($f_s > 0.8$) compared to ingot solidification. The actual accumulation of Fe is much weaker. This is another important factor to mitigate the formation of Fe-stabilised β -flecks.

Finally, the multiple thermal pulses and significant cyclic heating effects of the DED process, unlike conventional annealing, may decompose some of the Fe-rich β -phases. According to our literature review, this effect has not been investigated and in our opinion should not be neglected.

Therefore, we propose that the complete avoidance of Fe-stabilised β -flecks in these Ti-(3-7)Fe alloys may be the combined effect of the three factors mentioned above. **However, the prerequisite is the small melt pool, which determines the fast cooling rates (≥ 2000 °C/s) and limited Fe accumulation in the remaining liquid ($f_s > 0.8$).**

Changes made to the manuscript: A new section entitled “1.2 Avoidance of Fe-stabilised β -flecks by DED” has been added to Supplementary Note 1, covering the entire discussion above. On this basis, we have added the following sentence **in red** to the main text (Page 5):

No Fe-stabilised β -flecks were observed in any of these printed alloys, but they were prevalent in the copper-mould-cast Ti-0.35O-3Fe alloy (Extended Data Fig. 1). The reasons are discussed in Supplementary Note 1.

Fig. R2 Backscattered electron images of the microstructures of the Ti-7Fe alloy printed under three sets of DED conditions (**Table R2**). **(a, b)** 500 W, 400 mm/min, 50 J/mm². **(c, d)** 500 W, 800 mm/min, 25 J/mm². **(e, f)** 500 W, 1200 mm/min, 16.7 J/mm². **(a, c, e)** Low magnification. **(b, d, f)** High magnification. No β -flecks were observed along the build height of each sample layer by layer. The microstructure shown was observed from the region around the layer 18 of each sample, which exhibits approximately the lowest solidification cooling rate in each sample. No β -flecks were observed in any of these alloy samples.

Fig. R3 Backscattered electron images of the microstructures of the Ti-3Fe alloy (a, c, e) and Ti-5Fe alloy (b, d, f) printed under three sets of DED conditions (Table R2). (a, b) 500 W, 400 mm/min, 50 J/mm². (c, d) 500 W, 800 mm/min, 25 J/mm². (e, f) 500 W, 1200 mm/min, 16.7 J/mm². The microstructure shown was observed from the region around the layer 18 of each sample, which exhibits approximately the lowest solidification cooling rate in each sample. No β -flecks were observed in any of these alloy samples. Most of the *irregular black dots* in (a) and (c) are not pores — they are coarser α -phase particles (see subsequent clarifications in Fig. R7).

Comment: Does this relate to a critical melt pool size beyond which flecks start to form? If so, how the CALPHAD method is still applicable?

Response: We appreciate these insightful questions! As mentioned earlier, the melt pool size affects not only the cooling conditions, but also the Fe accumulation in the remaining liquid. Therefore, in our opinion, it is a key factor affecting the formation of Fe-stabilised β -flecks. For example, if the melt pool size increases to such a level that the cooling rate falls below ~ 1000 °C/s or a local equilibrium (k_{eq}) is maintained at the S-L interface [10], then Fe-stabilised β -flecks will tend to form (back-diffusion is limited). Therefore, it is logical to assume that there is a critical melt pool size.

On the other hand, it is important to note that if the DED (powder) process does not provide high cooling rates and is only used as a near net shape manufacturing process, its main attraction will be greatly reduced. As such, in our opinion, mainstream DED (powder) practices generally aim to achieve both high cooling rates *and* near net-shape manufacturing.

The applicability of the CALPHAD method. Our answer is related to the importance of the melt pool size. If the melt pool size exceeds a critical value, the solidification cooling rate is likely to be slower than 1000 °C/s. Consequently, local equilibrium for solute partitioning between the liquid and solid at the S-L interface or k_{eq} is likely to be maintained [10]. In this regime, one can apply the Scheil equation or the CALPHAD-numerical-Scheil. Both assume the local equilibrium at the liquid/solid interface *as the key foundation, also no supercooling*.

As emphasised in our last response, the CALPHAD predictions presented in this work are limited to the approximate isothermal holding period in the solid state. The same CALPHAD predictions can still be made. However, in this case, due to the formation of the Fe-stabilised β -flecks, the overall alloy composition is no longer valid as the input composition for CALPHAD. There are different ways to determine the effective composition of the matrix. One approach is to measure the volume fraction of the Fe-stabilised β -flecks by metallographic methods and then determine the average Fe content in them using electron probe microanalysis or energy dispersive X-ray spectroscopy. This would allow one to calculate the effective alloy composition responsible for the development of the α - β microstructure after solidification. This effective alloy composition could then be used as the input composition for a further iteration of CALPHAD predictions for the selected approximate isothermal holding temperature in the solid state.

Changes made to the manuscript: We have added the following sentence in red to Supplementary Note 1 (“1.2 Avoidance of Fe-stabilised β -flecks in Ti-O-Fe alloys”)

Note that the Scheil equation or CALPHAD-numerical-Scheil is only applicable when the assumption of local equilibrium is valid.

Comment: I accept the general point of F3 although the authors should emphasise that DED is not a net-shape technology at least when it comes to complex shapes/fine features where extensive machine is required.

Response: We agree and have now emphasised that DED is a near-net-shape technology in the main text as follows:

Main text

We chose laser metal powder directed energy deposition (DED) which, with the aid of high-fidelity simulations, allows for the fabrication of large-scale near-net-shape components with a consistent microstructure.

We have also added the term “near-net-shape” to the abstract.

Comment: The link between processing and microstructure evolution is well expressed presented temperature vs. layers charts and highlighted a critical band of temperatures. However, the authors should demonstrate that "simufact welding" can accurately predict melt pool size and thermal cooling under such a vast range of process parameters, otherwise a difference in ductility could be due to different amounts of porosity in the builds.

Response: Over the course of the research that has led up to this manuscript (~ four years of research), we have given careful consideration to these issues in developing the DED process for these new alloys. Since the concern raised relates to the potential effect of porosity on ductility, we first investigate the tendency for porosity in different builds and then focus on the assessment of the melt pool size and thermal cooling predictions.

POROSITY: As previously indicated, our DED process (laser spot size: 1.5 mm; energy density: 25-35 J/mm²; laser power: ≥ 500 W; overlap: 70%; powder feed: 1.7 g/min; layer thickness: 200 μm ; layer interval: 0-15 s; scan strategy: bidirectional) consistently produces high-quality builds (Extended Data). Among all the printed Ti-0.35O-3Fe samples, the **lowest** tensile ductility ($\epsilon_f = 2.2 \pm 0.6$ %) was produced from 500 W – 800 mm/min – 120 s – 25 J/mm²; the **highest** tensile ductility ($\epsilon_f = 21.9 \pm 2.2$ %) from 500 W – 800 mm/min – 0 s – 25 J/mm², and an **intermediate** tensile ductility ($\epsilon_f = 14.0 \pm 0.7$ %) from 500 W – 600 mm/min – 0 s – 33.3 J/mm². In terms of porosity, in fact, the Ti-0.35O-3Fe alloy samples with **the lowest porosity** and **smallest pores** exhibited the poorest ductility ($\epsilon_f = 2.2 \pm 0.6$ %), as demonstrated below.

Fig. R4 shows a global view of the tensile fracture surfaces of these Ti-0.35O-3Fe samples with $\epsilon_f = 2.2 \pm 0.6$ % (**a**), $\epsilon_f = 14.0 \pm 0.7$ % (**b**) and $\epsilon_f = 21.9 \pm 2.2$ % (**c**). The number of discernible pores on each fracture surface is self-consistent and is typically ~ 5 -6 only (red circles). This observation is similar for other samples reported in our Extended Data Fig. 2(a, b). Metallographically, pores were rarely observed in polished cross-sections, as exemplified in **Fig. R5** for the Ti-0.35O-3Fe alloy that exhibited the lowest ductility.

As **Fig. R4a** confirms, five pores in the size range of 16-46 μm were randomly distributed on the entire tensile fracture surface of this lowest tensile ductility sample ($\epsilon_f = 2.2 \pm 0.6$ %). Compared to the same alloy samples in **Fig. R4b** and **Fig. R4c**, this sample has the least and smallest pores, but the lowest tensile ductility. Clearly, porosity is not the major factor controlling ductility here.

Compared with **Fig. R4a**, seven larger pores in the size range of 45-72 μm were observed on the entire tensile fracture surface of the sample with $\epsilon_f = 14.0 \pm 0.7$ % (**Fig. R4b**). However, the tensile ductility was **six** times higher (more pores with larger sizes). This reaffirms that porosity is not related to the substantially low ductility of the sample shown in **Fig. R4a**.

The last sample (the same alloy) shown in **Fig. R4c** exhibited the highest tensile ductility ($\epsilon_f = 21.9 \pm 2.2$ %). Six pores (red circles) in the size range of 20-65 μm were observed on the fracture surface. Similarly, there were more pores with larger sizes than those shown in **Fig. R4a**. However, the tensile ductility of this sample was almost **10** times higher. Again, this indicates that porosity is not responsible for the poor ductility of the sample shown in **Fig. R4a**.

In fact, this high tensile ductility ($21.9 \pm 2.2 \%$) may suggest that spherical pores in the size range of $20\text{--}65 \mu\text{m}$ (small quantity) are not significantly detrimental to the tensile ductility of these titanium alloys at the strain rates tested here.

Fig. R4 Influence of porosity on the tensile ductility of the Ti-0.35O-3Fe alloy printed under different DED conditions. (a) 500 W – 800 mm/min – 120 s – 25 J/mm²; tensile ductility: $2.2 \pm 0.6 \%$; pore size: 16-46 μm (five pores). (b) 500 W – 600 mm/min – 15 s – 33.3 J/mm²; tensile ductility: $14.0 \pm 0.7 \%$; pore size: 45-72 μm (seven pores). (c) 500 W – 800 mm/min – 0 s – 25 J/mm²; tensile ductility (ϵ_t): $21.9 \pm 2.2 \%$, pore size: 20-65 μm (six pores).

As mentioned earlier, our metallographic examinations rarely revealed any porosity in the microstructures of these Ti-O-Fe alloys deposited with the desired laser energy density conditions (25-35 J/mm², **Table R1**). **Fig. R5a** provides a low-magnification view of the as-manufactured microstructure of the Ti-0.35O-3Fe alloy, which exhibited the lowest tensile ductility ($\epsilon = 2.2 \pm 0.6 \%$). No pores were observed. There were some irregular black dots, which are not pores but coarser α -phase particles, as shown in **Fig. R5b** (right edge).

Fig. R5 Metallographic examination rarely reveals any porosity in the Ti-0.35O-3Fe alloy printed under the desired DED conditions described in the work ($E_d = 25-35 \text{ J/mm}^2$). Although this alloy exhibited the lowest tensile ductility ($\epsilon_f = 2.2 \pm 0.6 \%$), it is not due to porosity. There are some *irregular black dots* in (a), which are not pores, but coarser α -phase particles, as shown in (b) (right edge).

MELT POOL SHAPE AND SIZE: Before applying Simufact Welding (DED) to our Ti-O-Fe alloys, we systematically investigated the melt pool development by single-track DED experiments with Ti-6Al-4V under various conditions. Our Simufact Welding (DED) simulations are based on single-track DED experiments in order to properly define the heat source geometry (**Extended Data Table 2**) for our simulations. In other words, the basic input conditions for simulations are based on the single-track DED experiments obtained under various DED conditions.

As emphasized earlier, due to the lack of similar data for Ti-O-Fe alloys, our simulations used the temperature-dependent thermophysical data of Ti-6Al-4V (the room-temperature data cannot be used because the thermophysical properties vary considerably between T_{room} and T_{liquidus} (4-6 times) [11]. Therefore, to best assess the predictability of the Simufact Welding (DED) used for this study, we chose to compare the simulation results with the single-track DED experiments for Ti-6Al-4V.

Fig. R6a-f compares the single-track DED Ti-6Al-4V melt pools (experimental) with the simulated melt pools for three DED conditions, 500 W – 400 mm/min (**Fig. R6a-b**); 500 W – 800 mm/min (**Fig. R6c-d**); and 500 W – 1200 mm/min (**Fig. R6e-f**). See Table R3 or Extended Data Table 2 for other parameters. The comparisons are quantified in **Table R3**.

It is noteworthy that the density of Ti (ρ_{Ti}) increases profoundly when cooled from the melt pool temperature (up to 3100 °C by simulation for DED) to room temperature (RT, 4.51 g/cm³). The authors of Ref. [12] reviewed the density measurements of molten Ti and experimentally determined the following relationship for the density of molten Ti up to 2127 °C (2400 K)

$$\rho_{\text{Ti}} = 4.14 - 2.15 \times 10^{-4} (T - T_m) - 3.71 \times 10^{-8} (T - T_m)^2 \quad (2)$$

The latest study of the density of molten Ti (up to 1817 °C) [13] is consistent with Eq. (2).

Extrapolating Eq. (2) yields $\rho_{\text{Ti}} = 3.756 \text{ g/cm}^3$ at 3100 °C ($T_m = 1668 \text{ °C}$). This means a 20% increase in density when cooled to RT. Therefore, the simulated melt pool volume is expected to be at least 20% larger than that observed at RT. In other words, if the difference is within 20-30%, it should be considered highly consistent. Our **Fig. R6a-f** and **Table R3** confirm the high predictability (on the micron length scale) of the DED simulation module we used in this work.

Fig. R6 Experimental (single track DED) and simulated DED melt pools for Ti-6Al-4V. (a) Experimental (laser power: 500 W, spot size: 1.5 mm, scan speed: 1200 mm/min). (b) Simulated (Simufact Welding DED). (c) Experimental (laser power: 500 W, spot size: 1.5 mm, scan speed: 800 mm/min). (d) Simulated (Simufact Welding DED). (e) Experimental (laser power: 500 W, spot size: 1.5 mm, scan speed: 400 mm/min). (f) Simulated (Simufact Welding DED).

Table R3 Experimental and simulated melt pools for Ti-6Al-4V ($T_{\text{Solidus}} = 1604 \text{ }^\circ\text{C}$) via a 40 mm long single track (the melt pool profile was taken from the middle of the track).

DED condition	Melt pool (experimental) ^a			Melt pool (simulated) ^b		
	Width, μm	Height, μm	Penetration ^c , μm	Width, μm	Height, μm	Penetration, μm
500 W 400 mm/min	1479	719	68	1640	803	74
500 W 800 mm/min	1202	469	108	1264	538	133
500 W 1200 mm/min	1073	351	125	1160	412	159

^a Laser spot size: 1.5 mm; powder flow rate: 1.7 g/min; carrier gas (He) flow rate: 10 L/min; shielding gas (Ar) flow rate: 16 L/min.

^b The heat source for simulation has a Gaussian parameter of 1 and an absorption efficiency of 35%.

^c Penetration: the depth of the melt pool below the substrate surface

Substrate: a 200 mm \times 100 mm \times 12 mm Ti-6Al-4V plate for both experiments and simulations.

THERMAL COOLING: The above assessment can be used as a valid evaluation of the predicted thermal cooling predictions. Direct and accurate measurements of the cooling rate (solidification or solid state) remain a challenge for a typical DED process due to the small melt pool and the layer additive manufacturing nature (**the location of interest does not pre-exist**).

An indirect assessment of the cooling rate for DED of Ti alloys is to use the relationship between the secondary dendrite arm spacing (λ_2) or the prior- β grain size (λ_1) and the solidification cooling rate T . Unlike the Cu-mould cast ingots (**Fig. R1**), no prior- β dendrites were observed in any DED-fabricated Ti-O-Fe alloy samples of this work (all being elongated or equiaxed prior- β grains). Therefore, we focus on the prior- β grain size λ_1 .

Broderick et al [5] have established that the prior- β grain size λ_1 (μm) of Ti-6Al-4V under rapid solidification conditions can be described as a function of the T (K/s), i.e.

$$\lambda_1 = 3.1 \times 10^6 T^{-0.93 \pm 0.12} \quad (3)$$

Eq. (3) was established for Ti-6Al-4V, which may need to be modified for our Ti-O-Fe alloys, because the observed prior- β grains are much finer than the prior- β grains of Ti-6Al-4V under the same DED conditions. This implies that the exponent of T in Eq. (3) should be greater than 1.05 (0.93 + 0.12). By applying Eq. (3) to the DED-fabricated binary Ti-3Fe alloy samples shown earlier (**Fig. R3**) using the predictions of T in **Table R2**, the exponent of T was found to be around 1.15. Therefore, we used the following Eq. (4) to further evaluate the solidification rate T of our Ti-O-Fe alloys studied in this work and then compare the results with Simufact Welding (DED) predictions:

$$A_1 = 3.1 \times 10^6 T^{-1.15} \quad (4)$$

We measured the prior- β grain size in the surface layer of the Ti-0.35O-3Fe alloy samples printed under four sets of DED conditions (**Table R4**). Representative prior- β grain structures observed in the surface layer of these samples are shown in **Fig. R11** for each selected DED condition, while **Table R4** summarises the predictions and measurements.

The equiaxed prior- β grains are not uniform in the surface layer of each sample, **Fig. R11**, featured by a large standard deviation. We therefore focused on the mean prior- β grain size. As shown in **Table R4**, in each case, the prior- β grain size obtained from Eq. (4) based on the predicted cooling rate matched well with the measured mean prior- β grain size on the micron length scale, reaffirming the good predictability of the Simufact Welding (DED) used for this study.

Changes made to the revised manuscript: Two new sections entitled “2.2 Influence of porosity on tensile ductility” and “2.3 Predictability of the Simufact Welding (DED)” have been added to Supplementary Note 2, covering the entire discussion above. On this basis, we have added the following sentence in red to the "Thermal History Simulation" section of **Methods**:

The directed energy deposition (DED) module in *Simufact Welding* was used to track temperature evolution in the build²⁸. **The predictability of Simulated Welding (DED) in terms of melt pool shape and size and thermal cooling was evaluated in Supplementary Note 2.**

In addition, we have added the following note in red to the main text on Page 6:

Without optimization, our Ti-(0.34-0.50)O-(3.17-3.32)Fe alloys printed within the processing window exhibited tensile ductility (ϵ_f) from $9.0 \pm 0.5\%$ to $21.9 \pm 2.2\%$ (the change in ϵ_f is not due to porosity, Supplementary Note 2) and ultimate tensile strength (σ_{UTS}) from 1034 ± 9 to 1194 ± 8 MPa (Extended Data Table 1).

Fig. R11 Prior- β grain structures observed in the surface layer of each sample of the Ti-0.35O-3Fe alloy printed under different DED conditions. (a) 500 W – 800 mm/min – 120 s. (b) 500 W – 800 mm/min – 60 s. (c) 500 W – 800 mm/min – 0 s. (d) 500 W – 600 mm/min – 15 s. Other DED parameters are listed in Extended Data Table 2.

Table R4 Further assessment of the predictability of Simufact Welding DED — comparison of the predicted prior- β grain size using Eq. (4) and the measured mean prior- β grain size in the top surface layer of Ti-0.35-3Fe samples printed under four DED conditions.

DED condition	Predicted surface cooling rate \dot{T} ($^{\circ}\text{C/s}$)	Predicted prior- β grain size using Eq. (4) (μm)	Measured prior- β grain size (μm)
500 W-800 mm/min-120 s	13291	55	41 ± 20
500 W-800 mm/min-60 s	12551	59	64 ± 30
500 W-800 mm/min-0 s	9981	76	75 ± 43
500 W-600 mm/min-15 s	9695	78	76 ± 35

References

- [1] Kurz, W. and Trivedi, R., 1994. Rapid solidification processing and microstructure formation. *Materials Science and Engineering: A*, 179, pp.46-51.
- [2] Pinomaa, T., Laukkanen, A. and Provatas, N., 2020. Solute trapping in rapid solidification. *MRS Bulletin*, 45(11), pp.910-915.
- [3] Song, R., Dai, F. and Wei, B., 2011. Dendritic growth and solute trapping in rapidly solidified Cu-based alloys. *Science China Physics, Mechanics and Astronomy*, 54(5), pp.901-908.
- [4] Zhou, Q., Zhang, X.Z., Tang, H.P. and Qian, M., 2023. Electron beam additively manufactured Ti–1Al–8V–5Fe alloy: In-situ precipitation hardening, tensile properties and fracture characteristics. *Materials Science and Engineering: A*, p.144639.
- [5] Broderick, T.F., Jackson, A.G., Jones, H. and Froes, F.H., 1985. The effect of cooling conditions on the microstructure of rapidly solidified Ti-6Al-4V. *Metallurgical Transactions A*, 16(11), pp.1951-1959.
- [6] Mitchell, A., Kawakami, A. and Cockcroft, S.L., 2006. Beta fleck and segregation in titanium alloy ingots. *High Temperature Materials and Processes*, 25(5-6), pp.337-349.
- [7] Koziel, T., 2015. Estimation of cooling rates in suction casting and copper-mould casting processes. *Archives of Metallurgy and Materials*, 60(2A), pp.767-771.
- [8] Srivastava, R.M., Eckert, J., Löser, W., Dhindaw, B.K. and Schultz, L., 2002. Cooling rate evaluation for bulk amorphous alloys from eutectic microstructures in casting processes. *Materials Transactions*, 43(7), pp.1670-1675.
- [9] Liu, C., 2020. Investigating the Effect of Cooling Rate on the Secondary Dendrite Arm Spacing in Titanium Alloys. DOI: doi.org/10.14264/04e2fca. The University of Queensland.
- [10] Stefanescu, D.M., 2015. *Science and engineering of casting solidification*. P. 23. Springer.
- [11] Kim, D.H. and Lee, C.M., 2021. Experimental investigation on machinability of titanium alloy by laser-assisted end milling. *Metals*, 11(10), p.1552.
- [12] Wang, H., Yang, S. and Wei, B., 2012. Density and structure of undercooled liquid titanium. *Chinese Science Bulletin*, 57(7), pp.719-723.
- [13] Ozawa, S., Kudo, Y., Kuribayashi, K., Watanabe, Y. and Ishikawa, T., 2017. Precise density measurement of liquid titanium by electrostatic levitator. *Materials Transactions*, 58(12), pp.1664-1669.

Referee #2 (Remarks to the Author):

The authors sufficiently addressed the reviewer comments.

The writing in the supplementary notes is a bit informal (e.g., “This is remarkable and exciting”), and it is suggested to make the writing more factual.

I would suggest the authors reduce the number of significant figures in the tables in the supplementary notes.

The following sentence in supplementary notes is unclear: “They are not expected to exhibit outstanding tensile properties, but they can still be used for low-end applications.”

Response: Thank you for these further comments and suggestions.

We have discarded the statement “This is remarkable and exciting”. In addition, we have removed the word “exciting” (two more places) and all unnecessary adjectives.

In the supplementary notes, we have reduced the number of significant figures after the decimal point to three and used exponential notation (e.g. 10^{-3}) for particularly significant numbers.

We have rephrased “They are not expected to exhibit outstanding tensile properties, but they can still be used for low-end applications” to “Consequently, the resulting tensile mechanical properties are usually only comparable to their as-cast counterparts”.

Thank you again.